# WARP: Weight Teleportation for Attack-Resilient Unlearning Protocols

**Mohammad M Maheri**[1]  **Xavier Cadet**[2]  **Peter Chin**[2]  **Hamed Haddadi**[1]

[1]Imperial College London   [2]Dartmouth College

## Abstract

Approximate machine unlearning aims to efficiently remove the influence of specific data points from a trained model, offering a practical alternative to full retraining. However, it introduces privacy risks: an adversary with access to pre- and post-unlearning models can exploit their differences for membership inference or data reconstruction. We show these vulnerabilities arise from two factors: large gradient norms of *forget-set* samples and the close proximity of unlearned parameters to the original model. To demonstrate their severity, we propose unlearning-specific membership inference and reconstruction attacks, showing that several state-of-the-art methods (e.g., NGP, SCRUB) remain vulnerable. To mitigate this leakage, we introduce WARP, a *plug-and-play teleportation defense* that leverages neural network symmetries to reduce *forget-set* gradient energy and increase parameter dispersion while preserving predictions. This reparameterization obfuscates the signal of forgotten data, making it harder for attackers to distinguish forgotten samples from non-members or recover them via reconstruction. Across six unlearning algorithms, our approach achieves consistent privacy gains, reducing adversarial advantage (AUC) by up to 64% in black-box and 92% in white-box settings, while maintaining accuracy on retained data. These results highlight teleportation as a general tool for reducing attack success in approximate unlearning.

## 1 Introduction

Machine unlearning (MU) aims to enforce the "right to be forgotten" by updating a trained model so that a designated *forget-set* has no influence Bourtoule et al. (2021); Zhao et al. (2024). The ideal outcome matches retraining from scratch on the remaining *retain-set*, with both the model's parameters and predictions unaffected by the forgotten data, and without degrading generalization. A primary motivation for machine unlearning is to ensure privacy compliance for sensitive information Wang et al. (2025a). Once personal data is used for training, models may memorize specific details Ravikumar et al. (2024a), creating risks of privacy breaches Bourtoule et al. (2021); Carlini et al. (2022b). Unlearning addresses this by eliminating such traces, preventing exposure. The most direct solution is retraining from scratch without the *forget set*, but this is computationally prohibitive. *Exact Unlearning* methods such as SISA Bourtoule et al. (2021) reduce cost by modifying training to allow provable deletion, but they require proactive deployment and add overhead. To avoid full retraining, *Approximate Unlearning* methods finetune the original model to forget the target data while preserving utility Kurmanji et al. (2023); Chundawat et al. (2023a); Golatkar et al. (2020); Thudi et al. (2022), trading computational efficiency against formal guarantees.

At the same time, ML models are vulnerable to privacy attacks Rigaki & Garcia (2023). In Membership Inference Attacks (MIA), an adversary determines whether a given sample was part of the training set Shokri et al. (2017). In Data Reconstruction Attacks (DRA), the adversary seeks to recover raw data (or a close approximation) from model outputs or parameters Yin et al. (2021); Li et al. (2022); Jeon et al. (2021); Fang et al. (2023). These attacks have been demonstrated in both black-box (access to outputs) and white-box (access to weights) settings Nasr et al. (2019).

Ironically, MU itself can leak the very data it aims to erase. Given access to both the original and unlearned models, an adversary can mount differencing attacks Hu et al. (2024); Bertran et al. (2024), which substantially improve reconstruction success. Even models previously resistant to MIAs can become vulnerable once deletion is performed Bertran et al. (2024); Chen et al. (2021). The key

observation is that the parameter difference between the two models approximates the gradient of the forgotten sample (up to second-order terms), effectively releasing it to the adversary. Gradient inversion techniques, as in federated learning Geiping et al. (2020), can then reconstruct the forgotten data. Thus, approximate unlearning methods, especially gradient-ascent variants Kurmanji et al. (2023), can inadvertently compromise privacy instead of ensuring it.

In this work, we aim to strengthen MU against privacy attacks by characterizing two key factors driving leakage. The first, illustrated in Figure 1, is that a forgotten sample's privacy risk correlates with its gradient norm in the original model. Intuitively, samples with large gradient magnitudes during training or finetuning induce stronger parameter changes when removed, making them more detectable via MIA and more exploitable for reconstruction Ye et al. (2023).

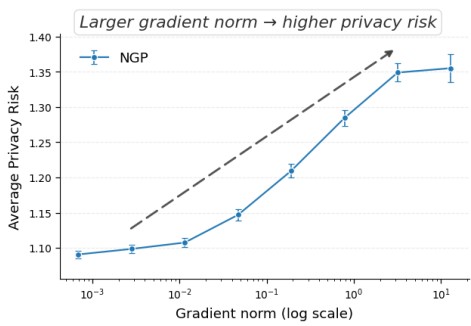

Second, as shown in prior work Thudi et al. (2022); Kurmanji et al. (2023), most approximate unlearning methods make minor parameter updates, typically by maximizing the *forget-set* loss while keeping retain-set accuracy stable. This keeps the unlearned model close to the original, so the parameter difference encodes information about the forgotten data. In gradient-ascent–based methods Kurmanji

Figure 1: Privacy risk vs. gradient norms of *forget-set* samples, measured with U-LiRA.

et al. (2023); Chundawat et al. (2023a), this difference is essentially the *forget-set* gradient. Recent studies confirm that such updates expose information equivalent to a single gradient step on the forgotten sample Bertran et al. (2024), which attackers can invert to reconstruct it.

To mitigate these risks, we propose WARP, a plug-and-play defense that integrates into existing unlearning algorithms without training-time statistics. Our method leverages neural network teleportation Armenta et al. (2023), exploiting parameter-space symmetries (e.g., rescaling or permutation) that preserve predictions. By applying selective teleportation steps before or during unlearning, we reduce *forget-set* gradient norms while injecting symmetry-preserving randomness. This yields unlearned models that retain accuracy yet are displaced in parameter space, making it harder for an attacker to disentangle forgetting from teleportation. Consequently, membership inference and reconstruction attacks are significantly weakened, as shown in Sections 4.2, 4.3, and 4.4.

Our **contributions** are summarized as follows:

- **Tailored privacy attacks.** We design MIA and DRA for the unlearning setting, where the adversary compares pre- and post-unlearning models. These attacks show that leading methods remain vulnerable, as parameter updates still expose information about the *forget-set*.

- **Symmetry-based defense.** We propose WARP, a plug-and-play defense that, building on existing teleportation and symmetry constructions, applies loss-preserving transformations to reduce *forget-set* gradient norms and increase parameter dispersion, thereby obscuring the signal exploited in reconstruction and inference, while remaining agnostic to the particular symmetry mechanism used to realize these transformations. WARP integrates into gradient-based post-hoc unlearning algorithms without requiring training-time statistics.

- **Comprehensive evaluation.** We evaluate our attacks and defense across three datasets—CIFAR-10, Tiny-ImageNet, and ImageNet-1K—using ResNet-18 and ViT-B/16 models under both black-box and white-box settings. Results across multiple unlearning algorithms show that teleportation consistently reduces privacy leakage while preserving accuracy on the retain set.

Overall, our work reframes unlearning privacy risk through the lens of *gradient norm reduction* and links it to neural network symmetry, an underexplored optimization principle that provides a conceptual foundation for designing unlearning algorithms with stronger resilience to privacy attacks. Related work is discussed in more detail in Appendix A. The implementation[1] is publicly available.

---

[1]Our code is available at `https://github.com/mammadmaheri7/WARP_Unlearning`.

## 2  THREAT MODEL

We consider a strong adversary performing *sample-wise membership inference*, distinguishing whether a sample belongs to the *forget-set* $\mathcal{D}_f$ or the *test set* $\mathcal{D}_{\text{test}}$. The attacker has access to both the pre- and post-unlearning models.

**Attacker Capabilities.**  The attacker has full access to both the original $\theta^{\text{org}}$ and unlearned model $\theta^u$, as well as complete knowledge of the unlearning algorithm $\mathcal{A}$unlearn and its hyperparameters $\mathcal{H}$unlearn (e.g., optimizer, learning rate, update steps, retain-set size).

We consider two settings: **Black-box** — the attacker queries outputs $f(x; \theta^u)$. **White-box** — the attacker additionally accesses full internals of both models $(\theta^{\text{org}}, \theta^u)$, including weights.

**Attack Objective**  Given a sample $(x, y)$ from either the *forget-set* $\mathcal{D}_{\text{forg}}$ or the held-out test set $\mathcal{D}_{\text{test}}$, the attacker computes a score $A'(x, y)$ and predicts membership as $A(x, y) = \mathbb{I}[A'(x, y) > \tau]$, where $\mathbb{I}[\cdot]$ is the indicator function and $\tau$ is a decision threshold. The attacker seeks a high true positive rate (TPR) on forgotten samples while maintaining a low false positive rate (FPR) on test samples. This directly measures privacy risk: if membership can be reliably inferred, incomplete unlearning is exposed and the forgotten samples identified. Unlike prior work Bourtoule et al. (2021); Maheri et al. (2025a), our goal is to audit unlearning algorithms from a *privacy perspective*, rather than evaluating indistinguishability between approximate and exact unlearning outcomes.

## 3  METHODOLOGY

### 3.1  PRIVACY ATTACKS

To systematically evaluate privacy leakage in unlearning, we consider two complementary classes of attacks: *membership inference* and *data reconstruction*.

**Black-box (U-LiRA).**  For the black-box setting, we adopt U-LiRA (Hayes et al., 2025), an adaptation of LiRA (Carlini et al., 2022a) to unlearning. U-LiRA leverages shadow models trained and unlearned with the same algorithm as the target, yielding a strong adaptive baseline for auditing privacy. We defer full algorithmic details to Appendix B.

**White-box (Gaussian Gradient–Difference).**  In the white-box setting, we extend the Gaussian gradient–difference framework of Leemann et al. (2023) to the unlearning case by contrasting gradients computed on both the original and unlearned models. This contrast provides a powerful signal of residual membership leakage when both model versions are available to attacker. The detailed proposed formulation and test statistic are presented in Appendix C.

**Reconstruction Attack in Unlearning.**  We develop a *white-box* reconstruction attack tailored to approximate unlearning with retain-set updates. Let $\Delta\theta = \theta^u - \theta^{\text{org}}$ be the observed parameter change after one unlearning stage (possibly aggregating multiple optimizer steps). As in gradient inversion, we seek an input whose parameter-gradient aligns with a target vector; here the natural target is $\Delta\theta$. Our baseline (single-sample) objective is:

$$\hat{x}, \hat{y} \in \arg\min_{x,y} \mathcal{D}\big(\nabla_\theta \ell\big(f(x; \theta^{\text{org}}), y\big), \Delta\theta\big), \tag{1}$$

where $\ell$ is the training loss, $f(\cdot; \theta)$ the network, and $\mathcal{D}$ a distance (e.g., $\ell_2$ or negative cosine).

With approximate unlearning, the update $\Delta\theta$ mixes retain and forget gradients. For a forget example $(x_f, y_f)$ and a retain minibatch $\mathcal{B}_r$,

$$\Delta\theta \approx -\eta\Big(g_r - \alpha\, g_f\Big), \qquad g_r = \frac{1}{|\mathcal{B}_r|} \sum_{(x_r, y_r) \in \mathcal{B}_r} \nabla_\theta \ell(f(x_r; \theta^{\text{org}}), y_r), \quad g_f = \nabla_\theta \ell(f(x_f; \theta^{\text{org}}), y_f), \tag{2}$$

with effective step size $\eta$ and ascent weight $\alpha > 0$. Directly targeting $\Delta\theta$ in equation 1 is therefore confounded by $g_r$. Even when equation 1 is instantiated with state-of-the-art gradient inversion

methods, naively inverting the unfiltered update $\Delta\theta$ remains ineffective, producing low accuracy of the reconstruction (see Section 4, Table 2).

Let $G_{\text{org}} = [g(b_i; \theta^{\text{org}})]_{i=1}^m$ and $G_u = [g(b_i; \theta^u)]_{i=1}^m$ be gradient snapshots on a small probe set drawn from the training distribution. We compute thin SVDs, $G_{\text{org}} = U_{\text{org}}\Sigma_{\text{org}}V_{\text{org}}^\top$ and $G_u = U_u\Sigma_u V_u^\top$, and keep the top-$k$ left singular vectors to obtain orthonormal bases (columns) for the dominant gradient subspaces. Define the *orthogonal projectors*

$$\Pi_{\text{org}} = U_{\text{org}}U_{\text{org}}^\top, \qquad \Pi_u = U_u U_u^\top, \qquad \Pi_u^\perp = I - \Pi_u.$$

Unlearning attenuates the forget component, so retain gradients are expected to persist in both models, whereas the forget component is prominent in $\theta^{\text{org}}$ but suppressed in $\theta^u$. We therefore *orthogonalize* the update against the unlearned subspace and keep only directions supported by the original model:

$$\tilde{g}_f \;=\; \Pi_{\text{org}}\,\Pi_u^\perp\left(-\tfrac{1}{\eta}\,\Delta\theta\right). \tag{3}$$

Intuitively, $\Pi_u^\perp$ removes directions consistent with retain gradients that remain after unlearning, while $\Pi_{\text{org}}$ preserves directions active before unlearning where the forget signal resides. If the retain subspace is well captured, then $\Pi_u^\perp g_r \approx 0$ and $\Pi_{\text{org}}\Pi_u^\perp(\alpha g_f) \approx \alpha g_f$, yielding a high-SNR estimate of the forget gradient.

We reconstruct the forgotten sample by solving the filtered inversion:

$$\hat{x}_f, \hat{y}_f \;\in\; \arg\min_{x,y}\; \mathcal{D}\big(\nabla_\theta\ell\big(f(x; \theta^{\text{org}}), y\big), \tilde{g}_f\big), \tag{4}$$

with optional priors or constraints on $(x, y)$. In practice, we choose $k$ to retain a fixed fraction of gradient energy (e.g., 90–95%), which stabilizes the projectors and reliably isolates the forget component via orthogonalization. We empirically validate that orthogonal subspace filtering boosts reconstruction success across models and datasets; see Section 4.4 and Appendix Table 3.

## 3.2 WARP (Teleportation-based Defense)

**Motivation I: Parameter closeness increases privacy leakage.** We formulate post-hoc unlearning as minimizing a composite objective that balances forgetting on $\mathcal{D}_f$ with utility on $\mathcal{D}_r$:

$$\min_\theta \; \underbrace{\ell_f\big(\theta \mid \mathcal{D}_f\big)}_{\text{Forget}} + \lambda \underbrace{\ell_r\big(\theta \mid \mathcal{D}_r\big)}_{\text{Retain}}, \qquad \lambda \geq 0, \tag{5}$$

where $\theta$ denotes model parameters; $\ell_f$ is any differentiable *forgetting surrogate* that penalizes high confidence or reduces fidelity on $\mathcal{D}_f$ (e.g., loss-inflation, uniform/soft labels, margin expansion); and $\ell_r$ is the standard training/consistency loss on $\mathcal{D}_r$ to preserve performance. The trade-off coefficient $\lambda$ controls how strongly the unlearning step remains anchored to the retain-set: larger $\lambda$ keeps $\theta^u$ closer to $\theta^{\text{org}}$, preserving accuracy but reducing the parameter shift introduced by forgetting. A first–order optimizer with mini-batches $\mathcal{B}_f \subset \mathcal{D}_f$ and $\mathcal{B}_r \subset \mathcal{D}_r$ yields the iterative update

$$\theta_{t+1} = \theta_t - \eta_t\Big(\nabla_\theta\ell_f\big(\theta_t \mid \mathcal{B}_f\big) + \lambda\,\nabla_\theta\ell_r\big(\theta_t \mid \mathcal{B}_r\big)\Big), \tag{6}$$

which encompasses common post-training approximate unlearning schemes; for instance, "negative-gradient" methods are recovered by taking $\ell_f(\cdot) = -\ell_{\text{train}}(\cdot)$ (i.e., ascent on the standard training loss over $\mathcal{D}_f$), whereas rehearsal/consistency-based approaches instantiate $\ell_r$ with supervised loss or distillation on $\mathcal{D}_r$ Thudi et al. (2022); Kurmanji et al. (2023); Chundawat et al. (2023a).

Because equation 5 explicitly regularizes utility on $\mathcal{D}r$ and is optimized with small steps and early stopping on $\mathcal{D}f$, the resulting unlearned parameters $\theta^u$ typically remain *close* to the original $\theta^{\text{org}}$ in parameter space. The displacement $\Delta\theta = \theta^u - \theta^{\text{org}}$ is well-approximated (to first order) by a weighted combination of gradients on the *forget-set*, mildly contaminated by retain gradients Thudi et al. (2022); Kurmanji et al. (2023); Huang et al. (2024). This proximity creates a privacy attack surface: An adversary with access to $(\theta^{\text{org}}, \theta^u)$ can leverage $\Delta\theta$ to perform membership inference or gradient-based reconstruction of $\mathcal{D}_f$ Hu et al. (2024); Bertran et al. (2024), motivating the defenses applied over unlearning algorithms.

**Motivation II: Gradient norm and curvature amplify leakage.** Recent evidence suggests that the per-sample gradient trajectory is a strong predictor of privacy vulnerability. Tobaben et al. (2024) show that training examples that accumulate larger gradient norms during optimization are significantly more prone to MIA, reflecting the intuition from differential privacy that each update's privacy loss scales with gradient magnitude. Complementing this, Ravikumar et al. (2024b) demonstrate that curvature around training samples—captured via local sharpness of the loss—serves as a reliable discriminator between members and non-members, with sharper regions implying higher membership exposure. These findings aligns with theoretical analyses such as Ye et al. (2023), who prove that large per-sample gradients at initialization inflate the KL divergence between neighboring training trajectories, directly increasing the sample's privacy risk. Motivated by this, we hypothesize that approximate unlearning inherits the same vulnerability: samples with higher gradient norms tend to push parameters towards sharper local extrema during both training and unlearning, thereby overshooting the target update and leaving a stronger privacy footprint. Our experiments (Fig.1) confirm this intuition, revealing a clear correlation between a sample's gradient norm in the original model and its susceptibility to membership inference after unlearning.

To simultaneously address (i) the parameter–space proximity that enables differencing and (ii) the gradient–norm driver of leakage, we leverage *loss-invariant symmetries* of deep networks.

**Symmetry framework.** Let $\mathcal{G}$ denote a set of symmetry transformations acting on parameters $\theta$ (and, when needed, internal representations) such that the task loss is invariant: $\mathcal{L}(X, \theta) = \mathcal{L}(g(X, \theta))$ for all $g \in \mathcal{G}$ Zhao et al. (2022; 2023); Armenta et al. (2023); Simsek et al. (2021). A *teleportation* step chooses $g$ and updates $\theta \leftarrow g \cdot \theta$, moving within the loss level set. In our defense, we select $g$ to reduce the gradient norm of the *forget-set* while preserving utility on the retain-set:

$$g^\star \in \arg\min_{g \in \mathcal{G}} \left\{ \underbrace{\sum_{(x,y) \in \mathcal{D}_f} \|\nabla_\theta \ell(f(x; g \cdot \theta), y)\|_2^2}_{\text{shrink forget-set gradients}} - \beta \underbrace{\|g \cdot \theta - \theta\|_2^2}_{\text{increase parameter dispersion}} \right\} \quad (7)$$

$$\text{s.t.} \quad \ell_r(g \cdot \theta \,|\, \mathcal{D}_r) \leq \ell_r(\theta \,|\, \mathcal{D}_r) + \varepsilon.$$

with trade-off $\beta \geq 0$ and tolerance $\varepsilon \geq 0$. The first term reduces squared gradient norms of forget examples (Motivation II); the dispersion term adds symmetry-preserving randomness, displacing parameters from $\theta^{\mathrm{org}}$ (Motivation I); the constraint preserves retain performance.

WARP operates on an abstract prediction-preserving symmetry map $T_\phi$, and any such symmetry family can instantiate the framework. In practice, we use two concrete realizations—the retain–null-space projection introduced in the next paragraph, and the change-of-basis teleportation detailed in Appendix D—to illustrate this generality. To complement this algorithmic view, Appendix O develops teleportation-aware information-theoretic bounds on gradient-based reconstruction, showing how injecting symmetry-induced noise via $T_\phi$ expands the symmetry orbit and provably increases the expected reconstruction error for attackers observing $(\theta^{\mathrm{org}}, \theta^u)$.

**Primary instantiation: teleportation with retain null-space projection.** We first describe one convenient way to instantiate $T_\phi$ using retain–null-space projections Wu et al. (2025). To optimize equation 7 efficiently on modern architectures without explicit group actions, we adopt *teleportation with input null-space gradient projection* Wu et al. (2025) and instantiate it using the recent projector formulation that keeps updates on the loss-invariant level set by per-layer projections onto the input null space (thus leaving the task loss unchanged up to numerical error). Concretely, define the *teleportation loss*

$$\mathcal{L}_{\mathrm{tel}}(\theta) = \sum_{(x,y) \in \mathcal{B}_f} \left\| \nabla_\theta \ell(f(x; \theta), y) \right\|_2^2 - \beta \|\theta - \theta^{\mathrm{org}}\|_2^2,$$

where $\mathcal{B}_f$ is a minibatch from $\mathcal{D}_f$. Let $R_\ell$ be the per-layer representation matrix from a *retain* minibatch (layer-$\ell$ inputs), with thin SVD $R_\ell = U_\ell \Sigma_\ell V_\ell^\top$. We keep the top-$k$ left singular vectors $B_\ell = U_{\ell, 1:k}$ to span the retain subspace and define the orthogonal projector onto its complement $\Pi_\ell^\perp = I - B_\ell B_\ell^\top$. A teleportation step then applies the layer-wise update

$$W_\ell^{t+1} \leftarrow W_\ell^t - \eta_{\mathrm{tel}} \, \Pi_\ell^\perp \left( \nabla_{W_\ell} \mathcal{L}_{\mathrm{tel}}(\theta^t) \right) \quad (8)$$

which (i) *reduces* the forget-set gradient norms by descending on $\mathcal{L}_{\text{tel}}$, (ii) *preserves* the function on the retain-set by restricting motion to the retain-orthogonal subspace. The projection operator in equation 8 corresponds to the input-null-space projector. This is implemented by subtracting the component in the subspace of the core gradient, leaving only the residual for the teleport step.

To align the invariance with utility preservation, we compute $B_\ell$ *only from retain data*. Let $R_\ell(\mathcal{D}_{\text{r}}) = [\phi_\ell(x)]_{x \in \mathcal{B}_{\text{r}}}$ denote the matrix formed by stacking the layer-$\ell$ inputs for a retain minibatch $\mathcal{B}_{\text{r}}$. Then:

$$R_\ell(\mathcal{D}_{\text{r}}) = U_\ell \Sigma_\ell V_\ell^\top, \qquad B_\ell = U_{\ell,1:k}, \qquad \Pi_\ell^\perp = I - B_\ell B_\ell^\top. \tag{9}$$

We set $k$ to capture a fixed fraction of retain variance (typically $95\%$–$99\%$) and apply the resulting projectors in equation 8. This confines each teleport step to the retain-orthogonal subspace, stabilizing predictions on $\mathcal{D}_{\text{r}}$ while suppressing gradient energy on $\mathcal{D}_{\text{f}}$. Since $\Pi_\ell^\perp$ removes directions spanned by retain representations, suitable choices of rank $k$ and step size $\eta_{\text{tel}}$ ensure that

$$\left| \ell_{\text{r}}(g \cdot \theta \,|\, \mathcal{D}_{\text{r}}) - \ell_{\text{r}}(\theta \,|\, \mathcal{D}_{\text{r}}) \right| \leq \varepsilon,$$

which matches the constraint below equation 7; in practice, prediction drift on $\mathcal{D}_{\text{r}}$ remains within numerical tolerance (see Appendix P for hyperparameter sensitivity). To underline that WARP is not tied to retain–null-space projections, Appendix D instantiates $T_\phi$ using the SVD-free change-of-basis symmetries introduce in Armenta et al. (2023).

**Plug-and-play scope.** Teleportation is interleaved with the standard unlearning update equation 6, requiring no training-time per-sample gradients or stored statistics. The update equation 8 is applied at intervals $t \in K \subset 0, \dots, T-1$ (e.g., every $S$ steps), keeping *forget-set* gradient norms low while preserving retention performance. The full algorithm appears in Appendix K.

## 4 EXPERIMENTS

We now empirically evaluate the proposed teleportation-based defense across multiple unlearning algorithms, datasets, and model architectures. Our experiments are designed to answer the following research questions: (i) How vulnerable are state-of-the-art unlearning algorithms to privacy attacks under both black-box and white-box threat models? (ii) To what extent does teleportation reduce membership and reconstruction leakage without sacrificing utility on the retain-set?

**Experimental Setup.** We conduct experiments on CIFAR-10, Tiny-ImageNet, and ImageNet-1K. On CIFAR-10 we use ResNet-18, while on ImageNet we evaluate ViT-B/16, covering both convolutional and transformer models. All models are trained with SGD and standard augmentation. Following prior work Kurmanji et al. (2023); Chundawat et al. (2023a), forget sets $\mathcal{D}_f$ are sampled as roughly $1\%$ of training data per class, with retain sets $\mathcal{D}_r$ comprising the rest.

**Baselines.** We benchmark six representative unlearning algorithms—NEGGRAD+ Kurmanji et al. (2023), SCRUB Kurmanji et al. (2023), SALUN Fan et al. (2023), PGU Hoang et al. (2024), BADTEACHER Chundawat et al. (2023a), and SRF-ON Huang et al. (2024)—covering paradigms of gradient ascent, regularization, saliency, projection, and distillation. Full details are in Appendix E.

### 4.1 OVERVIEW EFFECTIVENESS OF WARP

Figure 2 summarizes privacy and utility across six unlearning methods with and without our plug-in defense. Each radar chart reports black-box membership inference risk (AUC and TPR at low FPR), accuracy on the most-memorized subset, white-box membership inference risk (AUC and TPR at low FPR), and standard test accuracy. The most-memorized subset is selected following our U-LiRA protocol in Sec. 4.2, motivated by prior findings that highly memorized samples carry elevated unlearning risk Naderloui et al. (2025). For visualization, all metrics are min–max normalized across methods. Privacy metrics in which lower is better are inverted by plotting $1 - $ metric, so that larger polygons correspond to stronger privacy, while higher test accuracy remains preferable.

Three key observations emerge. First, no unlearning algorithm dominates across all axes. For instance, SF performs well under black-box auditing but is weaker under white-box auditing and in

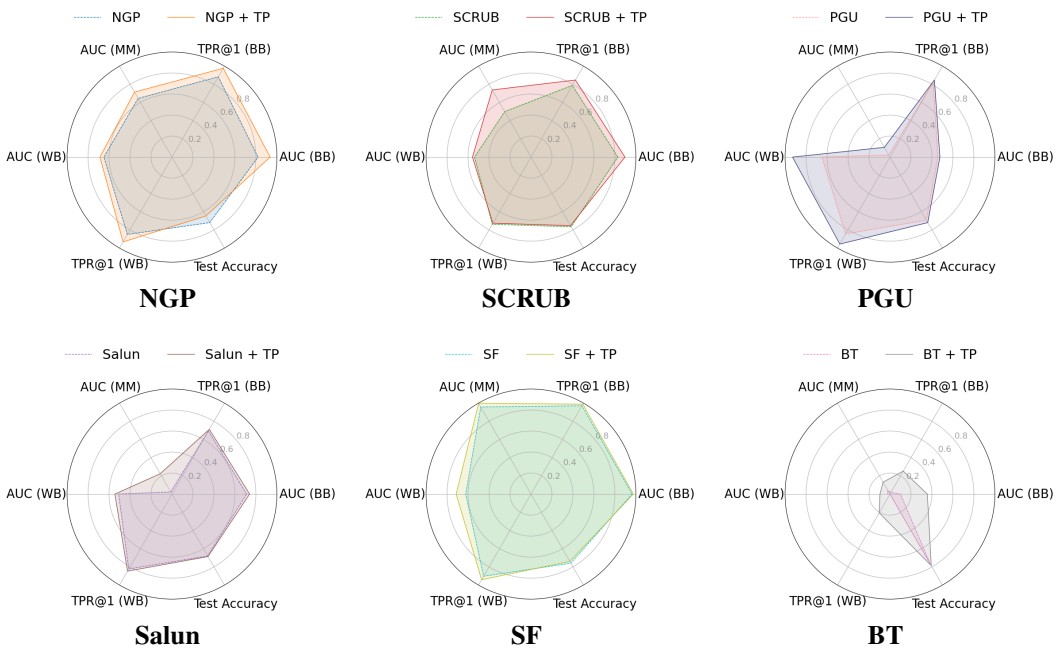

Figure 2: Comparison of unlearning vs. teleportation across six unlearning methods.

test accuracy, illustrating the necessity of evaluating under both threat models. Second, algorithms that appear robust under black-box evaluation such as NGP and SF still exhibit substantial leakage under our white-box test, underscoring the importance of auditing with gradient- or weight-based evidence. Third, adding our symmetry-based teleportation module, instantiated via retain null-space projection, consistently improves privacy across both black-box and white-box metrics while maintaining utility. In some cases, such as BT and SF, teleportation even improves test accuracy. The only noticeable accuracy drop occurs for NGP (about one percentage point), for which we provide a detailed privacy–utility trade-off analysis in Appendix I. The runtime overhead of teleportation is analyzed separately in Appendix J, and Appendix P presents ablations showing that WARP's performance does not hinge on fragile choices of teleportation hyperparameters. Overall, these results demonstrate that the proposed defense empirically reduces attack success consistently and effectively across a diverse set of unlearning algorithms and threat models. For completeness, we also compare WARP against the strongest noise-based alternative, namely projected DP–Langevin unlearning Chien et al. (2024b), using its formally calibrated update rule; the full comparison is provided in Appendix M.

## 4.2 U-LiRA (BLACK-BOX)

We evaluate our teleportation defense with U-LiRA Hayes et al. (2025), a state-of-the-art black-box unlearning auditor. Following Deep Unlearn Cadet et al. (2024), we train $T = 64$ shadow models with 10 random forget sets each. To model a strong adaptive adversary, shadows use the same unlearning algorithm, teleportation, and hyperparameters as the target, reducing proxy–target miscalibration Cretu et al. (2023). Details of U-LiRA appear in Appendix B.

As emphasized in prior work Carlini et al. (2022a), the most informative regime is low false-positive rates (FPR), where practical attacks must operate. We therefore report AUC as well as TPR@0.1, TPR@1, and TPR@5, which capture attacker success in this stringent regime. In addition, following RULI Naderloui et al. (2025), we stratify the *forget-set* by *memorization* (ranked by training confidence) and evaluate U-LiRA on the most–memorized slice. These points carry elevated privacy risk, so we report low-FPR TPR on this subset alongside aggregate metrics.

Table 1 shows that adding our teleportation plug-in reduces black-box membership leakage across all methods, on both the full *forget-set* and the most-memorized slice, with the largest gains at low FPR. For example, NGP's TPR@1 nearly halves (0.030→0.014), SCRUB's memorized-slice AUC

Table 1: **Privacy (Black-box) with and without WARP.** Reported are risks on *all forget samples* and the *most–memorized* 1% (AUC, TPR@0.1/1/5%), plus test accuracy. Each row shows baseline, WARP, and relative improvement (%).

| Method | All samples (BB) | | | | Most–memorized (top 1%) | | | | Acc. |
| | AUC | TPR@0.1 | TPR@1 | TPR@5 | AUC | TPR@0.1 | TPR@1 | TPR@5 | Test |
|---|---|---|---|---|---|---|---|---|---|
| NGP (base) | 0.545 | 0.012 | 0.030 | 0.077 | 0.649 | 0.058 | 0.157 | 0.277 | **0.808** |
| + WARP | **0.516** | **0.003** | **0.014** | **0.055** | **0.598** | **0.015** | **0.082** | **0.206** | 0.797 |
| Improvement (%) | 64.4 | 81.8 | 80.0 | 81.5 | 34.2 | 75.4 | 51.0 | 31.3 | -5.7 |
| SCRUB (base) | 0.543 | 0.020 | 0.047 | 0.092 | 0.710 | 0.086 | 0.227 | 0.397 | **0.815** |
| + WARP | **0.526** | **0.015** | **0.036** | **0.078** | **0.610** | **0.041** | **0.119** | **0.213** | 0.813 |
| Improvement (%) | 39.5 | 26.3 | 29.7 | 33.3 | 47.6 | 52.9 | 49.8 | 53.0 | -1.1 |
| PGU (base) | 0.636 | 0.024 | 0.040 | **0.098** | 0.910 | 0.201 | 0.511 | 0.706 | 0.804 |
| + WARP | **0.631** | **0.018** | **0.036** | 0.104 | **0.875** | **0.160** | **0.431** | **0.663** | **0.808** |
| Improvement (%) | 3.7 | 26.1 | 13.3 | -12.5 | 8.5 | 20.5 | 16.0 | 6.6 | +2.0 |
| Salun (base) | 0.572 | 0.020 | 0.062 | 0.121 | 0.910 | 0.129 | 0.321 | 0.520 | 0.802 |
| + WARP | **0.565** | **0.019** | **0.059** | **0.113** | **0.826** | **0.107** | **0.264** | **0.487** | **0.803** |
| Improvement (%) | 9.7 | 5.3 | 5.8 | 11.3 | 20.5 | 17.2 | 18.3 | 7.0 | +0.5 |
| SF (base) | 0.509 | 0.004 | 0.015 | 0.056 | 0.518 | 0.089 | 0.034 | 0.079 | **0.814** |
| + WARP | **0.506** | **0.002** | **0.012** | **0.051** | **0.501** | **0.006** | **0.026** | **0.068** | 0.811 |
| Improvement (%) | 33.3 | 66.7 | 60.0 | 83.3 | 94.4 | 94.3 | 33.3 | 37.9 | -1.6 |
| BT (base) | 0.725 | 0.000 | 0.177 | 0.287 | 0.902 | 0.119 | 0.295 | 0.582 | 0.816 |
| + WARP | **0.661** | 0.000 | **0.137** | **0.219** | **0.865** | **0.113** | **0.275** | **0.537** | **0.818** |
| Improvement (%) | 28.4 | – | 24.0 | 28.7 | 9.2 | 5.1 | 7.0 | 8.5 | +1.1 |

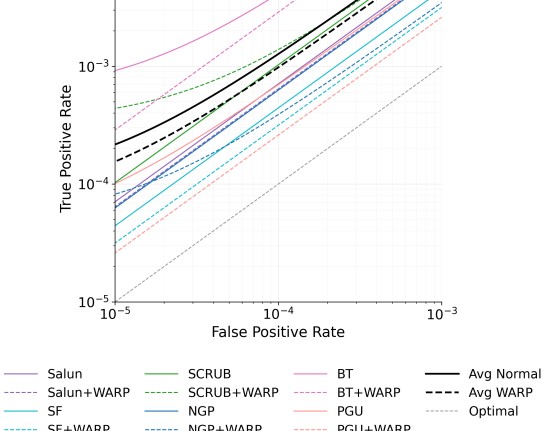

| Method | AUC | TPR@0.1 | TPR@1 | TPR@5 |
|---|---|---|---|---|
| NGP (base) | 0.642 | 0.004 | 0.034 | 0.139 |
| + WARP | **0.614** | **0.002** | **0.021** | **0.097** |
| Improvement (%) | 17.0 | 50.0 | 40.6 | 34.2 |
| SCRUB (base) | 0.700 | 0.011 | 0.102 | 0.287 |
| + WARP | **0.657** | **0.006** | **0.061** | **0.193** |
| Improvement (%) | 14.3 | 54.5 | 42.5 | 33.5 |
| PGU (base) | 0.659 | 0.007 | 0.064 | 0.215 |
| + WARP | **0.533** | **0.002** | **0.025** | **0.085** |
| Improvement (%) | 92.9 | 83.3 | 64.5 | 65.5 |
| Salun (base) | 0.721 | 0.008 | 0.069 | 0.230 |
| + WARP | **0.705** | **0.006** | **0.062** | **0.214** |
| Improvement (%) | 9.5 | 33.3 | 10.1 | 7.0 |
| SF (base) | 0.670 | 0.005 | 0.043 | 0.161 |
| + WARP | **0.629** | **0.003** | **0.030** | **0.124** |
| Improvement (%) | 29.2 | 50.0 | 34.9 | 23.2 |
| BT (base) | 0.938 | 0.037 | 0.346 | 0.809 |
| + WARP | **0.907** | **0.028** | **0.279** | **0.684** |
| Improvement (%) | 49.2 | 25.7 | 19.4 | 18.4 |

Figure 3: **White-box privacy with and without WARP.** Gaussian gradient–diff test on 640 unlearned models. ROC curves (left) and AUC/TPRs (right); full ROC plots are in Appendix F.

drops by 0.10 (0.710→0.610), and SF's AUC falls to near-random (0.501). Low-FPR TPR gains are often large even when aggregate AUC shifts are modest, showing that teleportation suppresses the high-confidence tails attacks exploit. Some methods remain leaky on memorized points, but teleportation frequently drives this slice close to random without hurting accuracy. Its impact is strongest on TPR@0.1 and TPR@1, as retain-null-space projection reduces forget gradients and shrinks extreme margins, weakening the rare signals enabling low-FPR success.

### 4.3 WHITE-BOX MIA

We evaluate the Gaussian gradient–difference test of Section C under the setup of Section 4, using ResNet-18 on CIFAR-10 and ViT-B/16 on Tiny-ImageNet (full ViT in Appendix H). For the null background we draw $m=1000$ non-members from $\mathcal{D}_{\text{test}}$, estimate $(\hat{\mu}, \hat{\Sigma})$ with ridge $\lambda=10^{-3}$, and restrict the test to the top-10% most-variant $\Delta(b)$ coordinates. Figure 3 shows ROC curves with and

Table 2: **Effect of teleportation defense** on reconstruction (ImageNet-1K, ResNet-18, NGP).

| Variant | PSNR (dB) ↑ | LPIPS (VGG) ↓ | LPIPS (Alex) ↓ | SSIM ↑ | Test MSE ↓ | Feat MSE ↓ |
|---|---|---|---|---|---|---|
| Ours (normal unlearning) | $10.74 \pm 0.31$ | $0.56 \pm 0.013$ | $0.34 \pm 0.015$ | $0.12 \pm 0.008$ | $0.10 \pm 0.007$ | $5.39 \pm 0.50$ |
| Ours + *WARP* | $7.38 \pm 0.40$ | $0.68 \pm 0.01$ | $0.46 \pm 0.02$ | $0.08 \pm 0.006$ | $0.21 \pm 0.02$ | $11.28 \pm 1.89$ |
| *Improvement of Defense (%)* | +45.5 | +21.2 | +26.1 | +31.6 | +52.4 | +52.2 |

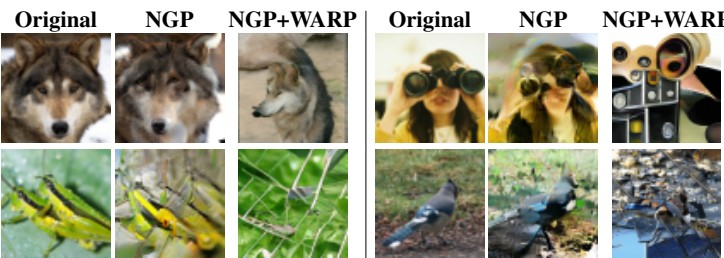

| **Original** | **NGP** | **NGP+WARP** | **Original** | **NGP** | **NGP+WARP** |

Figure 4: **Reconstructions under NGP vs. NGP+WARP.**

without teleportation (log–log for low-FPR). Across methods, teleported variants shift toward chance (TPR = FPR) and flatten between $10^{-5}$–$10^{-2}$ FPR, suppressing high-confidence tails. The strongest effect appears for BT and PGU, which show the largest AUC drops, while NGP, SF, and SALUN show smaller but consistent shifts. An exception is SCRUB, where teleportation lowers ROC above $10^{-3}$ FPR but slightly raises TPR at $< 10^{-3}$, due to knowledge distillation interacting with symmetry moves that amplify high-leverage directions. Overall, null-space teleportation reduces white-box evidence at low FPR, with a narrow corner case for SCRUB.

## 4.4    RECONSTRUCTION ATTACK RESULTS

We evaluate the white-box reconstruction attack of Section 3.1 on **ImageNet-1K** with **ResNet-18**, focusing on **NGP**. We reconstruct a *single* forgotten example and average over **100** uniformly sampled points. For each target we use a retain minibatch of size $|\mathcal{B}_r| = 5$. Subspace projectors are built per layer from probe gradients: we draw $m=100$ training samples to form $G_{\text{org}}, G_u$, compute thin SVDs, and keep rank $k$ preserving **90%** gradient energy. We then apply $\Pi_u^{\perp}$ and $\Pi_{\text{org}}$ layerwise to obtain the filtered target $\tilde{g}_f$. The attacker knows the label $y_f$ and optimizes equation 4 with a TV regularizer Geiping et al. (2020). The matching loss uses *masked* per-layer gradients: for each layer, all coordinates are kept and a weighted dot-product alignment is computed Fang et al. (2023).

**Effect of teleportation.**    Table 2 and Figure 4 compare reconstruction risk under standard NGP unlearning and its teleported variant using change-of-basis reparameterization. Despite negligible cost, this symmetry-based randomization disrupts reconstruction: even strong generative-prior attacks fail to recover meaningful features of forgotten data. Teleportation injects a symmetry component into $\Delta\theta$ that is nearly orthogonal to per-sample gradients Armenta et al. (2023), reducing alignment with the true forget gradient $g_f$ and driving gradient-matching toward low signal-to-noise optima. It also undermines our subspace-filtered attack (Eq. 3), since teleportation reshapes gradient subspaces so $U_{\text{org}}$ and $U_u$ overlap little, leaving the residual $\Pi_{\text{org}}\Pi_u^{\perp}(-\Delta\theta/\eta)$ small and noisy. In practice, optimization collapses to the generative prior or class cues, yielding label-consistent but semantically poor reconstructions (Figure 4). Symmetry moves thus decouple updates from data-dependent directions, removing the geometric handle exploited by white-box reconstruction. This motivates examining how teleportation reshapes the information relationship between parameters and training data (forget-set); a stronger symmetry-aware adaptive reconstruction attack is evaluated in Appendix N, and Appendix O provides complementary information-theoretic bounds showing how teleportation expands the symmetry orbit and increases expected reconstruction error.

## 5 CONCLUSION AND FUTURE WORK

Approximate unlearning provides scalability but introduces privacy risks. We showed that adversaries with access to original and unlearned models can mount strong membership inference and reconstruction attacks. These risks stem from two properties: parameter proximity and large forget-set gradient norms, which amplify leakage.

To counter this, we proposed WARP, a symmetry-based defense that interleaves teleportation with unlearning. By exploiting network symmetries, WARP reduces forget-set gradient energy and displaces parameters in symmetry-preserving directions, weakening both membership and reconstruction leakage while preserving retain performance. Across six unlearning algorithms, WARP improves privacy, cutting adversarial advantage by up to $64\%$ in black-box and $92\%$ in white-box settings. We also stress the need for white-box auditing: methods seemingly robust in black-box mode (e.g., SF Huang et al. (2024)) still leak when gradients are exposed. Even simple teleportation disrupts reconstruction, reducing quality by $\sim 45\%$.

Our findings suggest future directions. First, extending Langevin-based privacy analyses to practical unlearning with gradient ascent and symmetry moves is promising. Second, recent work shows approximate unlearning leaves low-rank weight signals, reversible via re-unlearning Fan et al. (2025) or removed by quantization Zhang et al. (2024). Exploring teleportation directly on weights may help obscure these signals and mitigate reversals. Finally, as the study of neural network symmetries continues to evolve and more efficient estimators and richer invariance families become available, WARP can directly inherit these advances by instantiating its symmetry map with stronger or cheaper symmetry mechanisms, which further strengthens its resistance to unlearning attacks.

ACKNOWLEDGMENTS

The research in this paper was supported by the UKRI Open Plus Fellowship (EP/W005271/1 Securing the Next Billion Consumer Devices on the Edge) and EU CHIST-ERA GNNs for Network Security and Privacy (GRAPHS4SEC) projects and by the Defense Advanced Research Projects Agency (DARPA), under contract W912CG23C0031.

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

## A    RELATED WORK

**Approximate Unlearning.**    The removal of training samples was introduced by Cao & Yang (2015) in the context of the "right to be forgotten." Retraining from scratch guarantees deletion but is infeasible for modern networks Vatter et al. (2023). Exact unlearning methods such as SISA Bourtoule et al. (2021) and Amnesiac Unlearning Graves et al. (2021) lower cost through partitioning or selective retraining but still require storage and scale poorly Nguyen et al. (2022).

Approximate unlearning directly updates the trained model to erase the *forget-set* Kurmanji et al. (2023); Chundawat et al. (2023a); Golatkar et al. (2020); Thudi et al. (2022). These methods aim to match the predictive distribution of retraining while preserving retain accuracy, offering a practical forgetting–utility trade-off with large savings in computation and memory. Related methods target structured forget sets such as entire classes Chundawat et al. (2023b); Seo et al. (2025), or tackle the harder instance-wise setting, where arbitrary samples must be removed Fan et al. (2024); Cha et al. (2024); Zhao et al. (2024). Many approaches rely on training-time side information like per-sample gradients Qiao et al. (2024); Mehta et al. (2022), or assume specialized regimes with adversarial robustness Liu et al. (2023) or differential-privacy noise Chien et al. (2024b;a); Sepahvand et al. (2025). While effective, these assumptions add resource overhead, limiting post-hoc use. Our focus, therefore, is training-agnostic, instance-wise unlearning that takes only a pretrained classifier and a designated *forget-set*, without stored gradients or training modifications Kurmanji et al. (2023); Thudi et al. (2022).

**Privacy Unlearning.**    The effectiveness of approximate unlearning is accessed by two criteria: (I) the model should maintain accuracy on non-forgotten data, and (II) its outputs on the *forget-set* should be indistinguishable from those of a model with no access to it Naderloui et al. (2025). In practice, this is evaluated using MIA Shokri et al. (2017); Carlini et al. (2022a), which test whether a sample was part of training. Effective unlearning removes this membership advantage on the *forget-set*.

Most prior work evaluates unlearning by comparing outputs of the unlearned model to a retrained reference on the *forget-set* Cadet et al. (2024); Kurmanji et al. (2023); Hayes et al. (2025); Georgiev et al. (2024); Naderloui et al. (2025). This black-box view ignores parameters, even though in practice—such as MU on edge devices—an adversary may access both original and unlearned models. Some studies consider this stronger setting: Chen et al. (2021) showed that output-comparison across models can detect unlearning, while others adapted reconstruction to infer forgotten data from parameter differencesSalem et al. (2020); Hu et al. (2024); Bertran et al. (2024). These works, however, are limited to toy models and simplified updates, leaving privacy risk under realistic conditions unclear. In particular, they do not capture the robustness of recent multi-step approximate methods such as NGP or SCRUB Kurmanji et al. (2023); Chundawat et al. (2023a), where iterative

updates with retain-set supervision weaken inversion of *forget-set* gradients. We address this gap with stronger white-box MIAs (Sec.C) and DRAs (Sec.3.1) tailored to realistic unlearning.

**Neural Network Symmetry.** Continuous symmetries in neural networks arise when transformations of the weights leave the output unchanged. Such invariances, a byproduct of overparameterization, mean that many distinct weight configurations represent the same function Głuch & Urbanke (2021). They appear in homogeneous activations Badrinarayanan et al. (2015); Du et al. (2018); Maheri et al. (2025b) and in components like softmax and batch normalization Kunin et al. (2020), and have been linked to both improved optimization and generalization. Neural teleportation leverages these symmetries by relocating parameters within the loss-invariant level set, yielding equivalent models that accelerate optimizationArmenta & Jodoin (2021); Armenta et al. (2023). Building on this idea,Zhao et al. (2022) introduced symmetry teleportation, which searches for beneficial relocations while providing a framework for analyzing symmetry-induced minima. More recently, teleportation with null-space gradient projection Wu et al. (2025) leverages the input null space: moving along projected directions leaves the function unchanged, directly aligning with the goal of teleportation.

## B  U-LiRA ALGORITHM

To evaluate sample-wise privacy leakage, we employ the U-LiRA attack Cadet et al. (2024); Hayes et al. (2025), an adaptation of LiRA Carlini et al. (2022a) to the unlearning setting. The attack relies on shadow models to estimate two distributions for a target sample $(x, y)$: (i) models trained with $(x, y)$ and subsequently unlearned using the same unlearning algorithm, and (ii) models trained from scratch without $(x, y)$. By fitting simple parametric models (e.g., Gaussians) to the outputs of these shadow ensembles, U-LiRA computes the likelihood of the target model's output under each case and classifies membership according to a likelihood ratio test.

Crucially, all shadow models are trained with the *same unlearning algorithm and hyperparameters* as the audited model. This makes U-LiRA effectively an *adaptive attack*, since it tailors the proxies to each specific unlearning method. Such alignment minimizes miscalibration between shadow and target models and is known to increase attack success Cretu et al. (2023). Therefore, U-LiRA serves as a strong black-box baseline for auditing privacy in unlearning. A complete description of the algorithm can demonstrated in Algorithm 1.

---

**Algorithm 1** U-LiRA (used for auditing unlearning)

---

**Require:** Target model $\theta^*$, learning algorithm $A$, unlearning algorithm $U$, number of shadows $T$, sample $(x, y)$
**Ensure:** Prediction: is $(x, y)$ in the *forget-set*?
 1: Initialize empty lists $O \leftarrow \{\}$ and $\hat{O} \leftarrow \{\}$
 2: **for** $t = 1$ to $T$ **do**
 3:     Sample dataset $D$ containing $(x, y)$
 4:     Train $\theta^0 \leftarrow A(D)$
 5:     Unlearn $\theta^f \leftarrow U(\theta^0, \{(x, y)\})$
 6:     Retrain $\theta^r \leftarrow A(D \setminus \{(x, y)\})$
 7:     Record $O[t] \leftarrow f(x; \theta^f)_y, \quad \hat{O}[t] \leftarrow f(x; \theta^r)_y$
 8: **end for**
 9: Fit Gaussian $(\mu, \sigma^2)$ to $O$, and $(\hat{\mu}, \hat{\sigma}^2)$ to $\hat{O}$
10: Compute $o^* \leftarrow f(x; \theta^*)_y$
11: Compute likelihood ratio:

$$p_{\text{member}} = \frac{\mathcal{N}(o^*; \mu, \sigma^2)}{\mathcal{N}(o^*; \mu, \sigma^2) + \mathcal{N}(o^*; \hat{\mu}, \hat{\sigma}^2)}$$

12: **if** $p_{\text{member}} > 0.5$ **then**
13:     **return** "member of training"
14: **else**
15:     **return** "non-member"
16: **end if**

---

## C White-box Gaussian Gradient–Difference Attack Algorithm

Guided by the GLiR framework of Leemann et al. (2023), we formulate sample-wise MIA in the unlearning setting as a binary hypothesis test that uses *both* the pre-unlearning and post-unlearning models. Let $A$ denote the training algorithm, $U$ the unlearning operator, $S$ the original training set, and $F \subseteq S$ the forget subset. For a candidate example $(x, y)$, we test

$$H_0 : (x, y) \sim \mathcal{D}_{\text{test}}, \quad (\theta^{\text{org}}, \theta^u) = \big(A(S), U(A(S), F)\big) \text{ with } x \notin S, \ x \notin F,$$
$$H_1 : (x, y) \in \mathcal{D}_{\text{forg}}, \quad (\theta^{\text{org}}, \theta^u) = \big(A(S), U(A(S), F)\big) \text{ with } x \in S \text{ and } x \in F,$$

i.e., under $H_1$ the point participated in the original training and was subsequently targeted by unlearning, whereas under $H_0$ it was never used. With white-box access, we form the gradient-difference statistic

$$\Delta(x) \ = \ \nabla_\theta \, \ell(f(x; \theta^u), y) \ - \ \nabla_\theta \, \ell(f(x; \theta^{\text{org}}), y) \in \mathbb{R}^d.$$

Assuming access to draws from $\mathcal{D}_{\text{test}}$, the adversary builds a background set $B = \{(b_i, \tilde{y}_i)\}_{i=1}^m \sim \mathcal{D}_{\text{test}}^m$ and estimates the null (non-member) distribution of gradient differences via

$$\hat{\mu} = \frac{1}{m} \sum_{i=1}^m \Delta(b_i), \qquad \hat{\Sigma} = \frac{1}{m-1} \sum_{i=1}^m \big(\Delta(b_i) - \hat{\mu}\big)\big(\Delta(b_i) - \hat{\mu}\big)^\top.$$

Following Leemann et al. (2023), we adopt a Gaussian model for $\Delta(x)$ under $H_0$ and compute the whitened Mahalanobis statistic

$$s(x) \ = \ \big(\Delta(x) - \hat{\mu}\big)^\top \big(\hat{\Sigma} + \lambda I\big)^{-1} \big(\Delta(x) - \hat{\mu}\big),$$

with a small ridge $\lambda > 0$ for numerical stability. Under $H_0$, $s(x)$ is approximately $\chi_d^2$-distributed, yielding the log-$p$-value score

$$A'(x, y) \ = \ -\log\Big(1 - F_{\chi_d^2}\big(s(x)\big)\Big),$$

and the final decision rule

$$A(x, y) \ = \ \mathbb{I}[A'(x, y) > \tau],$$

predicting *forgotten* when the score exceeds threshold $\tau$. Algorithm 2 provides the full details of the proposed attack.

**Relation to GLiR and unlearning specifics.** GLiR aggregates evidence across training steps by comparing per-step sample gradients to a Gaussian background of batch gradients; our adaptation replaces the (typically unavailable) per-step trajectory with the two-model contrast $\Delta(x)$. The geometry is unchanged: Evidence corresponds to the squared norm of the whitened difference, $\|(\hat{\Sigma} + \lambda I)^{-1/2} \Delta(x)\|_2^2$. Unlike standard MIAs that query a single model, the test exploits white-box access to $\theta^{\text{org}}$ and $\theta^u$ and targets the unlearning-specific alternative $H_1$ (membership in both $S$ and $F$), providing a simple and powerful auditor for residual leakage after unlearning.

---

**Algorithm 2** White-box Gaussian Gradient–Difference Attack for Unlearning Audit

---

**Require:** Pre-unlearning model $\theta^{\mathrm{org}}$, post-unlearning model $\theta^u$, candidate sample $(x, y)$, loss $\ell$, predictor $f(\cdot; \theta)$, background sampler $\mathcal{S}_{\mathrm{test}}(m)$ that returns $m$ i.i.d. draws from $\mathcal{D}_{\mathrm{test}}$
**Require:** Hyperparameters: background size $m$, repetitions $T$, ridge $\lambda > 0$, decision threshold $\tau$
 1: $S \leftarrow 0$                                                ▷ initialize cumulative evidence
 2: **for** $t = 1$ **to** $T$ **do**
 3:       $B_t = \{(b_i, \tilde{y}_i)\}_{i=1}^m \leftarrow \mathcal{S}_{\mathrm{test}}(m)$         ▷ if labels are unavailable, set $\tilde{y}_i = \arg\max f(b_i; \theta^{\mathrm{org}})$
 4:       **for** $i = 1$ **to** $m$ **do**
 5:            $\Delta_i \leftarrow \nabla_\theta \ell(f(b_i; \theta^u), \tilde{y}_i) - \nabla_\theta \ell(f(b_i; \theta^{\mathrm{org}}), \tilde{y}_i) \in \mathbb{R}^d$
 6:       **end for**
 7:       $\hat{\mu}_t \leftarrow \frac{1}{m} \sum_{i=1}^m \Delta_i$
 8:       $\hat{\Sigma}_t \leftarrow \frac{1}{m-1} \sum_{i=1}^m (\Delta_i - \hat{\mu}_t)(\Delta_i - \hat{\mu}_t)^\top$
 9:       $\hat{\Sigma}_{t,\lambda} \leftarrow \hat{\Sigma}_t + \lambda I_d$                             ▷ ridge for numerical stability
10:       $\Delta_x \leftarrow \nabla_\theta \ell(f(x; \theta^u), y) - \nabla_\theta \ell(f(x; \theta^{\mathrm{org}}), y)$
11:       $v \leftarrow \Delta_x - \hat{\mu}_t$
12:       Solve $\hat{\Sigma}_{t,\lambda} w = v$ for $w$ (e.g., Cholesky);     $s_t \leftarrow v^\top w$
13:       $\ell_t \leftarrow -\log\left(1 - F_{\chi_d^2}(s_t)\right)$                      ▷ log tail $p$-value under $H_0$
14:       $S \leftarrow S + \ell_t$
15: **end for**
16: **return** FORGOTTEN if $S > \tau$; else TEST

---

## D    ALTERNATIVE SYMMETRY: CHANGE-OF-BASIS NEURAL TELEPORTATION.

We also support the "neural teleportation" family of symmetry moves from Armenta et al. (2023). Let $\tau_a > 0$ be a scale attached to neuron $a$. For an edge $a \to b$ with weight $\theta_{ab}$ the teleported weight is

$$\theta'_{ab} = \frac{\tau_b}{\tau_a} \theta_{ab}, \tag{10}$$

and if $f_d$ is the activation at neuron $d$ then the teleported activation is

$$g_d(x) = \tau_d f_d\left(\frac{x}{\tau_d}\right), \tag{11}$$

which preserves the function for positively homogeneous activations and extends naturally to batch-norm scales Armenta et al. (2023). In a subset of experiments, we choose $\tau$ to further increase parameter dispersion under loss invariance (outputs unchanged), thereby weakening the differencing signal and making reconstruction harder; most results rely on the null space instantiation in equation 8. In the experimental section, it is explicitly indicated when both mechanisms are enabled.

## E    BASELINES

We evaluate our teleportation-based defense as a *plug-and-play* module layered on top of several state-of-the-art approximate post-hoc unlearning methods. These baselines are representative of the most widely studied approaches in recent literature, requiring no access to training-time auxiliary statistics (e.g., per-sample gradients) and operating directly on a pretrained model. Specifically, we consider:

1. **NegGrad+ (NGP)** Kurmanji et al. (2023): An improved variant of GA that incorporates a regularization term on the retain-set. The method balances ascent on the *forget-set* with descent on the retain-set, aiming to preserve model utility while unlearning.

2. **SCRUB** Kurmanji et al. (2023): A knowledge distillation approach that aligns the unlearned model with the original model on the retain-set via a consistency loss, while simultaneously removing the *forget-set*'s influence. SCRUB represents one of the most competitive baselines in recent evaluations.

3. **SalUn** Fan et al. (2023): A saliency-based unlearning method that directs updates to a subset of weights deemed *salient* for forgetting, identified via gradient-based weight saliency maps. By restricting optimization to these salient weights, SalUn enhances stability and efficiency compared to updating the full parameter set, and aims to reduce the gap to exact retraining.

4. **Projected Gradient Unlearning (PGU)** Hoang et al. (2024): A method that projects the gradient ascent update for the *forget-set* onto a subspace orthogonal to retain-set, thereby mitigating catastrophic forgetting. PGU is particularly relevant as it addresses gradient-level entanglement between forget and retain data.

5. **BadTeacher (BT)** Chundawat et al. (2023a): A recent distillation-based unlearning method where the unlearned model (student) is trained against a deliberately corrupted teacher that provides noisy or adversarial labels for the *forget-set*, encouraging the student to erase their influence while preserving performance on the retain-set.

6. **SRF-ON (SF)** Huang et al. (2024): A geometry-aware unlearning method that decomposes updates into forget ascent, retain descent, and saliency modulation. By embedding updates into the manifold of retain data and approximating Hessian modulation with a fast–slow strategy, SRF-ON improves stability–plasticity trade-offs and enables efficient large-scale unlearning.

These methods span the main paradigms of approximate unlearning—gradient ascent, retain-aware regularization, distillation, and projection-based updates—making them representative state-of-the-art baselines.

## F   ADDITIONAL WHITE-BOX RESULTS ON CIFAR-10

Figure 5 reports the complete ROC curves for the Gaussian gradient–diff test, covering the entire FPR range. These correspond to the same 640 unlearned models as in Figure 3, shown here without zoom to provide the full view.

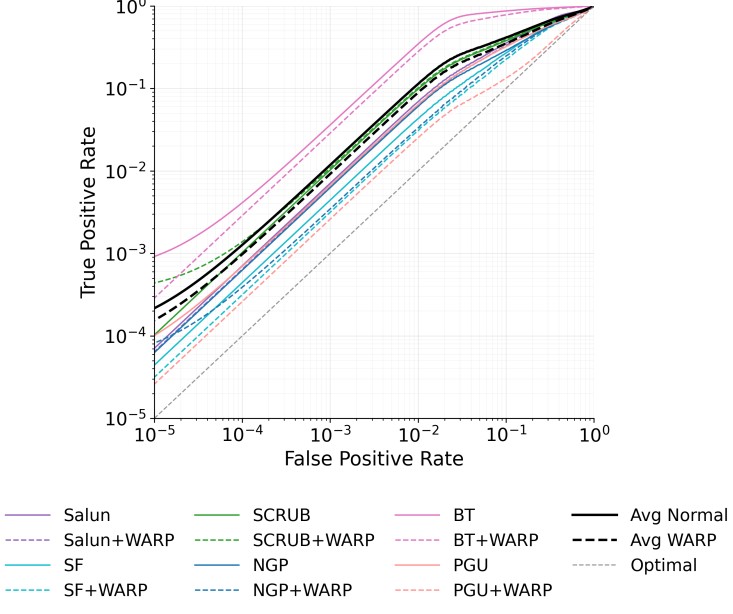

Figure 5: **Complete ROC curves for the white-box Gaussian gradient–diff test.** Averaged over 640 unlearned models, identical to Figure 3. Lower curves (closer to the random-guess diagonal) indicate stronger privacy.

Table 3: **Reconstruction on ImageNet-1K (ResNet-18), NGP (no defense).** Averages over 100 forgotten samples. Higher is better for PSNR/SSIM; lower is better for LPIPS/MSE.

| Method | PSNR (dB) ↑ | LPIPS (VGG) ↓ | LPIPS (Alex) ↓ | SSIM ↑ | Test MSE ↓ | Feat MSE ↓ |
|---|---|---|---|---|---|---|
| GIFD Fang et al. (2023) | $8.28 \pm 0.28$ | $0.630 \pm 0.012$ | $0.448 \pm 0.016$ | $0.098 \pm 0.007$ | $0.174 \pm 0.012$ | $6.725 \pm 0.506$ |
| **Ours** (subspace-filtered + GFID) | $\mathbf{10.74 \pm 0.31}$ | $\mathbf{0.564 \pm 0.013}$ | $\mathbf{0.345 \pm 0.015}$ | $\mathbf{0.117 \pm 0.008}$ | $\mathbf{0.100 \pm 0.007}$ | $\mathbf{5.388 \pm 0.497}$ |
| **Improvement (%)** | +29.7 | +10.5 | +22.9 | +19.4 | +42.5 | +19.9 |

Table 4: **White-box membership inference risk with and without teleportation (ViT, Tiny-ImageNet).** Results are reported as mean $\pm$ standard deviation across five splits. Improvements are computed as advantage reduction over random guessing.

| Method | AUC | TPR@0.01% | TPR@0.1% | TPR@1% | TPR@5% |
|---|---|---|---|---|---|
| NGP (base) | $0.792 \pm 0.019$ | $0.0019 \pm 0.001$ | $0.0188 \pm 0.009$ | $0.178 \pm 0.072$ | $0.444 \pm 0.035$ |
| + WARP | $\mathbf{0.755 \pm 0.019}$ | $\mathbf{0.0008 \pm 0.000}$ | $\mathbf{0.0079 \pm 0.003}$ | $\mathbf{0.075 \pm 0.027}$ | $\mathbf{0.302 \pm 0.054}$ |
| Improvement (%) | **12.7** | **61.1** | **61.2** | **61.2** | **36.1** |

## G   RECONSTRUCTION ATTACK BASELINES AND COMPARISON.

Table 3 compares three strategies for unlearning: (i) *simple differencing*, directly inverting $\Delta\theta$Hu et al. (2024); Bertran et al. (2024); (ii) *generative inversion* (GIFD)Fang et al. (2023) applied to $\Delta\theta$; and (iii) *Ours*, which adds *orthogonal subspace filtering* (Eq. equation 3) to a generative backbone. Results average 100 forgotten samples on ImageNet-1K with ResNet-18 under NGP unlearning.

## H   ADDITIONAL RESULTS: VIT ON TINY-IMAGENET

To extend the white-box analysis of Section 4.3, we evaluate Vision Transformer models trained on Tiny-ImageNet. We adopt ViT-B/16 as the base architecture and follow the same setup described in Section 4, with the *forget-set* constructed by randomly sampling $1\%$ of the training data and the retain-set consisting of the remainder. All models are trained with SGD and standard augmentations for ViT training. Unlearning is applied with NGP (NGP) and its teleported variant (NGP+WARP).

As shown in Table 4 and Figure 6, WARP substantially reduces attack success across all thresholds, with the largest relative gains at low false-positive rates where practical attacks operate. These results confirm that the symmetry-based defense proposed in WARP extends effectively to transformer models, demonstrating applicability beyond convolutional architectures.

## I   PRIVACY-UTILITY TRADE-OFF

Improving privacy in unlearning often comes at the cost of reduced model accuracy. Since test accuracy on the retain-set is one of the primary criteria for evaluating unlearning algorithms, it is critical to examine whether the proposed defense introduces unfavorable trade-offs.

We focus this analysis on NGP, as Figure 2 indicates that teleportation applied to NGP yields the most noticeable accuracy drop (roughly one percentage point), whereas for other methods accuracy remains stable or even improves. To probe this trade-off more carefully, we follow the hyperparameter tuning procedure described in Section 4 and select the top 20 trials with the highest validation score. From this pool we examine: (i) the single best-performing trial reported in Figure 2, (ii) the two trials with the highest validation accuracy, and (iii) the two trials with the lowest validation attack AUC.

Figure 7 plots test accuracy against privacy ($1-$AUC of black-box MIA, higher is better) for NGP and NGP+WARP across the selected hyperparameter trials. The overall trade-off is clear: higher accuracy typically coincides with lower privacy. Yet teleportation consistently shifts the Pareto frontier upward, delivering strictly better privacy at nearly every accuracy level. While NGP saturates around privacy $\approx 0.455$, teleportation extends this frontier up to $0.484$, breaking through the baseline ceiling. At the highest-accuracy setting, teleportation still provides a $\sim18\%$ reduction in attack advantage over random, demonstrating that even at stringent accuracy targets the defense yields nontrivial privacy

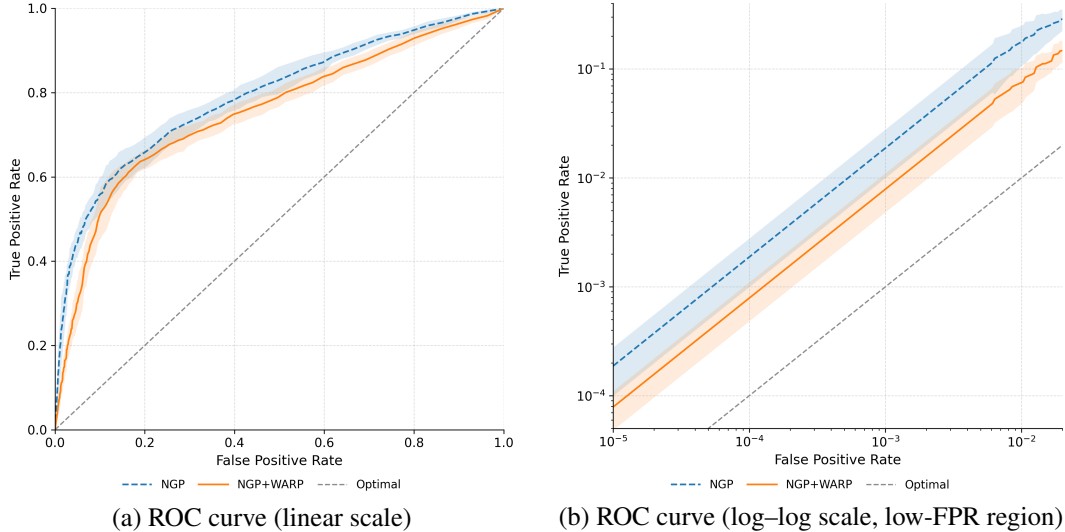

(a) ROC curve (linear scale)  (b) ROC curve (log–log scale, low-FPR region)

Figure 6: **White-box ROC for the Gaussian gradient–difference test on ViT-B/16 (Tiny-ImageNet).** Each curve is averaged over five different forget-set splits, with shaded regions showing the standard deviation. Both figures compare NGP and NGP+WARP; (a) presents the full ROC on a linear axis, while (b) zooms into the low-FPR regime on log–log scale, which is the operational region for practical attacks.

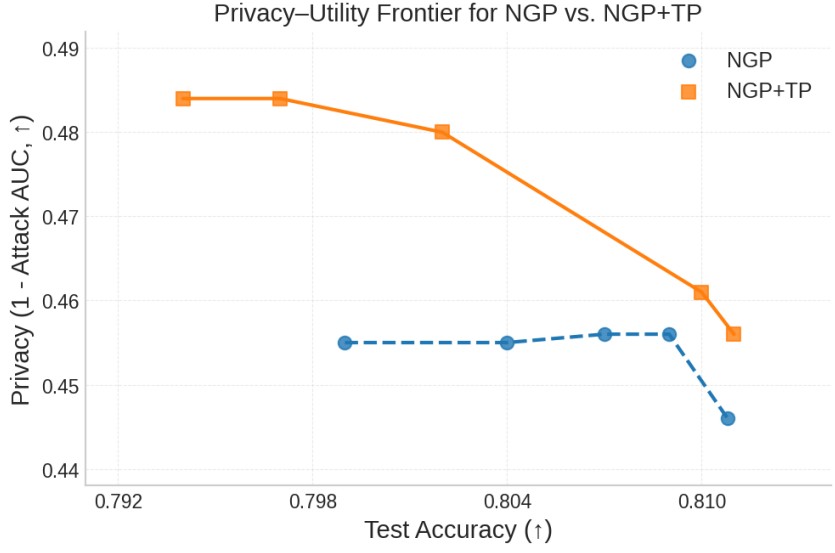

Figure 7: **Privacy–utility trade-off for NGP with and without WARP.** Each point is a hyperparameter trial, with privacy (1–AUC) averaged over $640$ shadow models ($64$ shadows $\times$ $10$ forget sets) under the U-LiRA protocol. Points further to the right (higher accuracy) and upward (higher privacy) indicate better trade-offs.

gains. Across the frontier, improvements remain stable, confirming that teleportation meaningfully reshapes the privacy–utility boundary in favor of the defender.

## J    RUNTIME ANALYSIS

In this appendix we focus on the retain–null-space instantiation of $T_\phi$, which is the only variant that requires explicit SVDs; the change-of-basis teleportation in Appendix D is SVD-free and without its

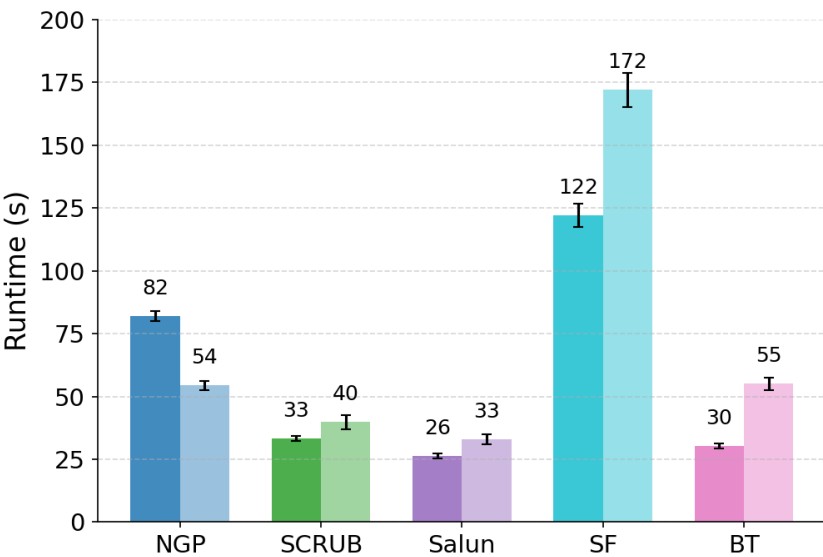

Figure 8: **Runtime overhead of teleportation.** Average runtimes (seconds) of unlearning algorithms with and without the WARP plugin, evaluated on CIFAR-10 with ResNet-18. Each bar reports the mean over five runs, with error bars showing standard deviations.

computational overhead as a result. Moreover, Section L introduces FastWARP, which replaces full SVD with randomized low-rank approximations and further reduces this overhead.

We benchmark the runtime of our teleportation defense across unlearning algorithms on a machine equipped with an NVIDIA GeForce RTX 4090 GPU (24 GB memory) and an Intel 13th Gen Core i9-13900KF CPU (24 cores, 32 threads, base 3.0 GHz, boost up to 5.8 GHz). Each experiment was repeated five times, and Figure 8 reports averages with standard deviations in the caption. All algorithms were run with the hyperparameters used in Table 1 and Figure 3, ensuring runtime reflects the same conditions as our privacy–utility evaluations.

For this particular SVD-based instantiation, teleportation increases runtime by approximately $+27\%$ relative to the baseline on average, reflecting the overhead of constructing the retain subspace. The main exception is NGP, where teleportation reduces runtime by about $-32\%$, due to more stable updates that in turn lower the required number of unlearning epochs. Since subspace computation can be pre-computed offline and does not need to be repeated after every teleportation step, this overhead can be amortized in practice. While updating the retain subspace less frequently can reduce cost, the primary computational overhead from full SVD is addressed directly by an approximate low-rank implementation (Appendix L), which removes the per-step bottleneck entirely.

## K  TELEPORTATION-BASED UNLEARNING ALGORITHM

In Algorithm 3, $T_\phi$ denotes an abstract symmetry operator; in our experiments we instantiate it either with retain–null-space teleportation or with change-of-basis teleportation, but any other loss-preserving symmetry could be used in its place.

---

**Algorithm 3** WARP (retain–null-space instantiation): teleportation-augmented gradient-based unlearning.

---

**Require:** $\theta^{\text{org}}, \mathcal{D}_{\text{f}}, \mathcal{D}_{\text{r}}, \ell_{\text{f}}, \ell_{\text{r}}, \lambda, \beta, \{\eta_t\}, \eta_{\text{tel}}, k, S$ or $\tau_{\text{grad}}, \sigma^2, \varepsilon, T$

1: $\theta_0 \leftarrow \theta^{\text{org}}$
2: **for** $t = 0, \ldots, T-1$ **do**
3:     sample $\mathcal{B}_{\text{f}} \subset \mathcal{D}_{\text{f}}, \ \mathcal{B}_{\text{r}} \subset \mathcal{D}_{\text{r}}$
4:     $\theta_{t+\frac{1}{2}} \leftarrow \theta_t - \eta_t\big(\nabla_\theta \ell_{\text{f}}(\theta_t \mid \mathcal{B}_{\text{f}}) + \lambda \nabla_\theta \ell_{\text{r}}(\theta_t \mid \mathcal{B}_{\text{r}})\big)$
5:     **if** $(t \bmod S = 0) \ \vee \ \|\nabla_\theta \ell_{\text{f}}(\theta_{t+\frac{1}{2}} \mid \mathcal{B}_{\text{f}})\|_2 > \tau_{\text{grad}}$ **then**
6:         **for** layer $\ell$ **do**
7:             build $R_\ell(\mathcal{B}_{\text{r}}); \quad R_\ell = U_\ell \Sigma_\ell V_\ell^\top$ (SVD)
8:             $B_\ell \leftarrow U_{\ell,1:k}; \quad \Pi_\ell^\perp \leftarrow I - B_\ell B_\ell^\top$
9:         **end for**
10:        $\mathcal{L}_{\text{tel}}(\theta) = \frac{1}{2}\sum_{(x,y)\in\mathcal{B}_{\text{f}}}\|\nabla_\theta \ell(f(x;\theta), y)\|_2^2 - \frac{\beta}{2}\|\theta - \theta^{\text{org}}\|_2^2$
11:        **for** layer $\ell$ **do**
12:            $W_\ell^{t+1} \leftarrow W_\ell^{t+\frac{1}{2}} - \eta_{\text{tel}}\, \Pi_\ell^\perp \big(\nabla_{W_\ell}\mathcal{L}_{\text{tel}}(\theta_{t+\frac{1}{2}})\big)$
13:        **end for**
14:        $\theta_{t+1} \leftarrow \{W_\ell^{t+1}\}_\ell$
15:        **if** $\ell_{\text{r}}(\theta_{t+1} \mid \mathcal{B}_{\text{r}}) > \ell_{\text{r}}(\theta_t \mid \mathcal{B}_{\text{r}}) + \varepsilon$ **then**
16:            $\theta_{t+1} \leftarrow \theta_{t+\frac{1}{2}}$                                          ▷ backtrack/safeguard
17:        **end if**
18:    **else**
19:        $\theta_{t+1} \leftarrow \theta_{t+\frac{1}{2}}$
20:    **end if**
21: **end for**
22: **return** $\theta^u \leftarrow \theta_T$

---

## L   APPROXIMATE NULL-SPACE TELEPORTATION

**Low-rank structure of retain representations.** For a retain minibatch $\mathcal{B}_{\text{r}}$ and layer $\ell$, let $R_\ell(\mathcal{D}_{\text{r}}) \in \mathbb{R}^{|\mathcal{B}_{\text{r}}| \times d_\ell}$ denote the matrix whose rows collect the layer-$\ell$ inputs $\{\phi_\ell(x)\}_{x\in\mathcal{B}_{\text{r}}}$. Empirically, $R_\ell(\mathcal{D}_{\text{r}})$ exhibits strong spectral decay: its spectrum is dominated by a small number of singular values, and most of the energy lies in a low-dimensional subspace. Such low-rank structure of activations, gradients and Hessians has been observed repeatedly in modern deep networks (Arora et al., 2019; Ghorbani et al., 2019; Fort et al., 2020; Gur-Ari et al., 2018), and is often attributed to overparameterisation and the implicit regularisation of SGD. In WARP, the retain subspace at layer $\ell$ is defined by the top-$k$ left singular vectors of $R_\ell(\mathcal{D}_{\text{r}})$:

$$R_\ell(\mathcal{D}_{\text{r}}) = U_\ell \Sigma_\ell V_\ell^\top, \qquad B_\ell = U_{\ell,1:k}, \qquad \Pi_\ell^\perp = I - B_\ell B_\ell^\top.$$

Since only the span of these dominant directions matters for teleportation, *exact* SVD is not required: any procedure that recovers a good approximation to the top-$k$ principal subspace suffices.

**Covariance-based PCA and subspace iteration.** Instead of computing a full thin SVD of $R_\ell(\mathcal{D}_{\text{r}})$, FASTWARP estimates $B_\ell$ via a covariance eigen-decomposition and a small number of subspace-iteration updates, following classical PCA and online PCA methods (Golub & Van Loan, 2013; Oja, 1982; Warmuth & Kuzmin, 2008; Mitliagkas et al., 2013). We first form the covariance

$$C_\ell \ = \ X_\ell X_\ell^\top \in \mathbb{R}^{d_\ell \times d_\ell},$$

where $X_\ell \in \mathbb{R}^{d_\ell \times N}$ is a layer-wise input matrix constructed from $\mathcal{B}_{\text{r}}$ (for convolutional layers we use unfolded patches; for batch-norm we aggregate per-channel features). We then compute the eigen-decomposition $C_\ell = Q_\ell \Lambda_\ell Q_\ell^\top$ and retain the smallest $k$ such that the cumulative explained

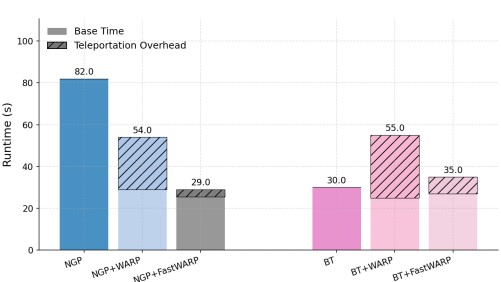

Figure 9: Runtime of the WARP plug-in on CIFAR-10 with ResNet-18. Each bar reports the mean over five runs. The top hatched segments correspond to the additional teleportation time; the solid base is the runtime of the underlying MU algorithm.

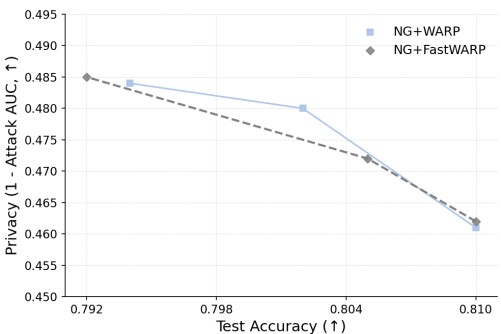

Figure 10: Privacy–utility comparison of NG+WARP and NG+FASTWARP. The approximate teleportation method (FASTWARP) matches the privacy–utility frontier of the exact variant, achieving nearly identical privacy and test accuracy.

variance exceeds a threshold $\tau$:

$$k = \min\left\{j : \frac{\sum_{i=1}^{j} \max(\lambda_{\ell,i},0)}{\sum_{i=1}^{d_\ell} \max(\lambda_{\ell,i},0)} \geq \tau\right\}, \qquad B_\ell = Q_{\ell,1:k},$$

optionally capped by a user-specified $k_{\max}$. For subsequent teleportation steps, we update $B_\ell$ using a few iterations of subspace iteration (Golub & Van Loan, 2013; Halko et al., 2011; Musco & Musco, 2015; Tropp et al., 2017; Woodruff, 2014):

$$Y \leftarrow C_\ell B_\ell, \qquad [B_\ell, \_] \leftarrow \mathrm{qr}(Y),$$

which amounts to an Oja-style streaming PCA update (Oja, 1982) with QR re-orthogonalisation. This reduces the cost of updating $B_\ell$ for a new minibatch from the $\mathcal{O}(|\mathcal{B}_\mathrm{r}|d_\ell^2)$ cost of a fresh thin SVD to $\mathcal{O}(|\mathcal{B}_\mathrm{r}|d_\ell k)$ for the covariance application plus $\mathcal{O}(d_\ell k^2)$ for QR, with $k \ll d_\ell$. The resulting projector $\Pi_\ell^\perp = I - B_\ell B_\ell^\top$ is then used exactly as in the original WARP update.

---

**Algorithm 4** FASTWARP basis update at layer $\ell$

**Require:** $d_\ell$, retain minibatch $\mathcal{B}_\mathrm{r}$, $B_\ell^{\mathrm{prev}}$ (or NONE), $\tau \in (0,1]$, $k_{\max}$, $T_{\mathrm{track}}$
 1: build $X_\ell \in \mathbb{R}^{d_\ell \times N}$ from $\mathcal{B}_\mathrm{r}$
 2: $C_\ell \leftarrow X_\ell X_\ell^\top$;   $C_\ell \leftarrow \frac{1}{2}(C_\ell + C_\ell^\top)$
 3: **if** $B_\ell^{\mathrm{prev}} = $ NONE **then**
 4:     $C_\ell = Q_\ell \Lambda_\ell Q_\ell^\top$
 5:     sort $\Lambda_\ell$ in descending order, permute $Q_\ell$ accordingly
 6:     $k \leftarrow \min\left\{k_{\max}, \min\left\{k : \frac{\sum_{i=1}^{k}\Lambda_{\ell,ii}}{\sum_i \Lambda_{\ell,ii}} \geq \tau\right\}\right\}$
 7:     $B_\ell \leftarrow Q_\ell[:, 1:k]$
 8: **else**
 9:     $B_\ell \leftarrow B_\ell^{\mathrm{prev}}$
10:     **for** $t = 1, \ldots, T_{\mathrm{track}}$ **do**
11:         $Y \leftarrow C_\ell B_\ell$
12:         $[B_\ell, \_] \leftarrow \mathrm{qr}(Y)$
13:         $B_\ell \leftarrow B_\ell[:, 1:k]$
14:     **end for**
15: **end if**
16: $\Pi_\ell^\perp \leftarrow I_{d_\ell} - B_\ell B_\ell^\top$
17: **return** $B_\ell$, $\Pi_\ell^\perp$

---

**Runtime and privacy–utility impact.** Figure 9 reports the runtime for NG and BT with and without teleportation on CIFAR-10/ResNet-18. The hatched segments correspond to the teleportation

component. Using full SVD yields a moderate yet visible overhead (e.g., BT+WARP increases runtime from 30 s to 55 s). Replacing full SVD with the covariance-based PCA and subspace iteration of Algorithm 4 (FASTWARP) shrinks this overhead substantially: total runtime drops to 29 s and 35 s for NG+FastWARP and BT+FastWARP, corresponding to a $2\times$–$3\times$ reduction in the teleportation time. The teleportation component becomes only a small fraction of the overall MU cost.

To measure the effect of this approximation on privacy and accuracy, Figure 10 compares NG+WARP and NG+FASTWARP along the privacy–utility frontier. The two curves are nearly indistinguishable: privacy $(1 - \text{AUC})$ differs by at most $0.3$–$0.6\%$ across operating points, and test accuracy changes by at most $0.2$–$0.3$ percentage points. We also track retain-set loss during teleportation and observe that the relative drift under FASTWARP remains below $2\%$, indicating that the approximate projector continues to enforce practical loss invariance. In some configurations, the additional numerical noise introduced by the approximation yields slightly *higher* privacy for the same utility. Overall, these results show that the privacy gains of WARP are robust to approximate PCA, and that FASTWARP preserves the empirical privacy–utility trade-off while significantly reducing computational overhead.

**Scalability to LLMs and calibration of the retain subspace.** A natural concern is whether null-space teleportation remains practical and stable at LLM scale, where layer widths reach $d_\ell \sim 10^3$–$10^4$ and a single minibatch may not span the retain subspace. Empirically, recent compression work shows that truncated SVD and related low-rank factorizations are already applied efficiently to full LLM weight matrices with comparable or larger dimensions: SVD-LLM Wang et al. (2024; 2025b) optimizes singular-value truncation for LLaMA Touvron et al. (2023)- and GPT Brown et al. (2020)-class models while preserving perplexity and throughput, demonstrating that rank-$k$ SVD with $k \ll d_\ell$ is tractable in practice on modern hardware. Complementary methods such as ResSVD Bai et al. (2025) leverage the residual matrix left by truncation to correct the approximation, further reducing the effective loss of expressivity at fixed rank. Orthogonal lines of work, e.g., weighted low-rank factorization for LMs, explicitly introduce data-dependent weights in the covariance (or Gram) operator to bias the recovered subspace toward high-importance tokens or examples, and report competitive compression ratios on transformer-based LMs Hsu et al. (2022); Sakr & Khailany (2024). In our setting, we can adopt the same design principles: instead of forming $R_\ell(\mathcal{B}_r)$ from an arbitrary minibatch, we maintain a small buffer of retain batches with large gradient norm Sakr & Khailany (2024) or Fisher information, and construct the activation matrix $X\ell$ from this "high-influence" pool. This yields a weighted or importance-sampled covariance $C_\ell = X_\ell X_\ell^\top$ whose top-$k$ eigenspace more faithfully captures the retain subspace seen over the full retain stream, while keeping the per-teleportation cost at $\mathcal{O}(|\mathcal{B}_r|d_\ell k)$. Combined with low-rank SVD implementations that are already optimized for LLM compression, these heuristics make the FastWARP projector construction compatible with large transformer architectures without breaking the retain loss invariance enforced by WARP. We leave the adaptation to large language models for future research. Our contributions target symmetry-based defenses for generic neural networks and established MU baselines, and do not address LLM-specific challenges in unlearning, which constitute a distinct line of investigation.

# M    COMPARISON WITH DP–LANGEVIN NOISE DEFENCES

While our goal is to make neural networks more resilient to privacy attacks *post hoc*, a natural question is how WARP compares with defences based on differential privacy (DP). DP is the strongest known framework for providing indistinguishability guarantees between neighbouring datasets, and a small number of recent unlearning methods have attempted to translate these guarantees into *certified* machine unlearning. Among these, noisy-gradient (Langevin) approaches provide the closest analogue to our setting; we therefore include them as a comparison point.

Certified unlearning methods such as Guo et al. (2020); Chien et al. (2024b) formalise unlearning as an indistinguishability requirement between (i) a model obtained by training on the full dataset, and (ii) a counterfactual model that has never seen the forget set. These works build on the principle that if the training algorithm is itself DP, then suitable post-processing can yield certified removal of training points. Such guarantees make DP–Langevin the strongest known *general-purpose* defence with explicit indistinguishability guarantees, hence a meaningful baseline to evaluate privacy–utility trade-offs.

Table 5: **NGP+WARP vs. Langevin noise (U-LiRA, black-box).** Reported are risks on *all forget samples* and on the *most–memorized* subset (top 5%), plus test accuracy. U-LiRA AUC and TPR@0.1% (FPR) are shown for each setting.

| Method | All samples (BB) | | Most–memorized (top 5%) | | Acc. |
| --- | --- | --- | --- | --- | --- |
| | AUC | TPR@0.1 | AUC | TPR@0.1 | Test |
| Langevin ($\varepsilon = 1$) | 0.523 | 0.004 | 0.671 | 0.029 | 0.682 |
| Langevin ($\varepsilon = 4$) | 0.571 | 0.006 | 0.766 | 0.048 | 0.718 |
| Langevin ($\varepsilon = 8$) | 0.627 | 0.020 | 0.912 | 0.166 | 0.771 |
| Langevin ($\varepsilon = 16$) | 0.650 | 0.027 | **0**.935 | 0.224 | **0.798** |
| NGP + WARP | **0.516** | **0.003** | **0.598** | **0.015** | 0.797 |

**What the DP guarantees actually require.** The formal guarantees in Guo et al. (2020); Chien et al. (2024b) rely on assumptions that do *not* hold in the deep, non-convex MU regime we consider:

1. **Convexity and strong dissipativity.** Both works require (strongly) convex, $\ell_2$–regularised objectives to bound the stationary distribution of the noisy dynamics. Deep convolutional networks trained with cross-entropy fundamentally violate these assumptions.

2. **DP-trained initial model required.** The certified-unlearning guarantee requires that the *original* model be obtained using *the same* noisy-gradient mechanism (noisy SGD or Langevin) applied throughout training on the full dataset. This is explicitly stated as a necessary condition in Chien et al. (2024b). In contrast, our setting begins from a standard ERM-trained model, which is non-DP and therefore outside the scope of their certification theorem.

As a result, the "$\varepsilon$" obtained from the RDP accountant in our experiments should be interpreted purely as a calibrated *noise level*, not as a valid DP guarantee. Our use of Langevin noise is therefore a *strong noise-based defence*, not a certified mechanism.

**Adapting projected Langevin unlearning to MU.** Following Chien et al. (2024b), we implement projected Langevin dynamics on top of the same MU objective used throughout the paper. For a per-sample clipped gradient with radius $C$ and loss

$$\mathcal{L}_{\mathrm{MU}}(\theta) = \alpha \left( \ell_{\mathrm{r}}(\theta) + \lambda \|\theta - \theta_p\|_2^2 \right) - (1 - \alpha)\,\ell_{\mathrm{f}}(\theta),$$

the DP–Langevin update is

$$g_t = \mathrm{clip}\bigl(\nabla_\theta \mathcal{L}_{\mathrm{MU}}(\theta_t),\, C\bigr), \tag{12}$$

$$\theta_{t+1} = \theta_t - \eta_t g_t + \sqrt{2\,\eta_t\,\lambda}\,\boldsymbol{\xi}_t, \qquad \boldsymbol{\xi}_t \sim \mathcal{N}(0, I), \tag{13}$$

where $\lambda$ is the regularisation parameter entering the RDP privacy analysis. Given a target privacy level $\varepsilon$, we follow the exact Rényi-DP accounting of Chien et al. (2024b) to compute the Gaussian noise standard deviation $\sigma$ required by their Langevin update. In our implementation, three quantities act as tunable hyperparameters: the learning rate $\eta$, the per-sample gradient-clipping radius $C$, and the regularisation coefficient $\lambda$ that appears in the RDP analysis. For any chosen $(\eta, C, \lambda)$ and target $\varepsilon$, the formulas of Chien et al. (2024b) uniquely determine the corresponding noise scale $\sigma$. To ensure fairness across baselines, we run the same number of hyperparameter-search trials as for the MU baselines, jointly sweeping $(\eta, C, \lambda)$ to obtain the set of reported results in Table 5.

**Interpretation under non-convexity.** Although the privacy accountant yields a numerical $\varepsilon$, none of the formal conditions needed for DP-certified unlearning hold for our deep ResNet models. Consequently, we reiterate that the resulting values should not be interpreted as DP guarantees but rather as a systematic way of calibrating the magnitude of injected noise. The comparison therefore isolates the *empirical* effect of noise injection on forgetting, retention, and attack success.

**Empirical privacy–utility trade-off.** Table 5 reveals a clear tension between nominal DP guarantees and empirical membership privacy. As the target privacy budget for Langevin is relaxed

from $\varepsilon = 1$ to $\varepsilon = 16$, test accuracy gradually recovers (from 0.682 up to 0.798), but U-LiRA risk monotonically *increases*: the all-sample AUC rises from 0.523 to 0.650, and the AUC on the top-5% most memorised points grows from 0.671 to 0.935, with TPR@0.1% FPR increasing from 0.029 to 0.224. In contrast, NGP+WARP simultaneously achieves competitive utility and strictly lower attack success: on all forget samples it attains the best AUC and TPR@0.1% (0.516 and 0.003), and on the most–memorised subset it reduces AUC to 0.598 and TPR@0.1% to 0.015, outperforming every Langevin configuration by a wide margin. Notably, relative to the lowest-noise setting ($\varepsilon = 16$), NGP+WARP matches accuracy (0.797 vs. 0.798) while cutting the memorised AUC from 0.935 to 0.598 and TPR@0.1% from 0.224 to 0.015. For stronger nominal privacy ($\varepsilon = 1$ or 4), Langevin noise severely degrades accuracy (down to 0.682) yet still leaves substantially higher attack AUC and TPR than WARP. Overall, these results suggest that isotropic DP noise is poorly aligned with the specific memorization patterns exploited by U-LiRA: it injects substantial randomness into all updates, harming utility without reliably protecting the most vulnerable examples, whereas WARP reshapes the parameter space in a targeted way that yields a markedly better empirical privacy–utility frontier.

Taken together, these observations clarify the roles of the two approaches. Langevin noise offers a principled mechanism for *certified* unlearning in the restricted setting of convex, DP-trained models, but its guarantees do not extend to the non-convex MU regime nor to pretrained models obtained without DP noise. Consequently, applying Langevin updates post hoc to deep networks provides no formal protection and yields an unfavourable privacy–utility trade-off in practice. By contrast, WARP operates directly on arbitrary pretrained models, targets the directions most responsible for memorization, and empirically achieves substantially stronger resistance to membership inference at comparable accuracy. A compelling direction for future work is to investigate whether the geometric structure exploited by WARP can be combined with, or serve as a foundation for, certified unlearning mechanisms that simultaneously handle non-convex objectives and non-DP initialisation—a capability not supported by current DP-Langevin frameworks.

## N    ADAPTIVE RECONSTRUCTION WITH SYMMETRY–AWARE ATTACKER

Teleportation acts by composing the unlearning update with a symmetry transform that preserves predictions but redistributes parameter mass along loss–invariant directions (Section 3.2). This raises a natural question: can a stronger white-box adversary, aware of the teleportation family, *invert* or compensate for these symmetry moves and recover the residual forget gradient? More concretely, if the attacker can parameterise and optimize over the change-of-basis (COB) scales $\tau$ used in neural teleportation (Armenta et al., 2023), does this restore reconstruction quality and defeat WARP?

It is worth noting that our privacy evaluation already includes two adaptive–attack families: U-LiRA and GLiR, both of which instantiate adaptive membership-inference attacks by optimising proxy models or surrogate loss landscapes. However, the reconstruction attack considered in Section 3.1—which directly targets instance-level recovery of the forgotten data—was *not* adaptive: the attacker optimized only over the dummy image while keeping the teleportation parameters fixed. To fully test the robustness of symmetry-based teleportation, we now consider a strictly stronger attacker that *jointly* optimizes both the dummy image and the teleportation parameters themselves.

Concretely, we study whether an attacker who can parameterise and optimize over the change-of-basis (COB) symmetry scales $\tau$ used in neural teleportation (Armenta et al., 2023) can undo the defender's symmetry moves, thereby restoring the clean gradient geometry required for successful reconstruction. This experiment directly probes whether teleportation is merely hiding the forget gradient behind a reversible reparameterisation, or whether it fundamentally reshapes the inverse problem faced by reconstruction attacks.

**Attack formulation.**    In the adaptive setting, we give the attacker full knowledge of the teleportation family and let them *shadow* the defender's operations. Specifically, starting from the original pretrained weights $\theta_{\mathrm{org}}$, the attacker first applies a change-of-basis symmetry parametrised by COB scales $\tau = \{\tau_a > 0\}$, obtaining

$$\theta_{\mathrm{org}}^{(\tau)} \;=\; T_\tau(\theta_{\mathrm{org}}), \tag{14}$$

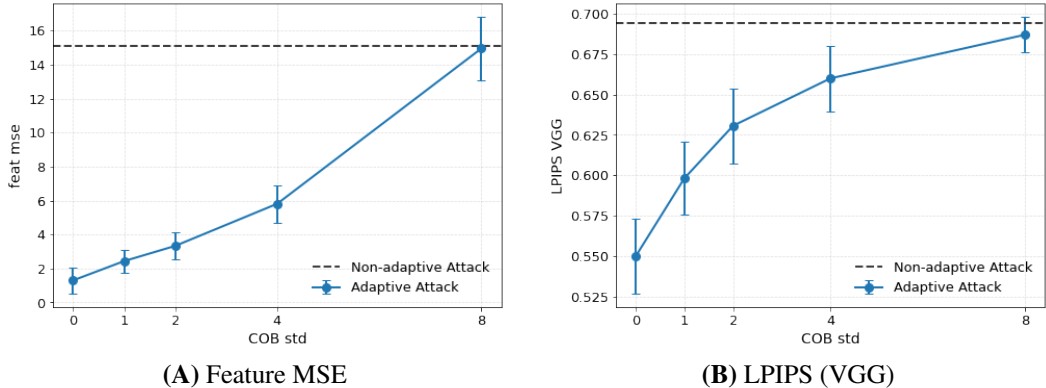

**(A)** Feature MSE  **(B)** LPIPS (VGG)

Figure 11: **Adaptive reconstruction under change-of-basis teleportation (NGP, ImageNet-1K).** (A) Feature MSE and (B) LPIPS (VGG) as a function of the COB standard deviation $\sigma_{\mathrm{cob}}$. Increasing the symmetry variance consistently worsens reconstruction quality across both metrics.

where $T_\tau$ is the COB teleportation map (Appendix D). They then perform a single gradient step in parameter space using a dummy image–label pair $(x, y)$:

$$\theta^{(\tau)}(x, y) \;=\; \theta_{\mathrm{org}}^{(\tau)} \;+\; \eta_{\mathrm{att}} \, \nabla_\theta \ell\big(f(x; \theta_{\mathrm{org}}^{(\tau)}), y\big), \tag{15}$$

with attack step size $\eta_{\mathrm{att}} > 0$. The attacker's goal is to choose $(x, \tau)$ so that the shadowed update in equation 15 closely matches the actual unlearned parameters $\theta_u$ produced by WARP. Formally, we solve

$$\hat{x}_f, \hat{\tau} \;\in\; \arg\min_{x, \tau} \Big[ D\big(\theta^{(\tau)}(x, y), \theta_u\big) + \lambda_{\mathrm{TV}} \, \mathrm{TV}(x) + \lambda_\tau \, \Omega(\tau) \Big], \tag{16}$$

where $D(\cdot, \cdot)$ is a parameter-space discrepancy (we use $\ell_2$ distance over all weights), $\mathrm{TV}(x)$ is the total-variation regulariser on the image, and $\Omega(\tau)$ implements a Gaussian prior $\tau_a \sim \mathcal{N}(1, \sigma_{\mathrm{cob}}^2)$ on each COB scale. We optimize equation 16 by alternating gradient steps on $x$ and $\tau$, with $\tau$ clipped to a bounded interval around 1 to avoid degenerate scalings.

**Experimental setup.** For a fair comparison, we reuse exactly the reconstruction protocol of Section 4.4 (same model, dataset, forgotten examples, optimizer, and image priors), and only extend the attack to optimize over the COB parameters $\tau$ via equation 16. We vary the COB prior variance $\sigma_{\mathrm{cob}}$ that defines $\Omega(\tau)$, treating each $\tau_a$ as a scalar random variable centred at 1 with variance $\sigma_{\mathrm{cob}}$. We sweep $\sigma_{\mathrm{cob}} \in 0, 0.1, 0.2, 0.4, 0.8$, where $\sigma_{\mathrm{cob}} = 0$ recovers the non-adaptive attack with fixed $\tau \equiv 1$, and larger values correspond to stronger dispersion along the symmetry orbit induced by WARP. Following the evaluation protocol of Table 2, we quantify reconstruction quality using PSNR, SSIM, LPIPS, and feature MSE, reporting averages over 30 randomly drawn forget examples.

**Results and connection to theory.** Figure 11 shows how reconstruction quality changes as we increase the COB prior std $\sigma_{\mathrm{cob}}$ that controls the spread of admissible symmetry scales. When $\sigma_{\mathrm{cob}} = 0$ the symmetry prior collapses around $\tau_a \approx 1$, so the attacker effectively searches over a narrow neighbourhood of the defender's true teleportation and can partially recover the forgotten signal: the adaptive attack achieves substantially lower feature MSE and LPIPS than the non–adaptive WARP attack (dashed line). However, the attacker never observes the ground-truth COB scales used by the defender; as $\sigma_{\mathrm{cob}}$ grows and the symmetry orbit broadens, the optimisation over $(x, \tau)$ quickly becomes unstable. Both metrics deteriorate almost monotonically with $\sigma_{\mathrm{cob}}$: already at moderate variance the gains over the non–adaptive attack largely disappear, and for the largest tested $\sigma_{\mathrm{cob}}$ the adaptive reconstructions are statistically indistinguishable from (or slightly worse than) the non–adaptive baseline. Importantly, the COB standard deviation is a defender-controlled knob: in this symmetry family we can raise $\sigma_{\mathrm{cob}}$ up to 1.0 without changing the realised network function, and in our main reconstruction experiments in Table 2 we set $\sigma_{\mathrm{cob}} = 0.8$, already placing the attacker in a high-variance regime where adaptive reconstruction is strongly impaired.

This trend is consistent with our theoretical analysis in Appendix O, which shows that the expected reconstruction error increases with the variance of the COB scales. Larger $\sigma_{\mathrm{cob}}$ expands the symmetry orbit of $\theta_{\mathrm{org}}$ and $\theta_u$, so the update $\Delta\theta$ admits many symmetry–equivalent decompositions whose gradients are nearly orthogonal to the true forget gradient $g_f$. The optimisation problem in equation 16 thus becomes a highly ill-posed inverse problem over the joint space $(x, \tau)$, where many different configurations of $(x, \tau)$ produce similar matches in parameter space. Empirically, the adaptive optimiser drifts toward such low–signal-to-noise solutions that satisfy the symmetry constraints but no longer encode the specific forgotten example, explaining the systematic degradation in reconstruction quality as symmetry variance (or std) increases.

**Takeaway.** Even under a strong white-box threat model—where the attacker knows the teleportation family and jointly adapts both the dummy input and the symmetry parameter—teleportation continues to disrupt reconstruction effectively. The injected symmetry components become entangled with the forget-induced update $\Delta\theta$, enlarging the attacker's search space and destroying the geometric alignment between parameter differences and the underlying forgotten example. Thus, teleportation does not merely reparameterise the model in a way that can be inverted; instead, by injecting symmetry variance into the update, it structurally increases reconstruction error and removes the clean gradient-based signal that standard reconstruction attacks depend on. This provides empirical and theoretical evidence that symmetry-based teleportation fundamentally hardens the inverse problem faced by adaptive adversaries.

## O  TELEPORTATION-AWARE INFORMATION-THEORETIC BOUNDS ON GRADIENT-BASED RECONSTRUCTION

### O.1  OVERVIEW OF THE THEORETICAL ANALYSIS

This appendix develops an information-theoretic lower bound on the minimal reconstruction mean-squared error (MSE) achievable by a gradient-based inversion adversary within a shared probabilistic model for gradients. We first adapt standard entropy–MSE relationships to the case where the attacker observes gradients rather than intermediate features, closely following the spirit of the analysis in Xia et al. Xia et al. (2025). We then introduce a Gaussian-mixture model (GMM) for gradient features and derive a parametric lower bound on the conditional entropy $H(x \mid g)$, analogous to the intermediate-feature analysis in Xia et al. (2025) but specialized to gradients. Finally, we incorporate teleportation (change-of-basis) noise as private randomness in the training dynamics and analyze its impact on the *same* lower-bound pipeline, under an explicit diagonal approximation and an energy-preserving design assumption on the change-of-basis distribution. Throughout, we keep the modelling assumptions identical between the teleported and non-teleported channels, so any improvement we prove directly reflects a genuine tightening of the analytic lower bound on reconstruction error—and hence a provable gain in information-theoretic privacy *within this common generative framework*. We emphasize that $H(x)$ is fixed by the dataset distribution, so only *relative* differences between the channels are meaningful.

### O.2  SETUP AND THREAT MODEL

**Data and model.** Let $x \in \mathbb{R}^d$ denote the $d$-dimensional input random variable, distributed according to some unknown data distribution on a measurable subset $\mathcal{X} \subseteq \mathbb{R}^d$. We assume throughout that $x$ admits a density w.r.t. Lebesgue measure and has finite second moment. (If one wishes to model discrete or manifold-supported data, the analysis can be recovered by adding an arbitrarily small Gaussian perturbation to $x$ as is standard in differential-entropy arguments; we implicitly assume such smoothing has been applied so that conditional covariances below are positive definite.)

Consider a deep network with parameters $W \in \mathbb{R}^{m \times d}$ and first-layer pre-activations

$$z = Wx \in \mathbb{R}^m,$$

and a subsequent decoder $F_d$. Let $\ell(\cdot, y)$ be a loss for a label $y$, and define the gradient with respect to $z$:

$$g_z = \nabla_z \ell(F_d(z), y) \in \mathbb{R}^m.$$

In the analysis below, the attacker's observation will be a gradient-based signal $g$ (not necessarily equal to $g_z$ directly) that is deterministically related to $(x, y, W)$ plus noise. In a white-box setting, for instance, the adversary can observe weight differences across steps, which are affine functions of the underlying gradient features; since mutual information and our entropy-based bounds are invariant under fixed invertible affine reparametrizations, it is without loss of generality to work with a canonical gradient feature $g$.

**Adversarial objective.** An inversion adversary aims to reconstruct $x$ from the observable $g$. Given an estimator $\hat{x}(g)$, we measure reconstruction quality by the mean-squared error (MSE)

$$\xi_g(\hat{x}) := \frac{1}{d}\, \mathbb{E}\big[\|x - \hat{x}(g)\|_2^2\big]. \tag{17}$$

The *minimal* MSE $\xi_g$ is the infimum of equation 17 over all measurable estimators $\hat{x}(\cdot)$. We interpret "information-theoretic robustness" as the regime where the attacker is Bayes-optimal under the assumed generative model, i.e. has access to the true posterior $P(x \mid g)$ induced by that model and implements the Minimum Mean Square Error (MMSE) estimator.

*Assumption* 1 (Basic regularity). We assume:

(i) $x$ has a density on $\mathbb{R}^d$ and finite second moment;
(ii) for the observation $g$, the conditional distribution $P(x \mid g)$ admits a density with finite second moment, and its covariance matrix $\mathrm{Cov}(x \mid g)$ is positive definite almost surely;
(iii) all entropies, mutual informations and expectations used below are finite.

These conditions are standard in information-theoretic MMSE analysis (see, e.g., Xia et al. (2025)) and ensure that all quantities are well-defined and that the maximum-entropy characterization for Gaussians can be applied without degeneracy.

### O.3    MINIMAL MSE FROM GRADIENTS AND AN ENTROPY-BASED LOWER BOUND

**Bayes-optimal reconstruction from gradients**    We first recall the standard MMSE characterization.

**Proposition 1** (Minimal reconstruction MSE from gradients). *Let $x \in \mathbb{R}^d$ and an observation $g$ satisfy Assumption 1. Consider estimators $\hat{x}(g)$ of $x$ based on $g$ and define $\xi_g(\hat{x})$ as in equation 17. Then:*

(i) *The estimator that minimizes $\xi_g(\hat{x})$ is the conditional mean $\hat{x}^\star(g) = \mathbb{E}[x \mid g]$.*
(ii) *The corresponding minimal MSE is*

$$\xi_g := \inf_{\hat{x}} \xi_g(\hat{x}) = \frac{1}{d}\, \mathbb{E}_g\Big[\mathrm{Tr}\big(\mathrm{Cov}(x \mid g)\big)\Big], \tag{18}$$

*where $\mathrm{Cov}(x \mid g)$ denotes the conditional covariance of $x$ given $g$ and $\mathbb{E}_g$ is expectation w.r.t. $g$.*

*Proof.* For any fixed $g$, the conditional risk $\mathbb{E}[\|x - \hat{x}(g)\|_2^2 \mid g]$ is uniquely minimized by $\hat{x}^\star(g) = \mathbb{E}[x \mid g]$ (standard MMSE theory, cf. Xia et al. (2025)). The minimal conditional risk at $g$ is

$$\mathbb{E}\big[\|x - \mathbb{E}[x \mid g]\|_2^2 \mid g\big] = \mathrm{Tr}\big(\mathrm{Cov}(x \mid g)\big),$$

since for any random vector $X$ with mean $\mu$ and covariance $\Sigma$ one has $\mathbb{E}\|X - \mu\|_2^2 = \mathrm{Tr}(\Sigma)$. Taking expectation over $g$ and dividing by $d$ yields equation 18. $\qquad\square$

Thus, when we refer to the "minimal MSE achievable by an attacker" for a given observation model, we mean $\xi_g$ as given in equation 18, corresponding to a Bayes-optimal adversary within that model.

**An entropy-based lower bound on the minimal MSE**    We now relate the minimal MSE $\xi_g$ to the conditional entropy $H(x \mid g)$, generalizing standard entropy–MMSE inequalities (cf. Xia et al. (2025)).

**Theorem 1** (Entropy-based lower bound on gradient reconstruction). *Under Assumption 1, let $H(x \mid g)$ be the conditional differential entropy of $x$ given the observation $g$. Then the minimal reconstruction MSE $\xi_g$ in equation 18 satisfies*

$$\xi_g \geq \frac{1}{2\pi e}\, \exp\!\Big(\frac{2}{d}\, H(x \mid g)\Big). \tag{19}$$

*Proof.* Fix $g$ and define $\Sigma(g) := \operatorname{Cov}(x \mid g)$. Under Assumption 1, $\Sigma(g)$ is symmetric and positive definite almost surely. For each such $g$, the conditional distribution of $x$ given $g$ has entropy bounded above by that of a Gaussian with the same covariance:

$$H(x \mid g = g) \ \leq \ \frac{1}{2} \log\big((2\pi e)^d \det(\Sigma(g))\big),$$

with equality iff $x \mid g$ is Gaussian. This is the usual maximum entropy property of Gaussians. Taking expectation over $g$ gives

$$H(x \mid g) \ = \ \mathbb{E}_g\big[H(x \mid g = g)\big] \ \leq \ \mathbb{E}_g\Big[\tfrac{1}{2} \log\big((2\pi e)^d \det(\Sigma(g))\big)\Big]. \tag{20}$$

Let $\lambda_1(g), \ldots, \lambda_d(g)$ be the eigenvalues of $\Sigma(g)$ (all positive). Then

$$\det(\Sigma(g)) = \prod_{j=1}^{d} \lambda_j(g), \quad \operatorname{Tr}(\Sigma(g)) = \sum_{j=1}^{d} \lambda_j(g).$$

By the Arithmetic Mean-Geometric Mean (AM–GM) inequality,

$$\prod_{j=1}^{d} \lambda_j(g) \ \leq \ \Big(\frac{1}{d} \sum_{j=1}^{d} \lambda_j(g)\Big)^d = \Big(\frac{\operatorname{Tr}(\Sigma(g))}{d}\Big)^d,$$

so

$$\log \det(\Sigma(g)) \ \leq \ d \log\Big(\frac{\operatorname{Tr}(\Sigma(g))}{d}\Big).$$

Substituting into equation 20,

$$H(x \mid g) \ \leq \ \mathbb{E}_g\Big[\tfrac{1}{2} \log\big((2\pi e)^d \det(\Sigma(g))\big)\Big] \ \leq \ \mathbb{E}_g\Big[\frac{d}{2} \log\Big(2\pi e \frac{\operatorname{Tr}(\Sigma(g))}{d}\Big)\Big].$$

Since $\log(\cdot)$ is concave, Jensen's inequality yields

$$\mathbb{E}_g\Big[\log\Big(2\pi e \frac{\operatorname{Tr}(\Sigma(g))}{d}\Big)\Big] \ \leq \ \log\Big(2\pi e \frac{\mathbb{E}_g[\operatorname{Tr}(\Sigma(g))]}{d}\Big).$$

Therefore

$$H(x \mid g) \ \leq \ \frac{d}{2} \log\Big(2\pi e \frac{\mathbb{E}_g[\operatorname{Tr}(\Sigma(g))]}{d}\Big). \tag{21}$$

By Proposition 1, $\mathbb{E}_g[\operatorname{Tr}(\Sigma(g))] = d\,\xi_g$, so equation 21 becomes

$$H(x \mid g) \ \leq \ \frac{d}{2} \log(2\pi e\,\xi_g).$$

Rearranging,

$$\log(2\pi e\,\xi_g) \ \geq \ \frac{2}{d} H(x \mid g), \qquad 2\pi e\,\xi_g \ \geq \ \exp\Big(\frac{2}{d} H(x \mid g)\Big),$$

which yields equation 19. $\qquad\qquad\square$

Note that $H(x)$—and hence the absolute scale of these lower bounds—is fully determined by the underlying dataset distribution and does not depend on teleportation. In our comparisons between teleported and non-teleported channels, $H(x)$ cancels and only *differences* or ratios matter.

## O.4 A parametric lower bound on $H(x|g)$ via Gaussian mixtures

We now introduce a specific probabilistic model for the gradient signal and derive a tractable parametric lower bound on $H(x \mid g)$. The modelling choices mirror those used for intermediate features in Xia et al. (2025), but here are applied to gradients.

### O.4.1 GRADIENT FEATURE AND OBSERVATION MODEL

**Clean gradient feature.** Let $G : \mathbb{R}^d \to \mathbb{R}^m$ be a deterministic mapping producing a *clean* gradient feature from input $x$. Specifically, let $u = G(x) \in \mathbb{R}^m$ denote a feature derived deterministically from $(x, y, W)$ (e.g., the gradient with respect to first-layer pre-activations, or a flattened stack of first-layer weight gradients). Thus $u$ is a deterministic function of $x$ once the model and label are fixed.

*Assumption* 2 (Gaussian Mixture Model (GMM) for $u$). We assume that the marginal distribution of $u$ can be well approximated by a Gaussian mixture

$$u \sim \sum_{i=1}^{K} \pi_i \mathcal{N}(\mu_i, \Sigma_i), \quad \sum_{i=1}^{K} \pi_i = 1, \quad \pi_i > 0, \quad \Sigma_i \succ 0. \tag{22}$$

This GMM assumption is standard in information-theoretic analyses of representations Xia et al. (2025) and serves as our common surrogate model for gradient features.

**Noisy gradient observation.** We model the attacker's baseline observation as a noisy version of $u$:

$$g_0 = u + \varepsilon, \quad \varepsilon \sim \mathcal{N}(0, \Sigma_g), \quad \varepsilon \perp (x, u), \tag{23}$$

where $\Sigma_g \succ 0$ is a fixed positive-definite covariance matrix. This captures gradient perturbations due to stochastic training, subsampling, or other noise sources; $\Sigma_g$ is assumed known to the attacker, as in Xia et al. (2025). We use this Gaussian channel as the standard abstraction of gradient perturbations for the subsequent information-theoretic analysis.

### O.4.2 A MUTUAL-INFORMATION IDENTITY FOR DETERMINISTIC FEATURES

We will repeatedly use the following simple lemma for deterministic features.

**Lemma 1** (Mutual information for deterministic feature maps). *Let $u = G(x)$ be a deterministic function of $x$, and let $g$ be a random variable such that $p(g \mid x, u) = p(g \mid u)$ (i.e., $g$ depends on $(x, u)$ only through $u$). Then*

$$I(x; g) = I(u; g).$$

Where $I(x; g)$ denotes the mutual information between $x$ and $g$.

*Proof.* Since $u$ is a deterministic function of $x$, we have $H(u \mid x) = 0$ and $H(x, u) = H(x)$. Moreover, $p(g \mid x) = p(g \mid u)$ by the conditional-independence assumption, so

$$H(g \mid x) = \mathbb{E}_x H(g \mid x = x) = \mathbb{E}_x H(g \mid u = G(x)) = H(g \mid u).$$

Therefore

$$I(x; g) = H(g) - H(g \mid x) = H(g) - H(g \mid u) = I(u; g). \qquad \square$$

We will apply this lemma to both the baseline channel $g_0$ and the teleported channel $g$ below.

### O.4.3 PARAMETRIC GMM-BASED LOWER BOUND ON $H(x \mid g_0)$

We now adapt the mixture-entropy bound used in Xia et al. (2025) to gradients.

**Theorem 2** (Parametric lower bound on $H(x \mid g_0)$). *Under Assumption 1 and Assumption 2 and the channel equation 23, the conditional entropy $H(x \mid g_0)$ satisfies*

$$H(x \mid g_0) \geq H(x) - \sum_{i=1}^{K} \pi_i \left( -\log \pi_i + \frac{1}{2} \log \frac{|\Sigma_i + \Sigma_g|}{|\Sigma_g|} \right). \tag{24}$$

*Proof.* Because $u = G(x)$ is deterministic given $x$, and $g_0$ depends on $(x, u)$ only through $u$ via equation 23, we have $g_0 \perp x \mid u$ and the conditions of Lemma 1 hold. Thus

$$I(x; g_0) = I(u; g_0),$$

and

$$H(x \mid g_0) = H(x) - I(x; g_0) = H(x) - I(u; g_0).$$

We bound $I(u; g_0)$ from above using the GMM model. We have

$$I(u; g_0) = H(g_0) - H(g_0 \mid u).$$

From equation 23, $g_0 \mid u \sim \mathcal{N}(u, \Sigma_g)$, so

$$H(g_0 \mid u) = \tfrac{1}{2} \log\big((2\pi e)^m |\Sigma_g|\big).$$

Marginally, $g_0$ is the convolution of the GMM $u$ with the Gaussian $\varepsilon$, hence

$$g_0 \sim \sum_{i=1}^{K} \pi_i \, \mathcal{N}(\mu_i, \Sigma_i + \Sigma_g).$$

For any mixture density $p(z) = \sum_i \pi_i p_i(z)$ with components $p_i$, the differential entropy satisfies the standard upper bound

$$H(p) \ \leq \ H(\pi) + \sum_i \pi_i H(p_i),$$

where $H(\pi) = -\sum_i \pi_i \log \pi_i$ is the discrete entropy of the mixture weights (this follows by considering the joint entropy of the component index and the sample). Applying this with Gaussian components $p_i = \mathcal{N}(\mu_i, \Sigma_i + \Sigma_g)$ yields

$$H(g_0) \ \leq \ \sum_{i=1}^{K} \pi_i \left( -\log \pi_i + \frac{1}{2} \log\big((2\pi e)^m |\Sigma_i + \Sigma_g|\big) \right),$$

as in Xia et al. (2025). Therefore

$$\begin{aligned}
I(u; g_0) &\leq \sum_{i=1}^{K} \pi_i \left( -\log \pi_i + \frac{1}{2} \log\big((2\pi e)^m |\Sigma_i + \Sigma_g|\big) \right) - \frac{1}{2} \log\big((2\pi e)^m |\Sigma_g|\big) \\
&= \sum_{i=1}^{K} \pi_i \left( -\log \pi_i + \frac{1}{2} \log\big((2\pi e)^m |\Sigma_i + \Sigma_g|\big) \right) + \sum_{i=1}^{K} \pi_i \left( -\frac{1}{2} \log\big((2\pi e)^m |\Sigma_g|\big) \right) \\
&= \sum_{i=1}^{K} \pi_i \left( -\log \pi_i + \frac{1}{2} \log\big((2\pi e)^m |\Sigma_i + \Sigma_g|\big) - \frac{1}{2} \log\big((2\pi e)^m |\Sigma_g|\big) \right) \\
&= \sum_{i=1}^{K} \pi_i \left( -\log \pi_i + \frac{1}{2} \log \frac{|\Sigma_i + \Sigma_g|}{|\Sigma_g|} \right).
\end{aligned}$$

where the $(2\pi e)^m$ terms cancel. Substituting into $H(x \mid g_0) = H(x) - I(u; g_0)$ yields equation 24. $\square$

Theorem 2 yields a parametric lower bound on $H(x \mid g_0)$—parametric in the GMM and noise covariances. Via Theorem 1, this in turn induces a lower bound on the minimal reconstruction MSE for an attacker observing $g_0$. Our teleportation analysis will reuse exactly the same ingredients (GMM approximation and mixture-entropy bound) so comparisons are on equal footing.

## O.5 TELEPORTATION / CHANGE-OF-BASIS NOISE ON GRADIENTS

We now incorporate teleportation (change-of-basis; CoB) symmetry as a source of private randomness in the gradient dynamics and analyze its impact on the *same* lower-bound pipeline used for $g_0$.

### O.5.1 TELEPORTATION AS PRIVATE MULTIPLICATIVE NOISE

**Teleportation structure.** For each layer $\ell$, let $\tau^{[\ell]}$ denote the corresponding CoB vector (with all entries nonzero). The teleported gradient at layer $\ell$ is obtained by column-scaling with $\tau^{[\ell-1]}$ and row-scaling with $1/\tau^{[\ell]}$, i.e.

$$dV^{[\ell]} \;=\; \tau^{[\ell-1]} \bullet dW^{[\ell]} \bullet \left(1/\tau^{[\ell]}\right),$$

$$dV_{ij}^{[\ell]} \;=\; \tau_j^{[\ell-1]} \, dW_{ij}^{[\ell]} \, \left(1/\tau_i^{[\ell]}\right),$$

where the left operation multiplies each column of $dW^{[\ell]}$ by the corresponding coordinate of $\tau^{[\ell-1]}$, and the right operation multiplies each row by the corresponding coordinate of $1/\tau^{[\ell]}$. Consequently, each gradient entry acquires a multiplicative factor equal to a ratio of CoB coordinates. As such, each gradient entry picks up a multiplicative factor equal to a ratio of CoB entries. Flattening all gradient parameters into a single vector, we write the clean gradient feature as $u$ and its teleported version as

$$\tilde{u} = R(\tau)\,u, \tag{25}$$

where $R(\tau)$ is a diagonal matrix with entries $r_j(\tau) = \tau_{b(j)}/\tau_{a(j)}$ corresponding to the appropriate input/output channels $(a(j), b(j))$ of coordinate $j$. In practice, these ratios are constrained by the underlying channel-wise $\tau^{[\ell]}$ structure; our analysis below treats $\{r_j(\tau)\}$ as effective per-coordinate scalings induced by that structure.

**Threat model for teleportation.** We adopt the following threat model.

*Assumption* 3 (Teleportation threat model).

(i) The CoB parameters $\tau$ are sampled from a distribution $P_\tau$ that is independent of $(x, u)$.
(ii) Teleportation is applied internally in the training update rule, so that the observable gradient feature (e.g., weight differences across a step) is a function of $\tilde{u}$ rather than $u$. Algebraically, this yields an observation of the form equation 26 below.
(iii) The adversary has white-box access to the model architecture and weights but *does not* observe $\tau$ directly. They know the distribution $P_\tau$.

$$g = \tilde{u} + \varepsilon = R(\tau)\,u + \varepsilon, \quad \varepsilon \sim \mathcal{N}(0, \Sigma_g), \quad \varepsilon \perp (x, u, \tau). \tag{26}$$

This is the same additive-noise form as in equation 23, applied to a multiplicatively perturbed feature $R(\tau)u$.

### O.5.2 TELEPORTATION-AWARE ENTROPY LOWER BOUND

We now derive the teleportation-aware counterpart of Theorem 2, using the same GMM approximation for $u$. Here the relevant mutual-information identity is again supplied by Lemma 1.

**Theorem 3** (Teleportation-aware lower bound on $H(x \mid g)$). *Under Assumption 1, Assumption 2, Assumption 3 and the teleported channel equation 26, the conditional entropy $H(x \mid g)$ satisfies*

$$H(x \mid g) \;\geq\; H(x) - \sum_{i=1}^{K} \pi_i \left( -\log \pi_i + \frac{1}{2} \mathbb{E}_\tau \log \frac{|R(\tau)\Sigma_i R(\tau)^\top + \Sigma_g|}{|\Sigma_g|} \right), \tag{27}$$

*where the expectation is taken w.r.t. $\tau \sim P_\tau$.*

*Proof.* As before, $u = G(x)$ is deterministic given $x$, and $g$ depends on $(x, u)$ only through $(u, \tau)$ via equation 26. In particular, we have the Markov chain

$$x \to u \to (g, \tau) \to g,$$

and $g \perp x \mid (u, \tau)$. Integrating over the independent $\tau$ yields $p(g \mid x, u) = p(g \mid u)$, and hence the conditions of Lemma 1 hold, giving

$$I(x; g) = I(u; g), \quad H(x \mid g) = H(x) - I(x; g) = H(x) - I(u; g).$$

We bound $I(u; g)$ from above. By the chain rule and independence of $u$ and $\tau$,

$$I(u; g) = I(u; g, \tau) - I(u; \tau \mid g) = I(u; g \mid \tau) - I(u; \tau \mid g) \;\leq\; I(u; g \mid \tau),$$

since $I(u; \tau \mid g) \geq 0$. Here $I(u; g \mid \tau)$ is conditional mutual information and can be written as

$$I(u; g \mid \tau) = \mathbb{E}_\tau \big[ I(u; g \mid \tau = t) \big].$$

For a fixed realization $\tau = t$, the channel is linear with Gaussian noise:

$$g \mid \tau = t = R(t)u + \varepsilon.$$

Conditionally on mixture component $i$, $u \mid i \sim \mathcal{N}(\mu_i, \Sigma_i)$, so

$$g \mid (i, \tau = t) \sim \mathcal{N}\big(R(t)\mu_i, \ R(t)\Sigma_i R(t)^\top + \Sigma_g\big),$$

and $g \mid \tau = t$ is a GMM with components indexed by $i$. For this fixed $t$,

$$I(u; g \mid \tau = t) = H(g \mid \tau = t) - H(g \mid u, \tau = t).$$

Since $g \mid (u, \tau = t) \sim \mathcal{N}(R(t)u, \Sigma_g)$, we obtain

$$H(g \mid u, \tau = t) = \tfrac{1}{2} \log\big((2\pi e)^m |\Sigma_g|\big).$$

Using the same mixture-entropy bound as before, applied to the GMM $g \mid \tau = t$, we have

$$H(g \mid \tau = t) \ \leq \ \sum_{i=1}^K \pi_i \left( -\log \pi_i + \tfrac{1}{2} \log\big((2\pi e)^m |R(t)\Sigma_i R(t)^\top + \Sigma_g|\big) \right).$$

Therefore

$$I(u; g \mid \tau = t) \leq \sum_{i=1}^K \pi_i \left( -\log \pi_i + \tfrac{1}{2} \log\big((2\pi e)^m |R(t)\Sigma_i R(t)^\top + \Sigma_g|\big) \right) - \tfrac{1}{2} \log\big((2\pi e)^m |\Sigma_g|\big)$$

$$= \sum_{i=1}^K \pi_i \left( -\log \pi_i + \tfrac{1}{2} \log \frac{|R(t)\Sigma_i R(t)^\top + \Sigma_g|}{|\Sigma_g|} \right).$$

Taking expectation over $\tau$ yields

$$I(u; g \mid \tau) = \mathbb{E}_\tau I(u; g \mid \tau = t) \ \leq \ \sum_{i=1}^K \pi_i \left( -\log \pi_i + \tfrac{1}{2} \mathbb{E}_\tau \log \frac{|R(\tau)\Sigma_i R(\tau)^\top + \Sigma_g|}{|\Sigma_g|} \right).$$

Combining $I(u; g) \leq I(u; g \mid \tau)$ with $H(x \mid g) = H(x) - I(u; g)$ gives equation 27. $\qquad\square$

Theorem 3 is the teleportation analogue of Theorem 2, obtained via the same steps, with $\Sigma_i$ replaced by $R(\tau)\Sigma_i R(\tau)^\top$ and an additional expectation over $\tau$.

## O.6 DIAGONAL APPROXIMATION AND THE ROLE OF THE COB DISTRIBUTION

To make the teleportation effect more interpretable at a per-coordinate level, we now adopt a diagonal approximation. This is a modelling simplification, similar in spirit to Xia et al. (2025), and all comparisons between teleported and baseline channels will be made *within* this shared surrogate approximation.

### O.6.1 DIAGONAL APPROXIMATION

*Assumption* 4 (Diagonal covariance approximation). We work in the canonical channel basis in which teleportation is defined and posit that, in this basis,

$$\Sigma_i = \mathrm{diag}(\sigma_{i,1}^2, \ldots, \sigma_{i,m}^2), \quad \Sigma_g = \mathrm{diag}(\gamma_1^2, \ldots, \gamma_m^2),$$

and the teleportation matrix has the form

$$R(\tau) = \mathrm{diag}(r_1(\tau), \ldots, r_m(\tau)).$$

That is, we adopt a surrogate model in which gradient covariance, observation noise and CoB factors act coordinatewise in the natural channel basis, rather than attempting to diagonalize arbitrary covariances and then reinterpret teleportation in that rotated frame. This is not claimed to be an exact description of real networks, but a structured approximation for per-coordinate interpretation.

Under Assumption 4,

$$R(\tau)\Sigma_i R(\tau)^\top + \Sigma_g = \mathrm{diag}\big(\gamma_1^2 + r_1(\tau)^2 \sigma_{i,1}^2, \ldots, \gamma_m^2 + r_m(\tau)^2 \sigma_{i,m}^2\big),$$

and hence

$$\frac{|R(\tau)\Sigma_i R(\tau)^\top + \Sigma_g|}{|\Sigma_g|} = \prod_{j=1}^{m}\Big(1 + \alpha_{i,j} r_j(\tau)^2\Big), \quad \alpha_{i,j} := \frac{\sigma_{i,j}^2}{\gamma_j^2}. \tag{28}$$

Taking logs and expectation in equation 27, we obtain

$$\mathbb{E}_\tau \log \frac{|R(\tau)\Sigma_i R(\tau)^\top + \Sigma_g|}{|\Sigma_g|} = \sum_{j=1}^{m} \psi_{i,j},$$

where we define the per-coordinate quantities

$$\psi_{i,j} := \mathbb{E}_\tau\big[\log(1 + \alpha_{i,j} r_j(\tau)^2)\big]. \tag{29}$$

Thus Theorem 3 becomes, under Assumption 4,

$$H(x \mid g) \geq H(x) - \sum_{i=1}^{K} \pi_i \left(-\log \pi_i + \frac{1}{2}\sum_{j=1}^{m} \psi_{i,j}\right). \tag{30}$$

### O.6.2   BASELINE (NON-TELEPORTED) DIAGONAL BOUND

For comparison, if no teleportation is applied, we have $R(\tau) \equiv I$ and $r_j(\tau)^2 \equiv 1$. Under the same diagonal surrogate,

$$\frac{|\Sigma_i + \Sigma_g|}{|\Sigma_g|} = \prod_{j=1}^{m}(1 + \alpha_{i,j}),$$

and the GMM-based entropy lower bound equation 24 reduces to

$$H(x \mid g_0) \geq H_0^{\mathrm{lb}} := H(x) - \sum_{i=1}^{K} \pi_i \left(-\log \pi_i + \frac{1}{2}\sum_{j=1}^{m} \log\big(1 + \alpha_{i,j}\big)\right). \tag{31}$$

We explicitly introduce $H_0^{\mathrm{lb}}$ to denote the analytic lower bound on $H(x \mid g_0)$ obtained under the GMM and diagonal surrogate.

Similarly, in the teleported diagonal setting equation 30 we define

$$H(x \mid g) \geq H_{\mathrm{tele}}^{\mathrm{lb}} := H(x) - \sum_{i=1}^{K} \pi_i \left(-\log \pi_i + \frac{1}{2}\sum_{j=1}^{m} \psi_{i,j}\right). \tag{32}$$

Both $H_0^{\mathrm{lb}}$ and $H_{\mathrm{tele}}^{\mathrm{lb}}$ are computed from exactly the same modelling ingredients and diagonal surrogate.

### O.6.3   ENERGY-PRESERVING CoB AND IMPROVEMENT OF THE BOUND

To isolate teleportation as a pure source of *randomization* (rather than a trivial global rescaling of gradient energy), we consider energy-preserving CoB distributions at the level of the per-coordinate effective ratios.

*Assumption* 5 (Energy-preserving CoB marginals).  For each coordinate $j$, the marginal distribution of $r_j(\tau)^2$ satisfies $\mathbb{E}_\tau[r_j(\tau)^2] = 1$.

This condition enforces that, on average, teleportation does not inflate or shrink per-coordinate gradient energy; it only redistributes it stochastically. In practice, the defender controls the sampling of $\tau$ and hence the induced distribution of ratios $\{r_j(\tau)\}$, subject to architectural constraints (shared

channels, etc.). We do not model those constraints explicitly here; we treat $\{r_j(\tau)\}$ as effective per-coordinate scalings whose marginals can be chosen to satisfy Assumption 5.

We do not assume independence of $r_j(\tau)$ across $j$, only these marginals.

Define, for each $(i, j)$,

$$\Delta\psi_{i,j} \; := \; \log\big(1 + \alpha_{i,j}\big) - \psi_{i,j} \; = \; \log\big(1 + \alpha_{i,j}\big) - \mathbb{E}_\tau\big[\log(1 + \alpha_{i,j} r_j(\tau)^2)\big]. \tag{33}$$

Subtracting equation 31 from equation 32 yields an exact relation between the two analytic entropy lower bounds under the diagonal surrogate.

**Corollary 1** (Exact relation between diagonal entropy lower bounds)**.** *Under Assumption 4, the diagonal entropy lower bounds equation 31–equation 32 satisfy*

$$H_{\text{tele}}^{\text{lb}} \; = \; H_0^{\text{lb}} + \frac{1}{2}\sum_{i=1}^{K}\pi_i\sum_{j=1}^{m}\Delta\psi_{i,j}, \tag{34}$$

*with $\Delta\psi_{i,j}$ defined in equation 33. If, in addition, Assumption 5 holds, then each $\Delta\psi_{i,j}$ is non-negative, and hence*

$$H_{\text{tele}}^{\text{lb}} \; \geq \; H_0^{\text{lb}}. \tag{35}$$

*Proof.* Equation equation 34 is obtained by direct subtraction of equation 31 from equation 32 and using equation 33. For the sign of $\Delta\psi_{i,j}$, fix $\alpha > 0$ and define $\phi_\alpha(t) := \log(1 + \alpha t)$, which is concave on $t > 0$. Under Assumption 5,

$$\psi_{i,j} = \mathbb{E}_\tau[\phi_{\alpha_{i,j}}(r_j(\tau)^2)] \; \leq \; \phi_{\alpha_{i,j}}\big(\mathbb{E}_\tau[r_j(\tau)^2]\big) = \phi_{\alpha_{i,j}}(1) = \log(1 + \alpha_{i,j}),$$

so $\Delta\psi_{i,j} \geq 0$ for all $(i, j)$, implying equation 35. $\qquad\square$

*Remark* 1 (Scope and strength of the entropy result). Within the shared modelling assumptions (GMM, diagonal surrogate, energy-preserving CoB), Corollary 1 shows that teleportation *never decreases* the analytic entropy lower bound:

$$H(x \mid g_0) \; \geq \; H_0^{\text{lb}}, \qquad H(x \mid g) \; \geq \; H_{\text{tele}}^{\text{lb}} \; \geq \; H_0^{\text{lb}}.$$

We stress that we do *not* claim $H(x \mid g) \geq H(x \mid g_0)$ for the true channels. Rather, we compare the surrogate quantities $H_0^{\text{lb}}$ and $H_{\text{tele}}^{\text{lb}}$ arising under the same generative model; under this common lens, teleportation strictly improves the analytic lower bound on uncertainty about $x$.

### O.7 TELEPORTATION-AWARE RECONSTRUCTION LOWER BOUND

We now translate the entropy bounds into reconstruction MSE lower bounds using Theorem 1.

#### O.7.1 BASELINE AND TELEPORTED MSE LOWER BOUNDS

From equation 31–equation 32 and Theorem 1, we obtain analytic lower bounds on the minimal reconstruction MSE for the baseline and teleported channels:

$$\underline{\xi}_0 \; := \; \frac{1}{2\pi e}\exp\left(\frac{2}{d}H_0^{\text{lb}}\right), \tag{36}$$

$$\underline{\xi}_{\text{tele}} \; := \; \frac{1}{2\pi e}\exp\left(\frac{2}{d}H_{\text{tele}}^{\text{lb}}\right). \tag{37}$$

By construction and monotonicity of the exponential,

$$\xi_{g_0} \; \geq \; \underline{\xi}_0, \qquad \xi_g \; \geq \; \underline{\xi}_{\text{tele}}, \tag{38}$$

where $\xi_{g_0}$ and $\xi_g$ are the true minimal MSEs for the baseline and teleported channels, respectively. Again, $H(x)$ is common to both channels and cancels in all *relative* statements about $\underline{\xi}_{\text{tele}}/\underline{\xi}_0$.

O.7.2 IMPROVEMENT FACTOR ON THE ANALYTIC MSE BOUND

Combining the definitions, the ratio between the teleported and baseline MSE *lower bounds* satisfies

$$\frac{\underline{\xi}_{\text{tele}}}{\underline{\xi}_0} = \exp\left(\frac{2}{d}(H_{\text{tele}}^{\text{lb}} - H_0^{\text{lb}})\right) = \exp\left(\frac{1}{d}\sum_{i=1}^{K}\pi_i\sum_{j=1}^{m}\Delta\psi_{i,j}\right), \tag{39}$$

with $\Delta\psi_{i,j}$ as in equation 33.

Under the energy-preserving assumption (Assumption 5), $\Delta\psi_{i,j} \geq 0$, hence the exponential factor in equation 39 is at least 1, and the analytic teleportation-aware MSE lower bound is never smaller than the baseline one. In other words, teleportation provably raises the information-theoretic floor on reconstruction accuracy *as captured by this shared surrogate model*. We do not assert any ordering between the true minimal MSEs $\xi_{g_0}$ and $\xi_g$.

*Remark* 2 (Interpretation for privacy). Equation equation 39 provides a quantitative, distribution-aware guarantee: under the shared assumptions (GMM, diagonal surrogate, energy-preserving CoB), teleportation inflates the analytic lower bound on the attacker's reconstruction MSE by a factor given by the RHS of equation 39. This factor depends on the CoB distribution only through $\Delta\psi_{i,j}$, which in turn are functions of the per-coordinate signal-to-noise ratios $\alpha_{i,j}$ and the marginals of $r_j(\tau)^2$. Thus teleportation is not merely a heuristic perturbation: for any attacker whose behaviour is dominated by this generative model (in essentially the same sense as in Xia et al. (2025)), there is a formal lower bound on how accurately they can reconstruct $x$.

O.8 LOG-NORMAL COB FAMILY (EFFECTIVE MODEL)

We now specialize the general diagonal analysis to an effective log-normal model for the CoB-induced per-coordinate scalings $r_j(\tau)^2$, to make the dependence on CoB variance explicit in the analytic MSE lower bounds.

**Log-normal marginal model.** We model each per-coordinate scaling as

$$r_j(\tau)^2 = \exp(Y_j),$$

where

$$Y_j \sim \mathcal{N}\left(-\tfrac{1}{2}s_j^2, \ s_j^2\right),$$

so that

$$\mathbb{E}[r_j(\tau)^2] = \mathbb{E}[e^{Y_j}] = \exp\left(-\tfrac{1}{2}s_j^2 + \tfrac{1}{2}s_j^2\right) = 1.$$

This ensures the energy-preserving condition $\mathbb{E}[r_j(\tau)^2] = 1$ (Assumption 5), while the parameter $s_j^2 \geq 0$ controls the strength of teleportation-induced variability on coordinate $j$. Practically, the defender can aim to implement such marginals by sampling $\tau$ so that the induced ratios $r_j(\tau)^2$ are approximately log-normal; we do not model the exact mapping from channel-wise $\tau^{[\ell]}$ to ratio marginals. We emphasize that this is an *effective parametric family* for $r_j^2$, chosen for analytical clarity; our rigorous inequalities rely only on Assumption 5, while log-normality is used to express the dependence on a small number of variance parameters.

Under this model, the per-coordinate quantities $\psi_{i,j}$ and $\Delta\psi_{i,j}$ admit explicit expressions.

**Corollary 2** (Log-normal teleportation and analytic MSE bound improvement). *Under the log-normal CoB marginal model above, for each mixture component $i$ and coordinate $j$,*

$$\psi_{i,j}(s_j^2) = \mathbb{E}_{Y_j \sim \mathcal{N}(-\frac{1}{2}s_j^2, s_j^2)}\left[\log(1 + \alpha_{i,j}e^{Y_j})\right], \tag{40}$$

*and*

$$\Delta\psi_{i,j}(s_j^2) = \log(1 + \alpha_{i,j}) - \mathbb{E}_{Y_j \sim \mathcal{N}(-\frac{1}{2}s_j^2, s_j^2)}\left[\log(1 + \alpha_{i,j}e^{Y_j})\right]. \tag{41}$$

*Let $\underline{\xi}_0$ and $\underline{\xi}_{\text{tele}}$ denote the analytic lower bounds on the minimal reconstruction MSE for the non-teleported and teleported channels, respectively, as defined in equation 36–equation 37. Then*

$$\frac{\underline{\xi}_{\text{tele}}(s^2)}{\underline{\xi}_0} = \exp\left(\frac{1}{d}\sum_{i=1}^{K}\pi_i\sum_{j=1}^{m}\left[\log(1 + \alpha_{i,j}) - \psi_{i,j}(s_j^2)\right]\right), \tag{42}$$

*where $s^2 = (s_1^2, \ldots, s_m^2)$ collects the log-variance parameters across coordinates.*

*Proof.* The identities equation 40–equation 41 are obtained by substituting $r_j^2 = e^{Y_j}$ with $Y_j \sim \mathcal{N}(-\frac{1}{2}s_j^2, s_j^2)$ into the definition equation 29 of $\psi_{i,j}$ and the definition equation 33 of $\Delta\psi_{i,j}$. The ratio equation 42 then follows immediately by plugging $\Delta\psi_{i,j}(s_j^2)$ into equation 39, which relates the analytic MSE lower bounds $\underline{\xi}_{\text{tele}}$ and $\underline{\xi}_0$ to the $\Delta\psi_{i,j}$. □

*Remark* 3 (Local small-variance expansion (heuristic)). To gain intuition about the dependence on teleportation strength, it is useful to consider the regime $s_j^2 \ll 1$, where the log-normal marginals are close to the degenerate case $r_j^2 \equiv 1$. This section provides a local Taylor expansion for intuition; it is *not* used in our rigorous inequalities, which already follow from Assumption 5.

For $t_j := r_j^2 = e^{Y_j}$ with $Y_j \sim \mathcal{N}(-\frac{1}{2}s_j^2, s_j^2)$ we have

$$\mathbb{E}[t_j] = 1, \qquad \text{Var}(t_j) = \mathbb{E}[t_j^2] - 1 = \exp(s_j^2) - 1.$$

Thus $\text{Var}(t_j) = s_j^2 + O(s_j^4)$ as $s_j^2 \to 0$. Writing $t_j = 1 + \delta_j$, we have $\mathbb{E}[\delta_j] = 0$ and $\text{Var}(\delta_j) = \text{Var}(t_j)$.

Since log-normal marginals have finite moments of all orders, a second-order Taylor expansion of $\phi_\alpha(t) := \log(1 + \alpha t)$ around $t = 1$ yields

$$\phi_\alpha(1 + \delta) = \log(1 + \alpha) + \frac{\alpha}{1 + \alpha}\delta - \frac{\alpha^2}{2(1 + \alpha)^2}\delta^2 + R_\alpha(\delta),$$

with $|R_\alpha(\delta)| \leq C_\alpha|\delta|^3$ for some constant $C_\alpha$ depending on $\alpha$. Taking expectations with $\mathbb{E}[\delta] = 0$ and $\mathbb{E}[\delta^2] = \text{Var}(t_j)$ gives

$$\mathbb{E}[\phi_\alpha(1 + \delta)] = \log(1 + \alpha) - \frac{\alpha^2}{2(1 + \alpha)^2}\text{Var}(t_j) + O\big(\mathbb{E}[|\delta|^3]\big).$$

Applying this with $\alpha = \alpha_{i,j}$ and $t_j = r_j^2$, and recalling that $\psi_{i,j} = \mathbb{E}[\log(1 + \alpha_{i,j}r_j^2)]$, we obtain the local approximation

$$\Delta\psi_{i,j}(s_j^2) = \log(1 + \alpha_{i,j}) - \psi_{i,j}(s_j^2) \approx \frac{\alpha_{i,j}^2}{2(1 + \alpha_{i,j})^2}\text{Var}(t_j),$$

with an error term controlled by $\mathbb{E}[|\delta_j|^3]$. For the log-normal model, $\text{Var}(t_j) = \exp(s_j^2) - 1$, so we arrive at the heuristic expression

$$\Delta\psi_{i,j}(s_j^2) \approx \frac{\alpha_{i,j}^2}{2(1 + \alpha_{i,j})^2}\big(\exp(s_j^2) - 1\big), \qquad s_j^2 \ll 1.$$

Substituting this into equation 42 yields the corresponding small-variance approximation for the logarithm of the analytic MSE bound ratio:

$$\log\frac{\underline{\xi}_{\text{tele}}(s^2)}{\underline{\xi}_0} \approx \frac{1}{2d}\sum_{i=1}^{K}\pi_i\sum_{j=1}^{m}\frac{\alpha_{i,j}^2}{(1 + \alpha_{i,j})^2}\big(\exp(s_j^2) - 1\big), \qquad s_j^2 \ll 1.$$

This approach highlight that, in the small-variance regime, the teleportation-induced improvement in the analytic reconstruction MSE lower bound grows approximately linearly in $s_j^2$ (via $\exp(s_j^2) - 1$), with a slope governed by the per-coordinate signal-to-noise ratios $\alpha_{i,j}$ and the mixture weights $\pi_i$

# P   ABLATION: SENSITIVITY OF TELEPORTATION HYPERPARAMETERS

Teleportation introduces a small set of additional hyperparameters that control how strongly we move along symmetry directions. In this section we study the sensitivity of WARP to two core choices: (i) the target retain-variance fraction used to choose the per-layer rank $k$ in the SVD projector (Section 3.2), and (ii) the size of the retain minibatch $B_r$ used to estimate the retain subspace. Both directly govern the geometry of the retain null space and the amount of stochasticity in the teleportation step, and were explicitly highlighted as potential sources of instability.

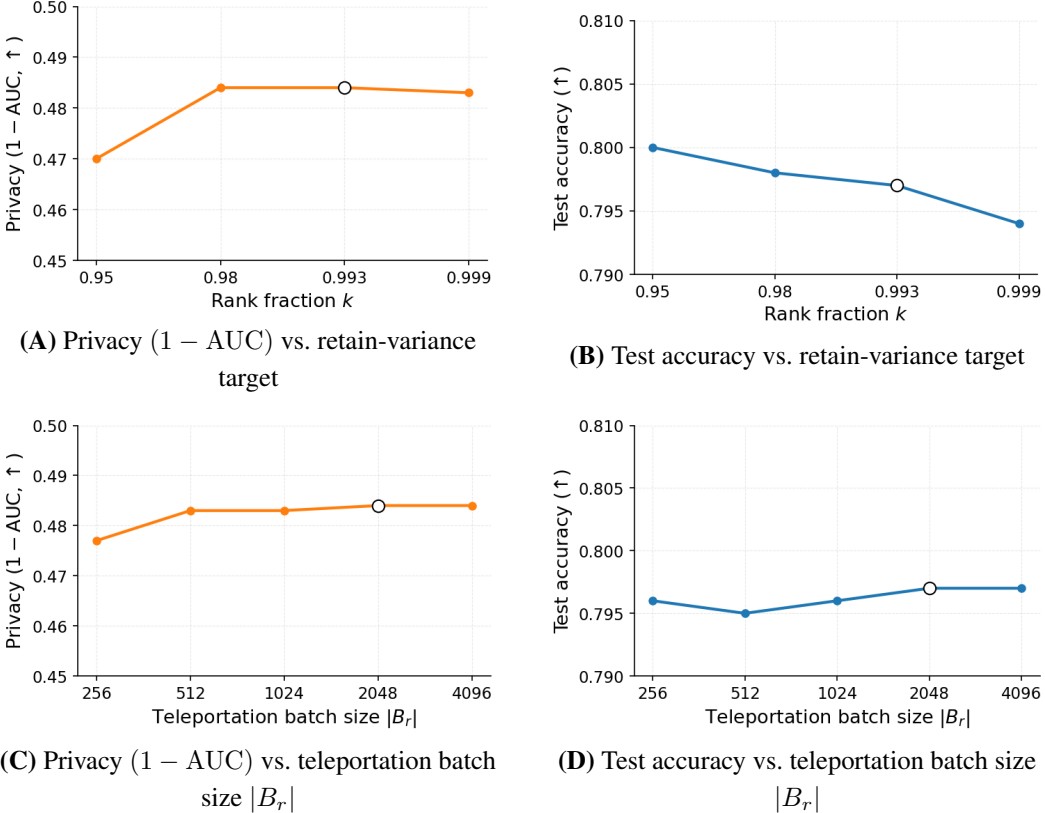

Figure 12: **Sensitivity of teleportation hyperparameters.** Plots (A,B) vary the target retain-variance level used to set the per-layer rank $k_\ell$; plots (C,D) vary the retain minibatch size $|B_r|$ used to estimate the retain subspace. Privacy is measured as $1 - \text{AUC}$ of U-LiRA (higher is better). Markers highlight the configuration used in our main experiments (95.3% retain variance and $|B_r| = 2048$).

**Setup.** We perform a controlled sweep on CIFAR-10 with ResNet-18 and NGP+WARP under the U-LiRA black-box auditor (Section 4.2). For the SVD projector, we vary the target retain-variance level from $95\%$ to $99.9\%$, which induces different per-layer ranks $k_\ell$ such that the top singular vectors of $R_\ell(D_r)$ capture the chosen fraction of retain energy. For the retain minibatch, we vary the teleportation batch size $|B_r| \in \{256, 512, 1024, 2048, 4096\}$ while keeping the forget minibatch and unlearning hyperparameters fixed. For each configuration we run the full unlearning pipeline and record test accuracy as well as privacy measured by $(1 - \text{AUC})$ of U-LiRA (higher is better).

**Results and discussion.** Figure 12 shows that teleportation is *remarkably insensitive* to both hyperparameters in the regime we consider.

*Retain-variance target.* Increasing the target retain-variance from $95\%$ to $99.9\%$ changes privacy $(1 - \text{AUC})$ by less than $0.015$ in absolute terms, while test accuracy varies in a narrow band of $\approx 0.79$–$0.80$. Privacy slightly improves as we move from $95\%$ to around $99.3\%$, after which the curve flattens: very high targets effectively make the retain projector full-rank, leaving less room for teleportation to move in symmetry directions and yielding diminishing returns. The configuration used in the main experiments (target retain-variance $\approx 99.3\%$) lies near this plateau, indicating that our chosen rank provides a good privacy–utility compromise.

*Retain minibatch size $|B_r|$.* Varying $|B_r|$ over an order of magnitude has only a minor effect: privacy $(1 - \text{AUC})$ shifts by at most $\sim 0.01$, and test accuracy remains within $\pm 0.2\%$ points of $0.796$. Even relatively small batches ($|B_r| = 256$) already provide a sufficiently representative retain subspace for teleportation, and larger batches only yield a slight, saturating gain in privacy. This suggests that the random retain minibatch need not tightly approximate the full retain set to obtain a stable projector

and effective defense; in practice, a modest $|B_r|$ balances computational cost with stable subspace estimation.

Overall, these ablations show that WARP's performance does not hinge on fragile hyperparameter choices: both privacy and utility are stable across wide ranges of the SVD rank and retain minibatch size. Moreover, the small spread in test accuracy ($< 0.6\%$ across all settings) empirically confirms that teleportation remains approximately loss-preserving on the retain set, providing an implicit bound on worst-case retain-loss drift in our experiments.

