$$\text{s.t.} \quad \ell_{\mathrm{r}}(g \cdot \theta \,|\, \mathcal{D}_{\mathrm{r}}) \leq \ell_{\mathrm{r}}(\theta \,|\, \mathcal{D}_{\mathrm{r}}) + \varepsilon.$$

with trade-off $\beta \geq 0$ and tolerance $\varepsilon \geq 0$. The first term reduces squared gradient norms of forget examples (Motivation II); the dispersion term adds symmetry-preserving randomness, displacing parameters from $\theta^{\mathrm{org}}$ (Motivation I); the constraint preserves retain performance.

WARP operates on an abstract prediction-preserving symmetry map $T_\phi$, and any such symmetry family can instantiate the framework. In practice, we use two concrete realizations—the retain–null-space projection introduced in the next paragraph, and the change-of-basis teleportation detailed in Appendix D—to illustrate this generality. To complement this algorithmic view, Appendix O develops teleportation-aware information-theoretic bounds on gradient-based reconstruction, showing how injecting symmetry-induced noise via $T_\phi$ expands the symmetry orbit and provably increases the expected reconstruction error for attackers observing $(\theta^{\mathrm{org}}, \theta^u)$.

**Primary instantiation: teleportation with retain null-space projection.** We first describe one convenient way to instantiate $T_\phi$ using retain–null-space projections Wu et al. (2025). To optimize equation 7 efficiently on modern architectures without explicit group actions, we adopt *teleportation with input null-space gradient projection* Wu et al. (2025) and instantiate it using the recent projector formulation that keeps updates on the loss-invariant level set by per-layer projections onto the input null space (thus leaving the task loss unchanged up to numerical error). Concretely, define the *teleportation loss*

$$\mathcal{L}_{\mathrm{tel}}(\theta) = \sum_{(x,y) \in \mathcal{B}_{\mathrm{f}}} \left\| \nabla_\theta \ell\big(f(x; \theta), y\big) \right\|_2^2 - \beta \|\theta - \theta^{\mathrm{org}}\|_2^2,$$

where $\mathcal{B}_{\mathrm{f}}$ is a minibatch from $\mathcal{D}_{\mathrm{f}}$. Let $R_\ell$ be the per-layer representation matrix from a *retain* minibatch (layer-$\ell$ inputs), with thin SVD $R_\ell = U_\ell \Sigma_\ell V_\ell^\top$. We keep the top-$k$ left singular vectors $B_\ell = U_{\ell,1:k}$ to span the retain subspace and define the orthogonal projector onto its complement $\Pi_\ell^\perp = I - B_\ell B_\ell^\top$. A teleportation step then applies the layer-wise update

$$W_\ell^{t+1} \leftarrow W_\ell^t - \eta_{\mathrm{tel}} \, \Pi_\ell^\perp \big(\nabla_{W_\ell} \mathcal{L}_{\mathrm{tel}}(\theta^t)\big) \tag{8}$$

which (i) *reduces* the forget-set gradient norms by descending on $\mathcal{L}_{\text{tel}}$, (ii) *preserves* the function on the retain-set by restricting motion to the retain-orthogonal subspace. The projection operator in equation 8 corresponds to the input-null-space projector. This is implemented by subtracting the component in the subspace of the core gradient, leaving only the residual for the teleport step.

To align the invariance with utility preservation, we compute $B_\ell$ *only from retain data*: ~~if $R_\ell(\mathcal{D}_r) = [\phi_\ell(x)]_{\overline{x} \in \mathcal{B}_r}$ stacks~~. Let $R_\ell(\mathcal{D}_r) = [\phi_\ell(x)]_{x \in \mathcal{B}_r}$ denote the matrix formed by stacking the layer-$\ell$ inputs for a retain minibatch $\mathcal{B}_r$~~, then~~

$$R_\ell(\mathcal{D}_r) = U_\ell \Sigma_\ell V_\ell^\top, \quad B_\ell = U_{\ell,1:k}, \quad \Pi_\ell^\perp = I - B_\ell B_\ell^\top,$$

. Then:

$$R_\ell(\mathcal{D}_r) = U_\ell \Sigma_\ell V_\ell^\top, \qquad B_\ell = U_{\ell,1:k}, \qquad \Pi_\ell^\perp = I - B_\ell B_\ell^\top. \tag{9}$$

We set $k$ to capture a fixed fraction of retain variance (typically ~~90%–95%~~ 95%–99%) and apply the resulting projectors in equation 8. This confines each teleport step to the retain-orthogonal subspace, stabilizing predictions on $\mathcal{D}_r$ while suppressing gradient energy on $\mathcal{D}_f$. Since $\Pi_\ell^\perp$ removes directions spanned by retain representations, suitable choices of rank $k$ and step size $\eta_{\text{tel}}$ ensure that

$$\left| \ell_r(g \cdot \theta \,|\, \mathcal{D}_r) - \ell_r(\theta \,|\, \mathcal{D}_r) \right| \leq \varepsilon,$$

which matches the constraint below equation 7; ~~empirically, the~~ in practice, prediction drift on $\mathcal{D}_r$ ~~is~~ remains within numerical tolerance ~~. An alternative instantiation of symmetry is detailed in~~ (

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

. ~~Approximations such as updating the subspace at intervals rather than per step could further lower the cost, though runtime optimization is left for future work~~While updating the retain subspace less frequently can reduce cost, the primary computational overhead from full SVD is addressed directly by an approximate low-rank implementation (Appendix L), which removes the per-step bottleneck entirely.

## K  TELEPORTATION-BASED UNLEARNING ALGORITHM

In Algorithm 3, $T_\phi$ denotes an abstract symmetry operator; in our experiments we instantiate it either with retain–null-space teleportation or with change-of-basis teleportation, but any other loss-preserving symmetry could be used in its place.

---

**Algorithm 3** ~~Teleportation–Augmented Unlearning~~ WARP (~~TAU~~retain–null-space instantiation): teleportation-augmented gradient-based unlearning.

---

**Require:** $\theta^{\mathrm{org}}, \mathcal{D}_{\mathrm{f}}, \mathcal{D}_{\mathrm{r}}, \ell_{\mathrm{f}}, \ell_{\mathrm{r}}, \lambda, \beta, \{\eta_t\}, \eta_{\mathrm{tel}}, k, S \text{ or } \tau_{\mathrm{grad}}, \sigma^2, \varepsilon, T$

1: $\theta_0 \leftarrow \theta^{\mathrm{org}}$
2: **for** $t = 0, \ldots, T-1$ **do**
3:      sample $\mathcal{B}_{\mathrm{f}} \subset \mathcal{D}_{\mathrm{f}}, \ \mathcal{B}_{\mathrm{r}} \subset \mathcal{D}_{\mathrm{r}}$
4:      $\theta_{t+\frac{1}{2}} \leftarrow \theta_t - \eta_t\big(\nabla_\theta \ell_{\mathrm{f}}(\theta_t \mid \mathcal{B}_{\mathrm{f}}) + \lambda \nabla_\theta \ell_{\mathrm{r}}(\theta_t \mid \mathcal{B}_{\mathrm{r}})\big)$
5:      **if** $(t \bmod S = 0) \ \vee \ \|\nabla_\theta \ell_{\mathrm{f}}(\theta_{t+\frac{1}{2}} \mid \mathcal{B}_{\mathrm{f}})\|_2 > \tau_{\mathrm{grad}}$ **then**
6:          **for** layer $\ell$ **do**
7:              build $R_\ell(\mathcal{B}_{\mathrm{r}}); \quad R_\ell = U_\ell \Sigma_\ell V_\ell^\top$ (SVD)
8:              $B_\ell \leftarrow U_{\ell,1:k}; \quad \Pi_\ell^\perp \leftarrow I - B_\ell B_\ell^\top$
9:          **end for**
10:         $\mathcal{L}_{\mathrm{tel}}(\theta) = \frac{1}{2}\sum_{(x,y)\in\mathcal{B}_{\mathrm{f}}} \|\nabla_\theta \ell(f(x;\theta), y)\|_2^2 - \frac{\beta}{2}\|\theta - \theta^{\mathrm{org}}\|_2^2$
11:         **for** layer $\ell$ **do**
12:             $W_\ell^{t+1} \leftarrow W_\ell^{t+\frac{1}{2}} - \eta_{\mathrm{tel}} \, \Pi_\ell^\perp \big(\nabla_{W_\ell} \mathcal{L}_{\mathrm{tel}}(\theta_{t+\frac{1}{2}})\big) + \textcolor{red}{\sqrt{2\,\eta_{\mathrm{tel}}\,\sigma^2}\,\varepsilon_{\ell,t}}$
13:                                                           $\varepsilon_{\ell,t} \sim \mathcal{N}(0, I)$
14:         **end for**
15:         $\theta_{t+1} \leftarrow \{W_\ell^{t+1}\}_\ell$
16:         **if** $\ell_{\mathrm{r}}(\theta_{t+1} \mid \mathcal{B}_{\mathrm{r}}) > \ell_{\mathrm{r}}(\theta_t \mid \mathcal{B}_{\mathrm{

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

\left[H(x \mid g = g)\right] \leq \mathbb{E}_g\left[\tfrac{1}{2} \log\left((2\pi e)^d \det(\Sigma(g))\right)\right]. \tag{20}$$

Let $\lambda_1(g), \ldots, \lambda_d(g)$ be the eigenvalues of $\Sigma(g)$ (all positive). Then

$$\det(\Sigma(g)) = \prod_{j=1}^{d} \lambda_j(g), \quad \mathrm{Tr}(\Sigma(g)) = \sum_{j=1}^{d} \lambda_j(g).$$

By the Arithmetic Mean-Geometric Mean (AM–GM) inequality,

$$\prod_{j=1}^{d} \lambda_j(g) \leq \left(\frac{1}{d} \sum_{j=1}^{d} \lambda_j(g)\right)^d = \left(\frac{\mathrm{Tr}(\Sigma(g))}{d}\right)^d,$$

so

$$\log \det(\Sigma(g)) \leq d \log\left(\frac{\mathrm{Tr}(\Sigma(g))}{d}\right).$$

Substituting into equation 20,

$$H(x \mid g) \leq \mathbb{E}_g\left[\tfrac{1}{2} \log\left((2\pi e)^d \det(\Sigma(g))\right)\right] \leq \mathbb{E}_g\left[\frac{d}{2} \log\left(2\pi e \frac{\mathrm{Tr}(\Sigma(g))}{d}\right)\right].$$

Since $\log(\cdot)$ is concave, Jensen's inequality yields

$$\mathbb{E}_g\left[\log\left(2\pi e \frac{\mathrm{Tr}(\Sigma(g))}{d}\right)\right] \leq \log\left(2\pi e \frac{\mathbb{E}_g[\mathrm{Tr}(\Sigma(g))]}{d}\right).$$

Therefore

$$H(x \mid g) \leq \frac{d}{2} \log\left(2\pi e \frac{\mathbb{E}_g[\mathrm{Tr}(\Sigma(g))]}{d}\right). \tag{21}$$

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

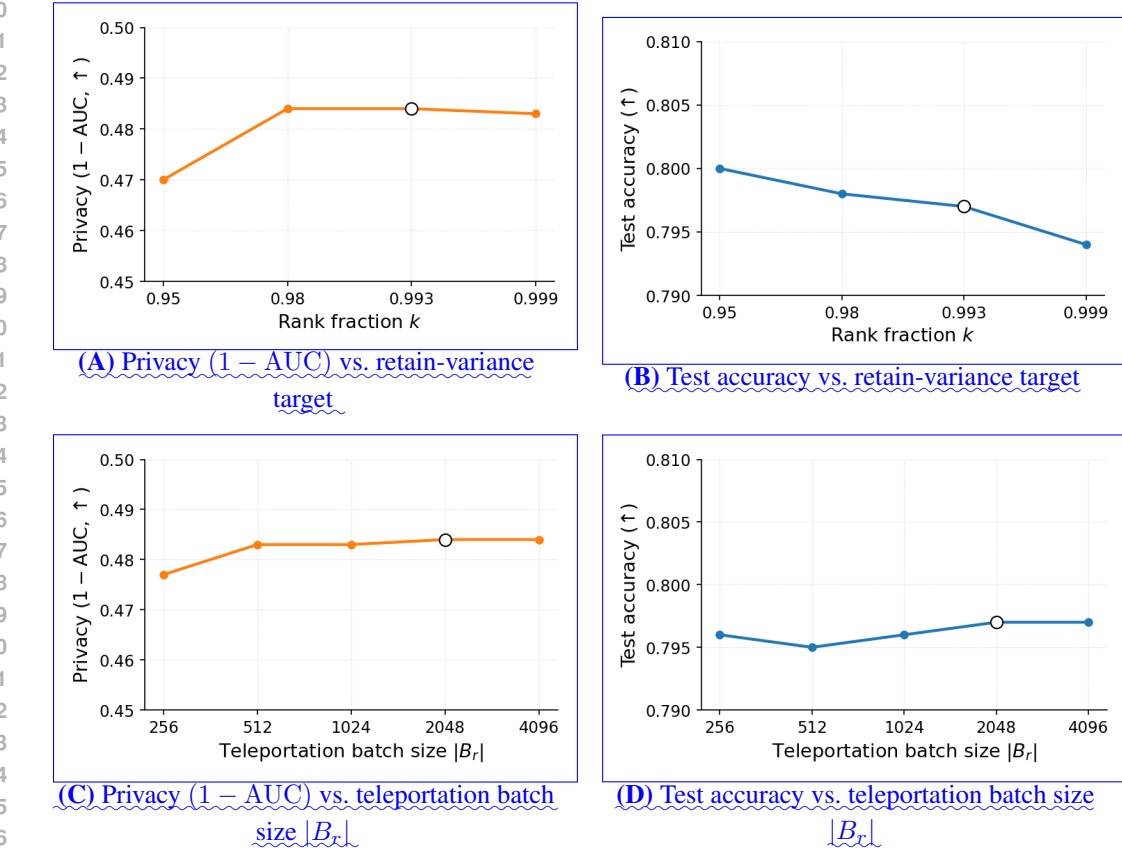

**(A)** Privacy $(1 - \text{AUC})$ vs. retain-variance target

**(B)** Test accuracy vs. retain-variance target

**(C)** Privacy $(1 - \text{AUC})$ vs. teleportation batch size $|B_r|$

**(D)** Test accuracy vs. teleportation batch size $|B_r|$

Figure 12: **Sensitivity of teleportation hyperparameters.** Plots (A,B) vary the target retain-variance level used to set the per-layer rank $k_\ell$; plots (C,D) vary the retain minibatch size $|B_r|$ used to estimate the retain subspace. Privacy is measured as $1 - \text{AUC}$ of U-LiRA (higher is better). Markers highlight the configuration used in our main experiments (95.3% retain variance and $|B_r| = 2048$).

Overall, these ablations show that WARP's performance does not hinge on fragile hyperparameter choices: both privacy and utility are stable across wide ranges of the SVD rank and retain minibatch size. Moreover, the small spread in test accuracy ($< 0.6\%$ across all settings) empirically confirms that teleportation remains approximately loss-preserving on the retain set, providing an implicit bound on worst-case retain-loss drift in our experiments.