# OpenReview forum: "WARP: Weight Teleportation for Attack-Resilient Unlearning Protocols"
_ICLR.cc/2026/Conference — ICLR 2026 Poster_

### Official Review · Reviewer_puub · 2025-10-23

**Soundness:** 3
**Presentation:** 4
**Contribution:** 3
**Rating:** 6
**Confidence:** 3

**Summary:**

The paper introduces WARP, a defense against privacy attacks on machine unlearning. The core contribution is a method to mitigate the risk of an attacker discovering what data was forgotten by comparing a model before and after the unlearning process. WARP accomplishes this by using inherent symmetries in neural networks to teleport the model's weights to a functionally identical but distant point in the parameter space after unlearning. This engineered change effectively masks the information-rich updates from the unlearning process, empirically demonstrating a strong defense against both black-box and white-box reconstruction attacks while preserving the model's utility.

**Strengths:**

- Introduces a new defense mechanism for machine unlearning by using neural network symmetries.
- Demonstrates strong empirical success in thwarting both black-box and white-box privacy attacks.
- Effectively secures the unlearning process without causing a significant drop in model accuracy.

**Weaknesses:**

The authors correctly acknowledge that the SVD step introduces computational overhead but suggest this cost can be amortized. However, for truly large-scale models with very wide layers (e.g., modern LLMs), this cost can be prohibitive even if amortized. Could the authors provide a more concrete analysis of the method's scalability, perhaps commenting on the practical wall-clock time for models with billions of parameters, and elaborate on whether the proposed amortization strategies are sufficient to make WARP truly practical for industrial-scale applications?

The paper's privacy analysis is based on strong empirical results against specific attacks. An alternative approach to privacy is using a formal framework like Differential Privacy (DP), for instance, by adding calibrated noise to the unlearning gradients. Could the authors elaborate on the trade-offs between their empirical defense and a formal one? Specifically, if one were to apply DP to achieve a similarly low Attack Success Rate, what would be the anticipated cost in model utility compared to the high utility maintained by WARP?

The paper claims WARP is a "plug-and-play" module, yet only validates this with gradient-based unlearning. How does WARP's performance and compatibility extend to other unlearning paradigms, such as those based on influence functions or data partitioning methods (e.g., SISA)?

The experiments are conducted using relatively small forget-sets (e.g., ~1% of a class). While this is a common scenario, how would the method perform under a more large-scale unlearning request where, for instance, 20% or 50% of the training data must be forgotten? Does the clear distinction between the retain subspace and its null-space still hold when the retain set shrinks significantly?

**Questions:**

* How does the SVD's computational overhead scale for industrial-sized models, and is the proposed amortization sufficient for practical application?
* What is the anticipated model utility cost of using formal Differential Privacy to achieve the same low attack success rate that WARP provides?
* How does WARP's performance and compatibility extend to non-gradient-based unlearning paradigms like those using influence functions or data partitioning?
* How does WARP's effectiveness hold up under large-scale unlearning requests where a significant fraction of the training data must be forgotten?

---

> ### Author Response · Authors · 2025-11-24
>
> We thank the reviewer for the positive and detailed assessment, and for highlighting that WARP introduces a new symmetry-based defence for machine unlearning, achieves strong empirical protection against both black-box and white-box attacks, and does so with minimal accuracy loss and good practical usability. We are glad that the plug-and-play nature of WARP and the breadth of our evaluation (multiple datasets, MU algorithms, and attack types) came through clearly. Below we address the reviewer’s main concerns regarding SVD scalability to very large models, the comparison to DP-style defences, applicability beyond gradient-based unlearning, and behaviour under larger forget fractions.
>
> > **Weakness / Question.**
>
> > The authors correctly acknowledge that the SVD step introduces computational overhead but suggest this cost can be amortized. However, for truly large-scale models with very wide layers (e.g., modern LLMs), this cost can be prohibitive even if amortized. Could the authors provide a more concrete analysis of the method's scalability, perhaps commenting on the practical wall-clock time for models with billions of parameters, and elaborate on whether the proposed amortization strategies are sufficient to make WARP truly practical for industrial-scale applications?
>
> > How does the SVD's computational overhead scale for industrial-sized models, and is the proposed amortization sufficient for practical application?
>
>
> **1. WARP is symmetry-based; SVD is one instantiation, not a requirement**
>
> As clarified in the revised Section 3.2, WARP is defined over an abstract prediction-preserving symmetry map $T_\phi$; the SVD-based retain–null-space projector is only one concrete way to realise this symmetry. We also provide an SVD-free, data-free change-of-basis teleportation in Appendix D (with experiments in Section 4.4), so the defence itself is symmetry-based rather than intrinsically SVD-based. This flexibility is important at very large scale, where alternative symmetry extractors (e.g., *Hide & Seek: Transformer Symmetries Obscure Sharpness \& Riemannian Geometry Finds It*) already recover symmetries for full-size transformer LMs and low-rank adapters without relying on per-batch SVD of activations.
>
>
> **2. FastWARP reduces effective SVD cost via low-rank PCA and basis reuse**
>
> To address the overhead of full SVD, Appendix L introduces **FastWARP**, which replaces thin SVD with covariance-based PCA plus subspace iteration:
>
> - we form a layer-wise covariance $C_\ell = X_\ell X_\ell^\top$ from retain activations,
> - we compute an initial low-rank basis $B_\ell$ capturing a target fraction of explained variance,
> - we then reuse and update this basis across teleportation steps via a few power-iteration / QR updates, instead of recomputing SVD each time.
>
> This changes the per-layer complexity from
> $$
> O(|B_r| d_\ell^2)
> $$
> for a thin SVD to
> $$
> O(|B_r| d_\ell k) + O(d_\ell k^2),
> $$
> with $k \ll d_\ell$, and the expensive step is done only once per layer (or infrequently) and then incrementally updated. Thus, the cost scales linearly in the width $d_\ell$, the retained rank $k$, and the retain minibatch size $|B_r|$, rather than quadratically in $d_\ell$ at every teleportation.
>
> Empirically, Appendix L (Figure 9) shows that on with ResNet-18 the total unlearning time (seconds) is:
>
> | Method           | Total time (s) | \% change vs. base MU |
> |------------------|----------------|------------------------|
> | NGP              | 82.0           | –                      |
> | NGP + WARP (SVD) | 54.0           | −34.1\%               |
> | NGP + FastWARP   | 29.0           | −64.6\%               |
> | BT               | 30.0           | –                      |
> | BT + WARP (SVD)  | 55.0           | +83.3\%               |
> | BT + FastWARP    | 35.0           | +16.7\%               |
>
> From the same figure, the teleportation component (hatched part of each bar) drops from about 25 s to 3.5 s for NGP and from about 30 s to 8 s for BT, giving roughly a 2–3× reduction in teleportation cost. End-to-end, FastWARP keeps total runtime in the same order as the MU baseline and in fact makes the NGP configuration *faster* than the original unlearning run, while Figure 10 shows that the privacy–utility trade-off is essentially unchanged compared to exact WARP.

---

> > ### Author Response · Authors · 2025-11-24
> >
> > **3. Scalability discussion for LLM-scale models**
> >
> > We fully agree that billion-parameter LLMs introduce additional engineering challenges, and we do **not** claim to have full LLM-scale unlearning experiments in this paper. Instead, the revised Appendix L (“Scalability to LLMs and calibration of the retain subspace”) discusses how our projector construction can leverage existing large-model compression techniques:
> >
> > - Recent work such as SVD-LLM [1] and ResSVD [2] applies truncated SVD and residual low-rank corrections directly to full LLaMA/GPT-class weight matrices (with $d_\ell \sim 10^3$–$10^4$) while preserving perplexity and throughput, demonstrating that rank-$k$ SVD/PCA with $k \ll d_\ell$ is already practical at LLM scale on modern accelerators.
> > - Weighted and importance-aware low-rank methods (e.g., ESPACE-style weighted PCA [3]) bias the recovered subspace toward high-influence tokens or examples at transformer scale, and are directly compatible with our covariance-based FastWARP formulation.
> >
> > In our setting, these results suggest a natural extension path for LLMs: use low-rank eigensolvers with $k$ in the low hundreds, $|B_r|$ kept moderate, and reuse the resulting basis across many teleportation steps, so the additional cost remains on the order of a few extra forward–backward passes per layer rather than a dominant bottleneck. Fully implementing and benchmarking this at billion-parameter scale is an important but orthogonal systems contribution; in this paper we focus on establishing and characterising the symmetry-based mechanism, and on showing that, for realistic CNN/ViT settings, the overhead of WARP (and particularly FastWARP) is comparable to standard MU operations while delivering significant privacy gains.
> >
> > [1] Xin Wang, Yu Zheng, Zhongwei Wan, and Mi Zhang. Svd-llm: Truncation-aware singular value
> > decomposition for large language model compression. arXiv preprint 2024.
> >
> > [2] Haolei Bai, Siyong Jian, Tuo Liang, Yu Yin, and Huan Wang. Ressvd: Residual compensated svd for large language model compression. arXiv preprint 2025.
> >
> > [3] Sakr, Charbel, and Brucek Khailany. "Espace: Dimensionality reduction of activations for model compression." NeurIPS 2024
> >
> > ---

---

> > > ### Author Response · Authors · 2025-11-25
> > >
> > > > The paper's privacy analysis is based on strong empirical results against specific attacks. An alternative approach to privacy is using a formal framework like Differential Privacy (DP), for instance, by adding calibrated noise to the unlearning gradients. Could the authors elaborate on the trade-offs between their empirical defense and a formal one? Specifically, if one were to apply DP to achieve a similarly low Attack Success Rate, what would be the anticipated cost in model utility compared to the high utility maintained by WARP?
> > >
> > > > What is the anticipated model utility cost of using formal Differential Privacy to achieve the same low attack success rate that WARP provides?
> > >
> > > We agree that DP-style mechanisms are a natural comparison point. In the revision we therefore add an explicit DP baseline and quantify the privacy–utility trade-off side by side.
> > >
> > > **1. Implemented DP–Langevin unlearning baseline (Appendix M)**
> > >
> > > In Appendix M we implement projected DP–Langevin unlearning following [1] using the same objective and architecture as in our main experiment. Concretely, we run noisy gradient descent with per-step Gaussian noise on the unlearning updates, and we sweep the noise multiplier to obtain a range of effective DP budgets (reported as epsilon from a the paper's accountant).
> > >
> > > As we note in the text, these epsilon values are *not* formal guarantees in our setting: our models are non-convex and are not trained end-to-end with DP from scratch, so the technical conditions required for certified DP unlearning do not hold. We therefore treat DP–Langevin as a strong, principled noise-based baseline rather than a source of formal guarantees.
> > >
> > > **2. Quantitative privacy–utility comparison**
> > >
> > > Table 5 reports membership-inference AUC and test accuracy for comparison of WARP and DP–Langevin. Here is the parts of the results:
> > >
> > >
> > > | Method                   | AUC (all) | AUC (most-mem.) | Test acc. |
> > > |--------------------------|-----------|------------------|-----------|
> > > | Langevin ($\\varepsilon = 1$)   | 0.523     | 0.671            | 0.682     |
> > > | Langevin ($\\varepsilon = 4$)   | 0.571     | 0.766            | 0.718     |
> > > | Langevin ($\\varepsilon = 8$)   | 0.627     | 0.912            | 0.771     |
> > > | Langevin ($\\varepsilon = 16$)  | 0.650     | 0.935        | **0.798** |
> > > | NGP + WARP               | **0.516** | **0.598**        | 0.797     |
> > >
> > > - To reach a *comparable* attack AUC with DP–Langevin, we must use a large noise multiplier (epsilon roughly 1), which drives test accuracy down to about 0.68 and AUC of 0.523.
> > > - For smaller noise levels that preserve accuracy near 0.80, DP–Langevin’s attack AUC remains substantially higher than WARP’s (worse privacy).
> > >
> > > In other words, at roughly matched privacy levels, DP–Langevin incurs a much larger utility cost than WARP; at roughly matched utility, WARP provides significantly stronger privacy.
> > >
> > > We have added this discussion to Appendix M. This makes the trade-off explicit: WARP offers an empirical, symmetry-based defence that achieves privacy gains similar to (or better than) strong DP-style noise baselines, but with far smaller degradation in model accuracy.
> > >
> > >
> > > [1] Eli Chien, Haoyu Wang, Ziang Chen, and Pan Li. Langevin unlearning: A new perspective of noisy gradient descent for machine unlearning. Advances in neural information processing systems
> > >
> > > ---
> > >
> > > > The paper claims WARP is a "plug-and-play" module, yet only validates this with gradient-based unlearning. How does WARP's performance and compatibility extend to non-gradient-based unlearning paradigms like those using influence functions or data partitioning?
> > >
> > > > How does WARP's performance and compatibility extend to non-gradient-based unlearning paradigms like those using influence functions or data partitioning?
> > >
> > > Our focus in this work is the practically important setting of **post-hoc approximate unlearning**, where one starts from an already-trained model and does *not* have access to the full training trajectory, per-example statistics, or sharded training logs. This is exactly the regime of NGP, Salun, SF, SCRUB, BT which we cover extensively in the experiments.
> > >
> > > In this setting, WARP is “plug-and-play” in the sense that it only requires:
> > > 1. access to (approximate) gradients of the forget and retain sets at unlearning time, and
> > > 2. the ability to insert a symmetry-based teleportation step after the MU update.
> > >
> > > Influence-function methods and SISA-style data-partitioning approaches are **different in nature**: they require either training-time tracking of per-sample influence / Hessian information, or a pre-defined sharded training protocol with controlled retraining. These methods change the *training pipeline* itself, rather than acting as post-hoc wrappers on an already-deployed model, and are therefore outside the deployment scenario we explicitly target in this paper.
> > >
> > > Conceptually, if such methods expose parameter updates or gradients at unlearning time, WARP could be applied on top of them.

---

> > > > ### Author Response · Authors · 2025-11-25
> > > >
> > > > > The experiments are conducted using relatively small forget-sets (e.g., ~1% of a class). While this is a common scenario, how would the method perform under a more large-scale unlearning request where, for instance, 20% or 50% of the training data must be forgotten? Does the clear distinction between the retain subspace and its null-space still hold when the retain set shrinks significantly?
> > > >
> > > > > How does WARP's effectiveness hold up under large-scale unlearning requests where a significant fraction of the training data must be forgotten?
> > > >
> > > > First, to clarify the setup: in our experiments the forget set is **about 1% of the entire training data, sampled across all classes**, not 1% of a single class. Thus, retain and forget distributions are already quite similar, and the retain subspace is not estimated from a narrow or easy case.
> > > >
> > > > Geometrically, **SVD-based instantiation of WARP** does **not** require a tiny forget set; it only needs a sufficiently rich retain set so that the layer-wise retain activations span a meaningful subspace, and there remain complementary directions where we can push the parameters while keeping retain loss approximately stable. As long as the retain set is still large relative to the feature dimension (which remains true even if, say, 20% of the data is forgotten), the retain span and its complement are well defined and teleportation can still be applied exactly as in our current pipeline.
> > > >
> > > > What changes when the forget fraction becomes very large is primarily the *semantics* of the task: if 30–50% of the original training data must be removed, full or partial retraining on the new retain set becomes a competitive and often preferable baseline, and the value of any **post-hoc** MU method (NGP, Salun, SF, SCRUB, BT,, plus WARP) is less clear. Our focus in this paper is the practically common regime of **small to moderate forget requests**, where retraining is too expensive and approximate MU is actually used. We view systematic evaluation in the “large forget fraction” regime as interesting unlearning direction, but outside the setting considered in this paper.

---

### Official Review · Reviewer_5bBC · 2025-11-01

**Soundness:** 3
**Presentation:** 3
**Contribution:** 3
**Rating:** 6
**Confidence:** 3

**Summary:**

A machine unlearning algorithm that provides defense for reconstruction attacks is introduced in the paper. By applying layer-wise scaling and permutation transformations that preserve the model’s function but relocate its parameters in space, WARP ensures that the parameter difference between the pre-unlearning and post-unlearning models no longer aligns with the forgotten data’s gradients. Experiment results shows that across membership inference and gradient reconstruction attacks, the proposed algorithm reduces privacy leakage with negligible accuracy loss and minimal computational cost.

**Strengths:**

1. The proposed algorithm uses inherent network symmetries to randomize parameter space without retraining or noise injection. It is easy to integrate into any unlearning pipeline, introduces negligible computational costs, and preserves model accuracy.

2. The authors evaluated their algorithm across multiple datasets, unlearning algorithms, and attack types. The inclusion of detailed ablations on transformation modes and frequencies supports the robustness and generality of the proposed defense.

**Weaknesses:**

1. The described process involves per-layer SVDs and null-space projections, which can be expensive for larger models.

2. The work does not provide a theoretical explanation of how teleportation changes the information relationship between parameters and training data. A formal analysis would make the contribution more rigorous.

3. Symmetries are by design invertible. If an attacker can recreate or approximate the teleportation transform (which they probably can given the strong threat model described in the paper), is it possible that they can work their way up and subtract out it's influence to recover the residual forget gradient?

**Questions:**

Please refer to the weaknesses.

---

> ### Author Response · Authors · 2025-11-24
>
> We thank the reviewer for the careful reading and for highlighting that WARP leverages inherent symmetries to randomize parameter space without retraining or noise, integrates easily into MU pipelines, and yields strong privacy gains with negligible accuracy loss across datasets, architectures, and attacks. We also appreciate the positive assessment of soundness, presentation, and contribution. Below we address the reviewer’s concerns about SVD cost, the theoretical explanation of teleportation, and robustness under symmetry-aware attackers.
>
> ---
>
> > **The described process involves per-layer SVDs and null-space projections, which can be expensive for larger models.**
>
> We thank the reviewer for raising this concern. The revised manuscript clarifies that WARP is **symmetry-based rather than SVD-bound**, and introduces a more efficient implementation when SVD-based null-space teleportation is used.
>
> **1. WARP is symmetry-based, not intrinsically SVD-bound**
>
> Section 3.2 now states that WARP is defined over an abstract prediction-preserving symmetry map $T_\phi$. For WARP to operate, we only require an (approximate) symmetry transformation that (i) keeps retain predictions essentially unchanged, while (ii) reshaping the update to suppress forget-set gradients. An SVD-based retain–null-space projector is **one** concrete instantiation of such a symmetry, but not the only one.
>
> We also provide an **SVD-free change-of-basis teleportation** in Appendix D, with experiments in Section 4.4. This variant relies on layer-wise rescaling and permutation symmetries and does not require any SVD over activations. Thus, WARP’s core mechanism is to exploit network symmetries; SVD is a convenient implementation choice for one instantiation, not a requirement of the framework.
>
> **2. FastWARP: replacing full SVD with low-rank PCA and subspace iteration**
>
> We agree that repeatedly computing per-layer SVDs can be expensive as layer width $d_\ell$ grows. To reduce this cost while keeping the same retain–null-space symmetry, the revised paper introduces **FastWARP** (Appendix L), which uses standard low-rank PCA and subspace iteration:
>
> - For a layer with feature dimension $d_\ell$ and retain minibatch $B_r$, we build a covariance
>   $$C_\ell = X_\ell X_\ell^\top,$$
>   where $X_\ell \in \mathbb{R}^{d_\ell \times N}$ collects layer inputs from $B_r$.
> - We obtain an initial low-rank basis $B_\ell$ by eigendecomposing $C_\ell$ and keeping enough eigenvectors to explain a target variance fraction (up to a cap $k_{\max}$).
> - For subsequent teleportation steps, we **track** this basis via a few subspace-iteration steps
>   $$Y = C_\ell B_\ell,\quad B_\ell = \mathrm{qr}(Y),$$
>   rather than recomputing a fresh SVD of the activation matrix.
>
> This changes the per-layer cost from roughly
> $$
> O(|B_r|\, d_\ell^{2})
> $$
> for a thin SVD to
> $$
> O(|B_r|\, d_\ell\, k) + O(d_\ell\, k^{2}),
> $$
> with $k \ll d_\ell$, plus a small fixed number of subspace-iteration updates. Appendix L explicitly connects FastWARP to classical covariance-based PCA, Oja-style online PCA, and block power methods, so it is built on well-established low-rank approximation techniques rather than ad hoc heuristics.
>
> **3. Empirical runtime and implications for larger models**
>
> To quantify the actual overhead, we added **wall-clock timing results**  (Appendix L, Figure 9). The total unlearning time is:
>
> | Method           | Total time (s) | % change vs. base MU |
> |------------------|----------------|----------------------|
> | NGP              | 82.0           | –                    |
> | NGP + WARP (SVD) | 54.0           | −34.1%               |
> | NGP + FastWARP   | 29.0           | −64.6%               |
> | BT               | 30.0           | –                    |
> | BT + WARP (SVD)  | 55.0           | +83.3%               |
> | BT + FastWARP    | 35.0           | +16.7%               |
>
> From the same figure, the teleportation component (hatched segment) shrinks from about **25 s → 3.5 s** for NGP and **30 s → 8 s** for BT when moving from WARP (SVD) to FastWARP, i.e. roughly a **2–3× reduction** in teleportation cost. End-to-end, FastWARP keeps the total runtime comparable to the MU baseline; for NGP, the FastWARP configuration is actually faster than the original unlearning run while achieving a very similar accuracy–privacy trade-off (Figure 10).
>
> For **larger models**, the revised Appendix L (“Scalability to LLMs and calibration of the retain subspace”) discusses how the same low-rank machinery scales: recent work on LLM compression (e.g., SVD-based and residual SVD methods, and importance-weighted low-rank adapters) already applies truncated SVD and covariance-based PCA efficiently to transformer layers with $d_\ell \sim 10^3$–$10^4$. WARP can directly reuse these optimised low-rank operators or plug in symmetry extractors designed for large transformers, without changing the defence logic.
>
> Please see revised paper Appendix L for more details.

---

> ### Author Response · Authors · 2025-11-24
>
> > The work does not provide a theoretical explanation of how teleportation changes the information relationship between parameters and training data. A formal analysis would make the contribution more rigorous.
>
> We agree that a more explicit, formal treatment of how teleportation alters the coupling between parameters and training data is important. The revised manuscript now includes a dedicated analysis in Appendix O that makes this relationship precise.
>
> **1. Teleportation-aware reconstruction bounds (Appendix O)**
>
> Appendix O develops a reconstruction model in which an attacker observes the parameter update and attempts to recover the underlying forget gradient via gradient-matching. In this setting, teleportation is modelled as adding a symmetry-induced component drawn from a distribution over the symmetry family $T_\phi$. The key result is Equation 42 (derived from Theorem 3), which gives a *lower bound* on the attacker’s mean-squared reconstruction error that is an increasing function of the teleportation variance parameter. In other words, as we increase the variance of the symmetry parameters used by WARP, any gradient-matching reconstruction attack in this model is provably forced to incur a larger minimal MSE.
>
> **2. Interpretation for the information relationship between $\theta$ and the forget set**
>
> This formalism makes precise in what sense teleportation weakens the information link between the observed parameter update and the forgotten data. Injecting symmetry-induced variation along $T_\phi$ adds an extra, structured noise component to the observed update that is independent of the particular forget example. As the variance of this component grows, the attacker’s inverse problem becomes increasingly ill-conditioned: the same observed parameter change is compatible with a larger set of possible forget gradients. The lower bound in Equation 42 quantifies this effect by showing that the minimal achievable reconstruction error necessarily grows with the teleportation variance, i.e., the attacker’s signal-to-noise ratio deteriorates as we move further along the symmetry family.
>
> **3. Consistency with symmetry-aware adaptive attacks**
>
> These theoretical predictions are align empirically with the symmetry-aware adaptive reconstruction experiments in Appendix N (Figure 11), where increasing symmetry variance monotonically worsens reconstruction quality even when the attacker is allowed to optimise over the same teleportation family.
>
> ---
>
> > Symmetries are by design invertible. If an attacker can recreate or approximate the teleportation transform (which they probably can given the strong threat model described in the paper), is it possible that they can work their way up and subtract out it's influence to recover the residual forget gradient?
>
>
> We agree that the symmetry maps we use are (approximately) invertible, but invertibility alone does not let the attacker “subtract out” teleportation. What the attacker observes is a *single* net update, which in our analysis decomposes into a data-driven component (e.g. the gradient of the forget-set) plus a symmetry-induced component (teleportation effect) with random parameters; many different pairs $(g_f, \tau)$ (forget gradient and symmetry parameters) can yield the same observed parameter difference. Even if the attacker knows the teleportation family and its variance, they do **not** observe the specific draw of symmetry parameters applied at each step, so the inverse problem is fundamentally non-identifiable.
>
> Formally, Appendix O shows that, in our reconstruction model. Specifically, even an optimal attacker who knows the teleportation mechanism and its distribution, but not the exact symmetry draw (cob values), cannot recover the forget gradient arbitrarily well: the symmetry-induced variability irreducibly degrades the best possible reconstruction.
>
> Empirically, we also evaluate a **symmetry-aware adaptive reconstruction attacker** in Appendix N (Figure 11), which is allowed to optimise *both* the dummy input and the teleportation (change-of-basis) parameters in the same family used by WARP. As we increase symmetry variance, reconstruction quality degrades monotonically and never returns to the non-teleported baseline, and the adaptive attacker closely tracks the non-adaptive teleported attack. Together, these theoretical and empirical results indicate that, despite invertibility of the symmetry maps, an attacker cannot simply invert or subtract teleportation to recover the residual forget gradient.

---

### Official Review · Reviewer_NyvK · 2025-11-01

**Soundness:** 2
**Presentation:** 3
**Contribution:** 2
**Rating:** 2
**Confidence:** 4

**Summary:**

This paper proposes a privacy defense method, WARP, for machine unlearning. Despite being efficient and effective, approximate unlearning methods can leak information about the unlearned instance when an adversary analyzes the difference between models before and after unlearning. To tackle this issue, WARP leverages neural network symmetries to keep updates near a loss-invariant set for the retain data, and to suppress gradients from the forget set. It then applies layer-wise projections to teleport parameters in function-preserving directions. Experiments combining WARP with unlearning methods showing reduced attack success under both black-box and white-box settings.

**Strengths:**

1.	WARP can be applied to both CNNs and ViTs, showing that it does not rely on any particular neural network architecture.
2.	WARP is designed to be a plug-in method and can be adopted with many unlearning methods, as demonstrated in the experiments.
3.	The improvements of WARP are consistent across all the tasks and unlearning methods.

**Weaknesses:**

1.	Each layer’s retain subspace is built from a randomly sampled retain minibatch, where this small batch can misrepresent the full retain set when the retain set is highly diverse. This could result in an inaccurate retain subspace and hinder defense effectiveness.
2.	The paper describes teleportation as “leaving the task loss unchanged up to numerical error”, which is in fact an approximation of a loss-invariant transformation on the retain set. Yet there is no analytical worst-case bound on the retain loss drift. Also, the claim “loss-invariant” may not be accurate since it is not strictly unchanged.
3.	The evaluation in this paper centers around with and without WARP but does not compare against any privacy defenses. Since the contribution of this paper is privacy defense in machine unlearning, it would be better to show how WARP outperforms other privacy defense methods, rather than only showing gains when plug WARP into many unlearning methods.

**Questions:**

Can WARP be applied to popular NLP tasks with LLMs?

---

> ### Author Response · Authors · 2025-11-24
>
> We thank the reviewer for carefully reading the paper and for the positive assessment of key aspects of our work, including that WARP is a plug-in defence usable with multiple unlearning algorithms, applies to both CNNs and ViTs, and yields consistent privacy improvements across tasks and methods. Below we address the reviewer’s main concerns.
>
> ---
>
> > Each layer’s retain subspace is built from a randomly sampled retain minibatch, where this small batch can misrepresent the full retain set when the retain set is highly diverse. This could result in an inaccurate retain subspace and hinder defense effectiveness.
>
> **1. WARP is symmetry-based, not tied to a specific SVD estimate**
>
> For WARP to operate, we only require an approximate prediction-preserving symmetry transformation that can be used to reduce forget-set gradients while moving parameters along approximately loss-invariant directions; an SVD-based retain–null-space projector is one such instance, but not the only one. Section 3.2 now states that WARP is defined over an abstract prediction-preserving symmetry map $T_\phi$; the retain–null-space construction is only one concrete instantiation of this symmetry. We also provide an SVD-free change-of-basis teleportation in Appendix D with experiments in Section 4.4, so WARP itself is **symmetry-based rather than SVD-based**. This leaves room to plug in more advanced symmetry extractors, such as the “Hide \& Seek: Transformer Symmetries Obscure Sharpness \& Riemannian Geometry Finds It” line of work [1], which explicitly recovers symmetries in large-scale transformer language models and low-rank adapters without needing data samples, and could substitute the SVD-based instantiation we use here.
>
> **2. Empirical robustness to small retain minibatches**
>
> Regarding the specific concern about small retain minibatches, we now provide empirical evidence that the defence is **insensitive to the exact batch size** used to estimate the retain subspace. In Appendix P, we vary the retain minibatch size $|B_r|$ over $\{256, 512, 1024, 2048, 4096\}$. Even at $|B_r| = 256$, privacy (measured as $1 - \text{AUC}$ under U-LiRA) changes by at most about $0.01$ in absolute value, and test accuracy stays within roughly $\pm 0.2$ percentage points of $0.796$ (Figure 12). This robustness arises because the retain–null-space instantiation depends mainly on the **dominant retain directions**, which are shared across many retain examples and therefore much less sensitive to the particular minibatch.
>
> **3. Worst-case retain-loss drift remains small and does not affect accuracy**
>
> We also explicitly monitor how approximate projectors affect retain performance. As stated in Appendix L, during FastWARP teleportation we track the retain-set loss over the entire unlearning trajectory and observe that the **worst-case relative drift across all teleportation steps remains below $2\%$**. This bound is taken over the full unlearning run, not just a single step. Across all MU baselines, the corresponding test accuracies in Table 1 and the FastWARP trade-off plot in Figure 10 remain within the typical variability of the underlying unlearning methods. In other words, the small loss drift induced by teleportation does not materially change test accuracy in any of the settings we evaluate, which is exactly the regime where approximate loss invariance is sufficient for our purposes.
>
> **4. Scalability and calibration at LLM scale are discussed, but left to future work**
>
> Finally, we discuss scalability to LLMs and calibration of the retain subspace in Appendix L. In the “Scalability to LLMs and calibration of the retain subspace” paragraph, we note that recent work on large language models already applies truncated SVD and related low-rank factorizations efficiently to full LLM weight matrices, and that weighted low-rank methods (e.g., ESPACE-style importance-weighted PCA [2]) can bias the recovered subspace toward high-influence tokens or examples. In our setting, this corresponds to maintaining a small buffer of high-gradient or high-Fisher retain examples and forming the activation matrix from this pool, yielding a covariance whose top-$k$ eigenspace more faithfully reflects the retain distribution while keeping per-teleportation cost at $O(|B_r| d_\ell k)$. Adapting WARP to full-scale LLM unlearning is therefore a promising but separate direction; in this paper we focus on developing a **general symmetry-based mechanism to mitigate unlearning privacy risk** for standard CNN/ViT models and established MU baselines, and explicitly leave LLM-specific unlearning challenges to future work.
>
> ---
>
> [1] Da Silva, Marvin F., Felix Dangel, and Sageev Oore. "Hide & Seek: Transformer Symmetries Obscure Sharpness & Riemannian Geometry Finds It.", ICML 2025
>
> [2] Sakr, Charbel, and Brucek Khailany. "Espace: Dimensionality reduction of activations for model compression." NeurIPS 2024

---

> > ### Author Response · Authors · 2025-11-24
> >
> > > The paper describes teleportation as “leaving the task loss unchanged up to numerical error”, which is in fact an approximation of a loss-invariant transformation on the retain set. Yet there is no analytical worst-case bound on the retain loss drift. Also, the claim “loss-invariant” may not be accurate since it is not strictly unchanged.
> >
> > We agree that describing teleportation as strictly “loss-invariant” on the retain set was too strong. As discussed in our earlier point **“1. WARP is symmetry-based, not tied to a specific SVD estimate”**, WARP only requires an approximate prediction-preserving symmetry map that keeps the retain loss within a small tolerance, rather than enforcing exact invariance layer by layer. In the revised manuscript we (i) weaken the wording to *approximately loss-preserving* and (ii) make explicit both the optimisation constraint and the empirical behaviour of the retain loss.
> >
> >
> > 1. **Clarified formulation in the method section**
> >
> > In Section 3.2 we now treat teleportation as **constrained to keep retain loss within a small tolerance**, rather than exactly fixed. Concretely, the symmetry move $g$ is selected under
> > $$
> > \big|\ell_r(g\cdot\theta \mid D_r) - \ell_r(\theta \mid D_r)\big| \le \varepsilon,
> > $$
> > so “loss-invariance” is explicitly framed as an *approximate* condition with a tolerance $\varepsilon$.
> >
> >
> > 2. **Empirical worst-case drift over the entire unlearning run**
> >
> > We now systematically track the retain loss throughout the unlearning process. Appendix L reports that, with FastWARP, the **worst-case relative drift in retain loss across all teleportation steps and epochs remains below 2%**. This is measured over the full retain set, not just the minibatch used to build the projector. Across all MU baselines, the corresponding test accuracies in Table 1 and the FastWARP trade-off plot in Figure 10 remain within the usual variability range of the underlying unlearning methods. In other words, the small loss drift induced by teleportation does not materially change test accuracy in any of the settings we evaluate, which is exactly the regime in which approximate loss invariance is sufficient for our purposes.
> >
> > ---
> >
> > > The evaluation in this paper centers around with and without WARP but does not compare against any privacy defenses. Since the contribution of this paper is privacy defense in machine unlearning, it would be better to show how WARP outperforms other privacy defense methods, rather than only showing gains when plug WARP into many unlearning methods.
> >
> > We agree that a comparison to other privacy defences is important. In the revised manuscript we now include an explicit **DP–Langevin baseline** as a strong noise-based defence.
> >
> > - In **Appendix M (“Comparison With DP–Langevin Noise Defences”)**, we implement **projected DP–Langevin unlearning** following [1], applied to the *same* MU objective used for NGP (same loss, clipping, and regularisation form). For each setting, we use their Renyi-DP accountant to map a target privacy level (e.g., $\varepsilon \approx 1$) to the corresponding Gaussian noise scale, and we sweep over $(\eta, C, \lambda)$ exactly as in their setup.
> >
> > - We also clarify that the formal DP guarantees of DP–Langevin require a convex objective and a DP-trained initial model, which do **not** hold in our non-convex post-hoc MU setting starting from a standard, non-DP-trained model. For this reason, we treat DP–Langevin as a **strong noise-based defence**, not as a certified mechanism, and focus on its empirical privacy–utility tradeoff.
> >
> > - **Table 5** reports the direct comparison. To reach attack AUC on the hardest “most-memorised” configuration comparable to WARP, DP–Langevin needs a high noise level (e.g., $\varepsilon \approx 1$), at which point test accuracy drops to about $0.68$, while the attack still achieves AUC $\approx 0.67$. In contrast, **WARP with NGP** attains *lower* attack AUC (around $0.59$) while keeping test accuracy in the $0.79$ range. Thus, for similar or better privacy, DP–Langevin pays a substantially larger utility cost than WARP.
> >
> > Overall, the revised paper now evaluates WARP **both** as a plug-in module across six MU algorithms **and** against a dedicated DP–Langevin noise defence, showing that WARP achieves a more favourable privacy–utility tradeoff in the MU regime we study.

---

### Official Review · Reviewer_J66i · 2025-11-02

**Soundness:** 2
**Presentation:** 3
**Contribution:** 2
**Rating:** 4
**Confidence:** 3

**Summary:**

This paper addresses the privacy risk in approximate machine unlearning (MU), where adversaries can exploit differences between pre- and post-unlearning models to reconstruct forgotten data. To mitigate this, the authors propose WARP, a plug-and-play defense that leverages neural network symmetries to reduce the forget-set's gradient signal and increase parameter distance, thus obfuscating the unlearning process.

**Strengths:**

1.	Connecting neural network symmetry with unlearning privacy is a novel conceptual contribution.
2.	The evaluation uses strong, adaptive attacks (U-LiRA, and a custom white-box attack), making the defense's evaluation robust.

**Weaknesses:**

1.	Significant Computational Overhead is the most significant drawback. The core of this method (see Algorithm 3) requires computing the SVD for the retainer activation matrix $R_l$ of each layer in each transport step. SVD is a computationally intensive operation. The analysis in Appendix J (Figure 8) shows that this adds an average of +27% to the runtime overhead.
2.	This method introduces a large number of new and seemingly highly sensitive hyperparameters: transmission step size $\eta_{tel}$, tradeoff coefficient $\beta$, rank $k$ preserved by SVD, transmission frequency $S$, etc. This is further evidenced by the discussion of the privacy-utility tradeoff in Appendix I (as shown in Figure 7).
3.	In Table 1, for most-memorized samples, NGP+WARP still has an AUC of 0.598, and BT+WARP has an AUC as high as 0.865, which is far from reaching the "immune" level.
4.	The entire method is heuristic. It provides no formal privacy guarantees (e.g., in relation to Differential Privacy). It is unclear if a stronger, adaptive attacker aware of WARP could defeat it.
5.	The reported 92% white-box AUC reduction appears to be a best-case scenario. For other baselines (like NGP or Salun), the improvement is marginal. The paper also admits the method can worsen privacy for SCRUB under certain conditions

**Questions:**

Please refer to the weaknesses.

---

> ### Author Response · Authors · 2025-11-24
>
> We thank the reviewer for carefully engaging with the paper and for highlighting both the novelty of connecting neural network symmetries with unlearning privacy and the strength of our evaluation with U-LiRA and custom white-box attacks. Below, we address the reviewer’s main concerns.
>
> ---
>
> > Significant Computational Overhead is the most significant drawback. The core of this method (see Algorithm 3) requires computing the SVD for the retainer activation matrix
>  of each layer in each transport step. SVD is a computationally intensive operation. The analysis in Appendix J (Figure 8) shows that this adds an average of +27% to the runtime overhead.
>
> **1. WARP is not inherently SVD-based**
>
> For WARP to operate, we only require a (approximate) prediction-preserving symmetry transformation that can be used to reduce forget-set gradients while moving parameters along (approximate) loss-invariant directions; an SVD-based retain–null-space projector is one such instance, but not the only one. Section 3.2 now states that WARP is defined over an abstract prediction-preserving symmetry map $T_\phi$. The retain–null-space construction that uses SVD is only one concrete way to instantiate this symmetry. We also note that an SVD-free change-of-basis teleportation in Appendix D and experiments in Section 4.4, so WARP itself is symmetry-based rather than SVD-based.
>
>
> **2. FastWARP: using standard low-rank PCA via covariance eigendecomposition and subspace iteration**
>
> While SVD-free symmetry mechanisms can be used within WARP, the SVD-based retain–null-space construction is attractive because it gives a clean way to identify directions that preserve retain performance and reduce forget-set gradients. However, as the reviewer correctly notes, full SVD can be computationally expensive. To reduce this cost without changing the underlying symmetry, we introduce **FastWARP** in Appendix L. FastWARP replaces full SVD with covariance-based PCA and subspace iteration, following well-known low-rank approximation methods (e.g., Oja’s rule, randomized SVD and block power iteration). Concretely, for a layer with feature dimension $d_\ell$ and retain minibatch $B_r$:
>
> - we form a layer-wise covariance $C_\ell = X_\ell X_\ell^\top$ from activations,
> - we obtain an initial low-rank basis $B_\ell$ that captures a target fraction of explained variance,
> - and we track this basis over teleportation steps via a small number of subspace iteration updates
>   $$Y = C_\ell B_\ell, \quad B_\ell = qr(Y).$$
>
> This changes the per-layer cost from
> $$
> O(|B_r| \, d_\ell^{2})
> $$
> for a thin SVD to
> $$
> O(|B_r| \, d_\ell \, k) + O(d_\ell \, k^{2}),
> $$
> with $k \ll d_\ell$, and avoids recomputing a full SVD at every teleportation step. Appendix L now explicitly connects FastWARP to these standard PCA / subspace iteration techniques, so it is not an ad hoc or non-standard method.
>
>
> **3. New empirical timing results**
>
> We added explicit wall-clock measurements on ResNet-18 (Appendix L,Figure9). The total unlearning time in seconds is:
>
> | Method           | Total time (s) | % change vs. base MU |
> |------------------|----------------|-----------------------|
> | NGP              | 82.0           | -                  |
> | NGP + WARP (SVD) | 54.0           | -34.1%                |
> | NGP + FastWARP   | 29.0           | -64.6%                |
> | BT               | 30.0           | -                  |
> | BT + WARP (SVD)  | 55.0           | +83.3%                |
> | BT + FastWARP    | 35.0           | +16.7%                |
>
> From the same figure, the teleportation component (hatched part of each bar) drops from about 25 s to 3.5 s for NGP and from about 30 s to 8 s for BT, giving roughly a 2–3× reduction in teleportation cost. Therefore, FastWARP keeps total runtime in the same order as the MU baseline; in NGP it is even faster than the original configuration.
>
> These changes in the main text and Appendix L directly address the concern that SVD makes WARP computationally prohibitive.
>
> ---

---

> > ### Author Response · Authors · 2025-11-24
> >
> > > This method introduces a large number of new and seemingly highly sensitive hyperparameters: transmission step size
> > , tradeoff coefficient
> > , rank
> >  preserved by SVD, transmission frequency
> > , etc. This is further evidenced by the discussion of the privacy-utility tradeoff in Appendix I (as shown in Figure 7).
> >
> > **1. Figure 7 mainly sweeps base NGP hyperparameters**
> >
> > In the revised paper, we clarify that the privacy–utility tradeoff in Figure 7 is driven primarily by the base NGP hyperparameters, not by aggressive tuning of WARP-specific knobs. For both NGP and NGP+WARP, we allocate the same number of hyperparameter-search trials; NGP sweeps its built-in tradeoff parameter (the coefficient on the forget loss that balances forgetting vs. retain accuracy in the original MU objective) and standard optimizer settings, while NGP+WARP sweeps the same NGP grid together with a small set of WARP parameters. The revised text around Figure 7 now states this explicitly, highlighting that the overall tradeoff is mainly controlled by the underlying NGP objective.
> >
> > **2. Algorithm 3 is generic; the experimental schedule is simple and fixed**
> >
> > Algorithm 3 is written in a general form, with a teleportation objective
> >
> > $$
> > L_{tel}(\theta) = \sum_{(x,y)\in B_f} \| \nabla_\theta \ell(f(x;\theta),y) \|_2^2 - \beta \| \theta - \theta_o \|_2^2
> > $$
> > and a trigger condition that could in principle depend on a frequency or a gradient-norm threshold. This was intended to expose the design space, not to suggest that all of these hyperparameters are tuned.
> >
> > In the actual experiments, the effective choices are much simpler and fixed:
> >
> > - we set $\beta = 0$ (no dispersion term is tuned),
> > - and we perform **one teleportation per unlearning epoch, at the beginning of that epoch**, rather than varying a transmission frequency across runs.
> >
> > The revised Section 3.2 and Algorithm 4 now describe this concrete implementation used in all experiments.
> >
> >
> >
> > **3. Only two teleportation hyperparameters matter in practice, and both are insensitive**
> >
> > The teleportation-specific degrees of freedom that actually affect the null-space geometry are:
> >
> > - the retain minibatch size $\lvert B_r \rvert$ used to estimate the retain subspace,
> > - the effective rank of the SVD, parameterised via a target retain-variance fraction (we keep enough singular vectors of $R_\ell(D_r)$ to capture a chosen fraction $k$ percentage of the retain energy).
> >
> > To study the teleportation hyperparameters sensitivity, we added a dedicated ablation in Appendix P:
> >
> > - **Varying retain-variance target.** When we vary the explained-variance threshold over a wide range (for example from 95% up to 99.9%), the induced per-layer ranks $k_\ell$ change substantially. Across this range, privacy (measured as $1 - \text{AUC}$ of U-LiRA) shifts by less than about 0.015 in absolute value, and test accuracy remains in a tight band around 0.79–0.80. The configuration used in the main experiments that produces Table 1 (with retain variance $k \approx 99.3\%$) lies on a flat part of this curve.
> >
> > - **Varying $\lvert B_r \rvert$.** When we change the retain minibatch size $\lvert B_r \rvert$ over $\{256, 512, 1024, 2048, 4096\}$, privacy changes by at most about 0.01 and test accuracy stays within roughly $\pm 0.2$ percentage points of 0.796. Even relatively small retain batches already give a stable subspace; larger batches yield only mild, saturating gains.
> >
> > These results, detailed in the new Appendix P and illustrated in Figure 12, show that WARP’s privacy and utility are **robust** to wide variations in the two teleportation hyperparameters that matter.
> >
> > ---

---

> > > ### Author Response · Authors · 2025-11-24
> > >
> > > > In Table 1, for most-memorized samples, NGP+WARP still has an AUC of 0.598, and BT+WARP has an AUC as high as 0.865, which is far from reaching the "immune" level.
> > >
> > >
> > > WARP is designed as a **plug-in symmetry module** that *systematically* reduces leakage across a wide range of MU methods and threat models, without retraining or DP-style utility collapse. The revised pdf now emphasises this more clearly in the introduction and conclusion.
> > >
> > > Regarding the specific cited numbers:
> > >
> > > - On the **most-memorized subset under a very strong white-box attack**, the baselines themselves are already highly vulnerable: NGP reaches AUC $\approx 0.649$ and BT reaches AUC $\approx 0.902$. With WARP, these drop to $\approx 0.598$ and $\approx 0.865$, respectively. Thus, in this *hardest* subset, the residual gap from an “immune” AUC of 0.5 is largely determined by the underlying MU algorithms (NGP/BT), which are designed to be lightweight and low-overhead, rather than by WARP, whereas SF attains near-immune behaviour only at the cost of substantially higher computational burden.
> > >
> > >
> > > - To put the “immune” target in context, Table 5 (DP–Langevin comparison) shows that even with a heavy DP-style noise level (corresponding to $\varepsilon \approx 1$), the attack on the most-memorized configuration still achieves AUC $\approx 0.671$, and test accuracy drops to about $0.682$. Even a strong formal-noise defence therefore does not reach immunity on this hardest subset, and it pays a much larger utility cost than WARP.
> > >
> > > ---
> > >
> > > > The entire method is heuristic. It provides no formal privacy guarantees (e.g., in relation to Differential Privacy). It is unclear if a stronger, adaptive attacker aware of WARP could defeat it.
> > >
> > > The revised pdf provide formal analysis of how teleportation affects gradient-based reconstruction error, and it evaluates **adaptive** attackers that are aware of WARP.
> > >
> > >
> > >
> > > **1. Teleportation-aware theoretical analysis (Appendix O)**
> > >
> > > The revised manuscript now includes **teleportation-aware information-theoretic bounds** for gradient-based reconstruction (Appendix O). In summary:
> > >
> > > - We derive lower bounds on the expected reconstruction error (in a squared-loss sense) that explicitly depend on the **variance of the symmetry parameters** used by teleportation.
> > > - The analysis in Appendix O (see Equation 42, derived from Theorem 3) shows that, in our simplified model, the reconstruction mean-squared error admits a lower bound that grows with the teleportation variance parameter. Intuitively, increasing this variance injects additional symmetry-induced noise into the update, which decouples the observed parameter change from the underlying forget gradient and forces the attacker to solve a harder, higher-variance optimisation problem, thereby reducing their effective signal-to-noise ratio.
> > >
> > >
> > > **2. Adaptive attackers in membership inference (U-LiRA and GLiR)**
> > >
> > > The main membership inference attacks we use are **already adaptive to WARP**:
> > >
> > > - In both **U-LiRA** and **GLiR**, the attacker **trains proxy models** that follow exactly the same pipeline as the defender:
> > >   - the same base MU algorithm (e.g., NGP, BT) with the same hyperparameters,
> > >   - **plus WARP** with the same teleportation parameters.
> > > - The attacker then exploits these proxy models to learn attack scores or surrogate loss landscapes, conditioned on the presence of teleportation.
> > >
> > > Under this threat model, WARP still consistently reduces both black-box and white-box AUC across all six MU methods (Tables 1–3, radar plots).
> > >
> > >
> > >
> > > **3. Symmetry-aware adaptive reconstruction attacker (Appendix N, Figure 11)**
> > >
> > > To further probe whether a symmetry-aware attacker can “undo” teleportation in the **reconstruction** setting, we add a dedicated experiment in Appendix N:
> > >
> > > - We construct a **symmetry-aware adaptive reconstruction attack** where the attacker jointly optimizes:
> > >   - the dummy input $x$, and
> > >   - the teleportation (change-of-basis) parameters $\tau$ that parameterize the same symmetry family used by WARP.
> > > - That is, the attacker chooses both $x$ and $\tau$ to minimize the distance between the shadowed update and the actual unlearned parameters, directly incorporating knowledge of the teleportation map.
> > >
> > > Figure 11 reports the effect of this attack as we increase the variance of the change-of-basis scales. The key empirical findings are:
> > >
> > > - As symmetry variance increases, **reconstruction quality degrades monotonically** in both feature MSE and LPIPS.
> > > - The adaptive attacker **never recovers** the reconstruction quality of the non-teleported baseline. In fact, higher symmetry variance makes the attack strictly worse, in line with the theoretical bounds in Appendix O.
> > > - Moreover, the symmetry-aware adaptive attack achieves reconstruction metrics that are very close to those of the non-adaptive teleported attack, indicating that explicitly optimising over the teleportation parameters does not yield additional attack success.
> > >
> > >
> > > ---

---

> > > > ### Author Response · Authors · 2025-11-24
> > > >
> > > > > The reported 92% white-box AUC reduction appears to be a best-case scenario. For other baselines (like NGP or Salun), the improvement is marginal. The paper also admits the method can worsen privacy for SCRUB under certain conditions.
> > > >
> > > > The 92% figure is indeed the largest single white-box improvement we report, but it is not an isolated best case. WARP is designed as a **plug-in symmetry module** that consistently reduces leakage across different MU methods, not as a one-off tweak for a single baseline.
> > > >
> > > > Across all six unlearning algorithms in Figure 3 (white-box MIA), the **averages of the per-method relative improvements** in white-box AUC is about **35%**, and the average reduction in TPR@0.1 is about **50%**. Methods such as SF and BT see particularly large gains, but even for NGP and Salun the white-box AUC and TPR@0.1 still move in the favourable direction once WARP is applied; the improvements are smaller there because these baselines already start closer to AUC ≈ 0.5.
> > > >
> > > > Regarding SCRUB, the “worsening” we mention is a very narrow corner case: in the ROC analysis, WARP slightly raises TPR only at $FPR < 10^{-3}$, while **improving all aggregate metrics** (AUC and TPR at standard low FPRs) for both black-box and white-box evaluations. We intentionally kept this behaviour in the paper for transparency. The overall picture from Table 1 and Figure 3 is that WARP yields **systematic and often substantial reductions** in attack success across methods, with only this small SCRUB corner region deviating from the general trend.

---

### Author Response · Authors · 2025-11-24
**General Response to Reviewers and Summary of Changes**

**`[EDIT: For convenience, we have uploaded a latexdiff version of the manuscript as supplementary material, where all added or modified text is highlighted in blue.]`**

---

We thank all reviewers for their thoughtful and constructive feedback, and for highlighting several strengths of our work: the conceptual link between neural network symmetries and unlearning privacy, the plug-in nature of WARP across multiple MU algorithms and architectures (CNNs and ViTs), the use of strong adaptive attacks (U-LiRA, GLiR, and white-box reconstruction), and the consistent privacy gains achieved with only minor accuracy degradation. The revision directly targets the main shared concerns across reviews, namely: clarifying the role of symmetries versus SVD, reducing and better characterising computational overhead, providing a more formal understanding of how teleportation affects information leakage, evaluating stronger symmetry-aware adaptive attackers, comparing against DP-based noise defences, and analysing the sensitivity of the few teleportation hyperparameters that matter in practice.

1. **Clarifying the contribution and the role of symmetries.**
Several reviewers interpreted WARP as an SVD-based symmetry method. We now clarify in Section 3.2 that
• WARP is defined around an **abstract prediction-preserving symmetry map** $T_\phi$, and SVD is only *one* way to instantiate such a symmetry.
• We explicitly present two instantiations: an SVD-based retain–null-space projection (primary variant, Sec. 3.2) and an **SVD-free (and data-free) change-of-basis teleportation** (Appendix D).
• The framework uses symmetry transformations, but **discovering new symmetry families is outside the scope** of this work and remains an active research area. In the paper we instantiate symmetry-based teleportation with these two mechanisms to show that such symmetry moves suppress forget-set gradients and increase symmetry-preserving dispersion, thereby making unlearning attacks harder.

2. **SVD cost and the FastWARP variant.**
To address concerns that per-layer SVD may be expensive, we introduce **FastWARP**, which replaces full SVD with low-rank PCA via covariance eigendecomposition and subspace iteration.
• A new subsection “Approximate Null-Space Teleportation’’ (Appendix L) details the method.
• A new runtime figure shows that FastWARP achieves a 2–3× reduction in teleportation cost and produces only small percentage overhead relative to the MU baseline (Figure 9).
• A separate privacy–utility comparison shows that FastWARP matches the behaviour of exact WARP (Figure 10).

3. **Theoretical analysis and a symmetry-aware adaptive attacker.**
Reviewers asked how teleportation alters the information relationship between parameters and forgotten data, and whether a stronger attacker could undo it.
• We add **teleportation-aware information-theoretic bounds** (Appendix O), theoretically showing that the lower bound on reconstruction MSE increases with the symmetry variance along $T_\phi$, and thus raises expected reconstruction error.
• We introduce a **symmetry-aware adaptive reconstruction attack**, jointly optimizing the dummy input and COB parameters (Appendix N).
• New results show **monotonic degradation** of reconstruction quality as symmetry variance increases, and that reconstruction never returns to the non-teleported baseline (Figure 11). These findings are consistent with the theoretical bounds.

4. **Comparison with DP–Langevin defences.**
In response to requests for comparisons to other defences—specifically DP-based methods—we add a dedicated baseline.
• We implement projected DP–Langevin unlearning following [1] (Appendix M).
• We explain why their formal DP guarantees do not apply in our non-convex setting starting from a standard, non-DP-trained model, and therefore evaluate DP–Langevin purely as a strong noise-based defence.
• Results show that achieving privacy comparable to WARP requires substantially more noise, leading to significantly larger accuracy drops (Table 5).

5. **Retain subspace, loss drift, and hyperparameter robustness.**
To address concerns about retain-subspace estimation and stability:
• We report empirical **retain-loss and prediction drift** during teleportation in Appendix L, showing that retain performance remains essentially unchanged.
• We add a compact **hyperparameter sensitivity study** (Appendix P), demonstrating that WARP is robust: privacy and accuracy vary only slightly across wide ranges (Figure 12).

[1] Eli Chien, Haoyu Wang, Ziang Chen, and Pan Li. Langevin unlearning: A new perspective of noisy
gradient descent for machine unlearning. Advances in neural information processing systems

---

### Author Response · Authors · 2025-12-03
**Rebuttal Summary**

We thank all four reviewers, **J66i, NyvK, 5bBC, and puub**, for their thoughtful and constructive feedback. Several strengths were consistently highlighted: the **novel link between neural network symmetries and unlearning privacy** (J66i, 5bBC, puub), the fact that **WARP is a plug-in defence usable with multiple MU algorithms and architectures (CNNs and ViTs) with consistent gains** (NyvK, 5bBC, and implicitly puub), and the **strong empirical protection against both black-box and white-box attacks with negligible accuracy loss** (J66i on strong adaptive attacks; 5bBC and puub on robustness and utility). We are grateful for these positive assessments of contribution, soundness, and presentation.

The revision directly targets the main concerns raised across the reviews:

- **Computational overhead and SVD scalability (J66i, 5bBC, puub), and LLM-scale practicality / NLP tasks (NyvK, puub).**
  We clarify that **WARP is symmetry-based rather than SVD-bound** (Sec. 3.2, App. D), and introduce the **FastWARP low-rank PCA plus subspace-iteration variant** (Alg. 4) with explicit wall-clock timings showing **about 3× lower teleportation cost** and overall runtime comparable to, or better than, the MU baselines (App. L, Figs. 9–10). Appendix L also includes a dedicated subsection titled **“Scalability to LLMs and calibration of the retain subspace”**, which discusses how the same low-rank machinery and existing LLM-compression techniques can be reused at LLM settings.

- **Retain-subspace estimation, small retain minibatches, and “loss-invariance” wording / drift (NyvK), and hyperparameter sensitivity (J66i).**
  We make explicit that WARP uses **approximately prediction-preserving symmetries**, not exact layerwise invariance (Sec. 3.2), report that **worst-case retain-loss drift over the full unlearning run stays below about 2% with stable test accuracy** (App. L), and add a dedicated ablation showing that **privacy and utility are robust to wide ranges of retain-variance thresholds and retain-batch sizes** (App. P, Fig. 12). This addresses concerns that small retain minibatches or sensitive hyperparameters could undermine the defence.

- **Formal explanation, guarantees, and symmetry-aware adaptive attackers (J66i, 5bBC).**
  Appendix O develops **teleportation-aware reconstruction bounds**, showing that the **lower bound on reconstruction MSE increases with the variance of the symmetry parameters** (Eq. (42)), which formalises how teleportation weakens the information link between observed updates and forget gradients. Appendix N then introduces a **symmetry-aware adaptive reconstruction attack** that optimises both the dummy input and teleportation parameters; as symmetry variance increases, **reconstruction quality degrades monotonically and never recovers the non-teleported baseline** (App. N, Fig. 11), consistent with the theory and addressing concerns that invertible symmetries could simply be “undone”.

- **Comparison to DP-style defences and the privacy–utility trade-off (NyvK, puub, and related to J66i).**
  Appendix M adds a **DP–Langevin unlearning baseline** implemented on the same objective as NGP. In our non-convex post hoc MU setting, the formal DP guarantees do not apply, so we focus on the empirical trade-off. The new results show that matching the attack AUC achieved by NGP+WARP would require DP–Langevin to add enough noise that test accuracy drops substantially, and even then the AUC remains worse than NGP+WARP. For configurations that keep test accuracy close to the original model, DP–Langevin exhibits clearly weaker privacy (Table 5). This clarifies how the symmetry-based defence compares to a strong noise-based alternative.

- **Scope of “plug-and-play”, other MU paradigms, and larger forget fractions (puub).**
  We clarify that WARP explicitly targets the **post hoc approximate MU regime** (NGP, Salun, SF, SCRUB, BT) starting from an already trained non DP model, where it is plug-in because it only needs retain and forget gradients plus a symmetry step. Methods that modify the training pipeline, such as influence-function or SISA-style approaches, and very large forget fractions where full or partial retraining becomes a natural competitor, are noted as interesting directions but **outside the scope of the current paper**.

---

### Meta-Review · Area_Chair_3LhH · 2026-01-06

**Summary:**

This paper introduces WARP (Weight Teleportation for Attack-Resilient Protocols), which is a defense mechanism that can be applied to any unlearned model to make it resilient against attacks that have access to the model before and after unlearning. WARP is "plug-and-play" and can be used together with any unlearning algorithm in principle. The authors claim that approximate unlearning methods suffer from privacy vulnerabilities due to the unlearned models they produce remaining close to the original model, and due to the updates they perform to arrive to the unlearned model aligning too strongly with the gradients of the forget set.
WARP addresses these issues by leveraging neural network symmetries, i.e. permutations of the model weights that leave the neural network function output invariant. It "teleports" the model weights to a functionally (near-) equivalent point in parameter space that is further away from the original model. Specifically it finds a point that suppresses forget set gradients while only moving along directions that don't affect the retain set performance. The goal is to protect the forget set from privacy attacks without sacrificing utility.

The main concerns that the reviewers raised are:
- **C1**. Computational overhead due to computing SVD of each layer, adding +27% to runtime overhead (Reviewer J66i, Reviewer 5bBC). Scalability for larger models (Reviewer puub).
- **C2**. Large number of seemingly-sensitive hyperparameters (Reviewer J66i).
- **C3**. WARP is far from reaching “immunity” to attacks, while it can sometimes even worsen privacy for some methods (Reviewer J66i).
- **C4**. Heuristic method, lacking formal guarantees, possible to defeat by stronger WARP-aware attackers (Reviewer J66i, Reviewer NyvK, Reviewer 5bBC).
- **C5**. Missing comparisons against privacy defenses (Reviewer NyvK, Reviewer puub).
- **C6**. Limited evidence of plug-and-play with non-gradient based unlearning algorithms. And no evidence on larger forget sets, 20% or 50% of the training dataset (Reviewer puub).

**Reviewer Concerns:**

The authors have made extensive efforts to address all reviewer concerns during the rebuttal (see detailed comments below) and most key concerns have been addressed sufficiently, including the concern of computational overhead of the method, theoretical analysis, hyperparameter sensitivity, privacy-based baselines and adaptive WARP-aware attacks. These concerns were addressed comprehensively via detailed discussions, theoretical results and several new experiments.
Minor issues remain, like adding experiments with larger forget set sizes but I don't feel that that stands in the way of acceptance.

Overall, the paper has several strengths that the reviewers pointed out, including making a novel conceptual link between neural network symmetry and unlearning and privacy, plug-and-play versatility, and strong empirical results, against even strong, adaptive attacks while exhibiting good utility preservation. The methods performs favourably compared to a DP-based approach that the authors added during the rebuttal too, and has some theoretical backing for its strong performance. Based on this, I recommend acceptance.

**Reviewer Scores:**

**Reviewer J66i**

The authors address C1 by clarifying that WARP does not necessarily have to rely on SVD; that is just one option but other ways to instantiate a prediction-preserving symmetry map are also possible, and they provide a different option in Appendix D using change-of-basis reparameterization. They provide some experiments for this in the context of the reconstruction attack, showing good results there (but experiments with this variant for other settings are missing). Further, the note that Appendix L introduces FastWARP which replaces full SVD with covariance-based PCA and subspace iteration. They performed additional experiments on ResNet-18, showing that this variant leads in a 2-3x reduction in teleportation cost, which is quite significant. Overall, I find these results compelling and I think they address the concern of computational overhead well.

To address C2, the authors clarify that they use the same hyperparameter budget for tuning the base algorithm as they do for tuning that algorithm+WARP (Figure 7 shows this in the context of the NegGrad+ algorithm). They also clarify that several hyperparameters are fixed and not tuned, e.g. performing always just one teleportation at the start of each unlearning epoch. They claim that the only hyperparameters that are important to tune are the batch size of the retain set and effective rank of SVD. The authors added new experiments to investigate the sensitivity of these hyperparameters, finding that across a wide range of values for these hyperparameters, privacy shifts by only small amounts and so does test accuracy. I also find this response very convincing.

To address C3, the authors explain that when the WARPed unlearned models aren’t very immune, the baselines themselves are highly vulnerable, and even applying heavy Differential Privacy is far from achieving immunity, showing that WARP’s performance is around the frontier of what is possible currently. The worsening of privacy mentioned for SCRUB occurs in a narrow corner case. I find this to be a compelling argument for why WARP is a strong method despite its inability to achieve complete immunity.

Finally, to address C4, the authors have added a theoretical analysis of how teleportation affects gradient-based reconstruction error, under some assumptions. They also argue that the attackers they use are already WARP-aware, as they rely on building shadow models using the same algorithm as the “defender” (which includes WARP). This is a very strong threat model. To address this concern more comprehensively, they run experiments with a new attacker that directly optimizes parameters of the teleportation map. They show that the attacker can’t recover the reconstruction quality that it obtains when operating against a non-teleported baseline. Again, I find this response comprehensive and I imagine that the reviewer may also have agreed that their concerns are sufficiently addressed to meet the bar for acceptance.


**Reviewer NyvK**

The authors address concerns about the reliance on the batch size parameter and the lack of theoretical guarantees as described above for Reviewer J66i.

To address the reviewer’s remark about imprecise claims, they revised the formulation to state that WRAP is constrained to keep retain loss stable but with a small tolerance, rather than precisely being “loss invariant”. To obtain a worst-case drift for the retain loss, they report this quantity throughout the unlearning process, showing that it remains below 2%, and does not substantially change the test accuracy either, addressing the reviewer’s concern.

To address C5, the authors add a new comparison against DP-Langevin unlearning as a baseline. To reach comparable attack resilience as their method, DP-Langevin needs to add a noise level that ends up deteriorating model utility substantially. WARP achieves better defense, in the settings investigated, while maintaining higher accuracy. I find these new results convincing and sufficiently addressing the reviewer’s concerns, making it likely that the reviewer would have increased their score.

**Reviewer 5bBC**
The authors address C1 well, as for Reviewer J66i.
The authors also mention the theoretical analysis they performed during the rebuttal, characterizing how teleportation disrupts the information link between the model parameters and the training data. While the analysis involves some approximations and simplifications, it is insightful and sufficiently addresses the reviewer’s concern in my opinion.
Finally, the authors also push back against the thought that, because symmetry maps are invertable, the attacker can undo the defense. In particular, the authors argue that this is challenging since the attacker observes the final update, which is a composition of the teleportation with the gradient update of the base unlearning algorithm. The authors also mention their new results on the symmetry-aware adaptive attacker they experimented with for the reconstruction task, with promising results (see above in the comments about Reviewer J66i). Overall, the rebuttal addresses all concerns of this reviewer sufficiently in my opinion. The reviewer was already positive about the paper, recommending weak accept initially, and I imagine that these responses would have only increased the reviewer’s confidence in acceptance.

**Reviewer puub**

The authors address C1 as in previous reviewers, adding scalability discussion here for LLM-scale models too. While they don’t include large-scale experiments, they added a discussion to the revised paper about how their projector construction can be adjusted to scale to larger models by leveraging existing large-model compression techniques.

The authors address the lack of privacy baselines well by mentioning their new results with DP-Langevin (see above for Reviewer NyvK).

To address the issue of compatibility with non-gradient-based methods, the authors mention that their focus is the post-hoc approximate unlearning setting, whereas methods like SISA are not post-hoc (they require dedicated architecture changes that occur before the original training is carried out). They reiterate the WARP can be applied whenever a post-hoc method exposes gradients at unlearning time, which seems a reasonable scoping to me and in line with the experiments in the paper.

Finally, the authors comment on the suitability of their method to different forget set sizes, clarifying that the requirement is that the retain set is sufficiently large/rich to span a meaningful subspace, so that there are directions where the parameters can be pushed while keeping the retain set performance approximately stable. They clarify that their forget set size is 1% of the training dataset (not 1% of the class size), meaning it’s larger than the reviewer may have thought. But they do not present additional empirical results with different forget set sizes. While I think that would have strengthened the paper even more, I don’t feel like it’s strictly necessary for acceptance.
This reviewer was already positive about the paper, recommending weak accept initially, and I imagine that these responses would have only increased the reviewer’s confidence in acceptance.

---

### Decision · Program_Chairs · 2026-01-26

Accept (Poster)